# MAGO: Beyond Fixed Hyperparameters with Multi-Objective Pareto Optimization for Hybrid LLM Reasoning

**Hongcheng Ding**[*][†]
Dongfang College,
Zhejiang University of
Finance and Economics

**Xuanze Zhao**[*]
Dongfang College,
Zhejiang University of
Finance and Economics

**Ruiting Deng**
Yunnan University of
Finance and Economics

**Shamsul Nahar Abdullah**
INTI International University

**Deshinta Arrova Dewi**
INTI International University

**Qingyu Liu**
SEGi University

## Abstract

Large language models (LLMs) with advanced step-by-step reasoning capabilities have achieved remarkable performance in complex problem-solving through chain-of-thought (CoT) reasoning. However, uniformly applying elaborate reasoning to all queries creates substantial computational inefficiency, as many problems can be solved directly without extended reasoning chains. Current hybrid reasoning approaches rely on static hyperparameters and heuristic single-objective optimization, leading to suboptimal trade-offs and poor adaptation to varying task complexities. To address these limitations, we propose a multi-objective adaptive generation optimization (MAGO) framework, which integrates multi-objective optimization with dynamic adaptive weighting into hybrid reasoning. MAGO optimizes three competing objectives simultaneously: accuracy (maintaining solution correctness), efficiency (minimizing computational costs through appropriate mode selection), and calibration (ensuring mode selection aligns with model capabilities). The framework employs Pareto frontier maintenance with correlation-aware optimization to automatically explore the full trade-off space, avoiding the spatial constraints that limit fixed-weight approaches to narrow cone-shaped regions of the objective space. Unlike existing methods requiring manual hyperparameter tuning, MAGO's Pareto optimization dynamically adapts weights based on task complexity and training progress, achieving principled and adaptive decision-making across varying problem complexities. Comprehensive evaluation on mathematical reasoning benchmarks including AIME, Minerva Algebra, MATH-500, and GSM-8K shows $2.2\times$ to $3\times$ token-efficiency gains and relative accuracy improvements of $0.6\%$ to $9.4\%$ over heuristic baselines, while remaining competitive with the strongest task-specific models. Additional experiments on CommonsenseQA and MedQA further confirm the framework's generalizability beyond mathematics, achieving 1 to $2\%$ higher accuracy and approximately $2\times$ efficiency improvement without additional fine-tuning.

## 1 Introduction

Recent breakthroughs in step-by-step reasoning capabilities have enabled LLMs to achieve unprecedented performance in complex problem-solving. Reasoning-enabled models such as DeepSeek-R1 (DeepSeek-AI, 2025) and Claude (Anthropic, 2025) employ CoT reasoning (Wei et al., 2022) to decompose complex problems into manageable sub-steps, thereby simulating human cognitive processes (Nye et al., 2021; Jung et al., 2022). This paradigm has proven particularly effective in mathematical reasoning (Hendrycks et al., 2021; Cobbe et al., 2021) and logical inference tasks (Saha et al., 2020; Wang et al., 2023).

---

[*]Equal contributions
[†]Corresponding Author: i24025877@student.newinti.edu.my (INTI International University)

Figure 1: (A) Traditional single-objective approaches and (B) MAGO's multi-objective framework.

However, uniformly applying elaborate reasoning to all queries creates significant efficiency problems in practical deployment scenarios. Large-scale deployment scenarios must handle diverse query types ranging from simple factual questions requiring direct answers to complex multi-step problems necessitating extensive reasoning (Rajpurkar et al., 2018; Khashabi et al., 2020). Indiscriminate use of reasoning models for all inputs leads to substantial computational waste, as reasoning models generate hundreds to thousands of tokens for problems that could be solved with direct answers, resulting in 5 to 20 times higher resource consumption compared to non-reasoning approaches (Kaplan et al., 2020; Hoffmann et al., 2022; Suzgun et al., 2022; Fu et al., 2023).

To address the substantial computational costs and resource consumption inherent in reasoning-enabled models, current research has concentrated on several key directions to improve inference efficiency. Hybrid reasoning mode selection approaches develop systems that dynamically choose between detailed reasoning and concise response generation through learnable control mechanisms (Fang et al., 2025; Zelikman et al., 2024; Raposo et al., 2024), utilizing specialized optimization algorithms for adaptive mode switching. Test-time compute scaling techniques allocate computational resources dynamically during inference to optimize the trade-off between accuracy and efficiency (Snell et al., 2024; Zhang et al., 2025; Lyu et al., 2025), enabling models to achieve better performance through adaptive inference-time computation rather than larger model parameters. Token-budget-aware reasoning methods explicitly incorporate computational cost constraints into the reasoning process (Han et al., 2024), developing frameworks that balance reasoning depth with predefined computational budgets. However, these methods often produce suboptimal solutions that excel in one aspect (such as accuracy or efficiency) while sacrificing others.

To address these challenges, we propose the MAGO framework, a theoretically grounded approach that reformulates hybrid reasoning as a multi-objective optimization problem. MAGO incorporates dynamic weight adaptation mechanisms that adjust with training progress and implements Pareto frontier maintenance (Deb et al., 2002; Miettinen, 1999) with correlation-aware weight selection to support more refined reasoning decisions. This method eliminates the need for manual hyperparameter tuning while achieving mathematically sound trade-offs among three competing objectives: accuracy (maintaining solution correctness), efficiency (minimizing computational costs), and decision calibration (ensuring mode selection aligns with the model's actual capabilities) (see Figure 1). The main contributions of this paper are as follows:

- We identify two performance gaps in existing hybrid reasoning systems: (1) static weight configurations lead to model under-performance across different scenarios, and (2) strong correlations between objectives cause multi-objective optimization to under-perform.

- We propose MAGO, a multi-objective optimization framework addressing these gaps through: (1) reformulating hybrid reasoning as a multi-objective optimization problem, (2) using Pareto optimization for dynamic weight selection, and (3) achieving end-to-end integration from training to deployment with zero inference overhead.

- Our framework achieves 2.2x to 3x computational efficiency improvements while simultaneously improving accuracy by 0.6% to 9.4% across mathematical reasoning benchmarks. Cross-domain evaluation on CommonsenseQA and MedQA demonstrates generalizability beyond mathematics without fine-tuning.

## 2 MOTIVATION

In this section, we introduce static weight challenges and multi-objective optimization challenges.

**Challenge #1: Static Weight.** Current hybrid reasoning approaches (Fang et al., 2025; Shao et al., 2024) rely on fixed hyperparameters that fail to adapt to varying task complexities across different queries and training datasets. We make a series of attempts across various $\alpha$ values on mathematical reasoning benchmarks to explore this limitation. The results reveal three critical limitations of static weighting schemes. First, different $\alpha$ values cause severe mode selection imbalances (Figure 2A): $\alpha$

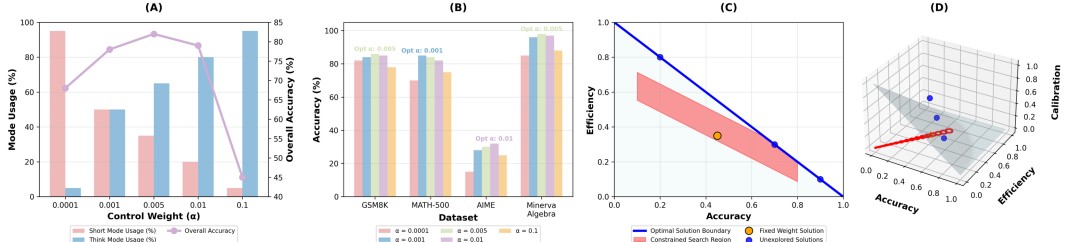

Figure 2: Static weight limitations in hybrid reasoning optimization. (A) Mode selection imbalance across control weight $\alpha$ values with accuracy performance. (B) Dataset-specific $\alpha$ sensitivity across mathematical reasoning benchmarks. (C) Fixed weight search constraints in objective space. (D) Cone-shaped optimization limitations in multi-objective landscape.

= 0.0001 leads to over 90% short-mode usage, sacrificing accuracy on complex problems, while $\alpha$ = 0.01 results in over 80% think-mode usage, negating efficiency gains. Second, optimal $\alpha$ varies significantly across datasets (Figure 2B), and no single fixed weight achieves consistent performance across diverse problem types. Third, exhaustive hyperparameter search for optimal $\alpha$ values is computationally prohibitive, requiring independent model training for each configuration with costs scaling linearly with search space size. These limitations demonstrate that static approaches cannot accommodate the inherent variability in problem complexity and dataset characteristics while remaining computationally feasible. More details can be found in Appendix A.1.

**Challenge #2: Multi-objective Optimization.** The hybrid reasoning problem involves three competing objectives: accuracy, efficiency, and decision calibration. These objectives exhibit interdependencies creating optimization challenges. High accuracy often requires longer reasoning chains, creating tension with efficiency goals. Decision calibration considerations may favor conservative mode selection strategies, potentially affecting both accuracy and efficiency outcomes (Song et al., 2024; Wilde et al., 2024; Albeaik et al., 2024). Traditional single-weight approaches constrain optimization to narrow regions within the objective space, as illustrated in Figure 2C. Fixed weight values restrict the search trajectory to predetermined directions, preventing exploration of alternative regions with superior solutions. This spatial constraint confines optimization to limited cone-shaped regions (Figure 2D), missing optimal solutions in unexplored objective space areas. The limitation becomes pronounced when objectives exhibit different gradient scales and convergence rates, causing premature convergence to spatially constrained local optima rather than exploring the full solution landscape. More details can be found in Appendix A.2.

## 3 BACKGROUND

In this section, we present reinforcement learning and multi-objective pareto optimization.

**Reinforcement Learning.** GRPO (Shao et al., 2024) provides the foundation for training hybrid reasoning models through reinforcement learning. In this framework, $x$ represents an input query or problem that requires the model to generate a response. Given control tokens $\mathcal{C} = \{\texttt{<short>}, \texttt{<think>}\}$, the hybrid reasoning model is parameterized as a policy $\pi_\theta$:

$$\pi_\theta(c, a|x) = \pi_\theta(c|x) \cdot \pi_\theta(a|x, c), \tag{1}$$

where $c \in \mathcal{C}$ denotes the reasoning mode selection and $a$ represents the generated response sequence. The sequence $a_i = (a_{i,0}, \ldots, a_{i,T_i})$ has length $T_i + 1$, where $a_{i,0} \in \mathcal{C}$ is the control token and $(a_{i,1}, \ldots, a_{i,T_i})$ form the response (Shao et al., 2024).

The standard GRPO objective treats all tokens uniformly through a single normalization factor:

$$J_{\text{GRPO}}(\theta) = \mathbb{E}_{x, a_i} \left[ \frac{1}{G} \sum_{k=1}^{G} \left( \frac{1}{T_k + 1} \left[ L_{k,0}(\theta) + \sum_{j=1}^{T_k} L_{k,j}(\theta) \right] - \beta D_{\text{KL}}[\pi_\theta(\cdot|x) \| \pi_{\text{ref}}(\cdot|x)] \right) \right], \tag{2}$$

where $L_{k,0}(\theta)$ and $L_{k,j}(\theta)$ represent control and response token losses, respectively. This creates two imbalances: the control token is overshadowed by $T_k$ response tokens, and longer sequences

suppress control gradients via the shared normalization $\frac{1}{T_k+1}$. DeGRPO (Fang et al., 2025) introduces separate normalization and a weight parameter $\alpha$ to balance mode selection against response accuracy, preventing mode collapse. More details can be found in Appendix A.3.

**Multi-objective Pareto Optimization.** Multi-objective optimization addresses problems with multiple competing objectives that cannot be simultaneously optimized. Rather than seeking a single optimal solution, the goal of Pareto optimization is to find the set of Pareto-optimal solutions:

$$\mathcal{P} = \{x^* \in \mathcal{X} : \nexists x \in \mathcal{X}, \mathbf{f}(x) \preceq \mathbf{f}(x^*), \mathbf{f}(x) \neq \mathbf{f}(x^*)\}, \tag{3}$$

where $\mathbf{f}(x) = [f_1(x), f_2(x), \ldots, f_m(x)]$ represents the objective vector, and $\preceq$ denotes Pareto dominance (Deb et al., 2023; Feng et al., 2021). Traditional single-objective approaches using fixed weight combinations $\sum_i \lambda_i f_i(x)$ often fail to capture the full trade-off space, as they restrict optimization to predetermined directions in the objective space. Multi-objective methods enable exploration of diverse trade-offs by adapting weights dynamically based on the problem characteristics and solution quality. More details can be found in Appendix A.4.

## 4 METHOD

In this section, we introduce the problem formulation and then present our solutions, including the MAGO framework, Pareto frontier maintenance, and end-to-end integration.

### 4.1 PROBLEM FORMULATION

In order to address the *Challenge #1* mentioned in previous sections, we formulate hybrid reasoning training as a dynamic adaptive optimization problem:

$$J(\theta) = \mathbb{E}_{x,a_i} \left[ \frac{1}{G} \sum_{k=1}^{G} \left( \underbrace{m(x)}_{\text{adaptive}} L_{k,0}(\theta) + \frac{1}{T_k} \sum_{j=1}^{T_k} L_{k,j}(\theta) - \beta D_{\text{KL}}[\pi_\theta(\cdot|x) \| \pi_{\text{ref}}(\cdot|x)] \right) \right], \tag{4}$$

where $m(x)$ represents an adaptive weighting mechanism that adjusts based on input characteristics. Unlike existing approaches that rely on fixed hyperparameters, $m(x)$ dynamically adapts to balance competing training objectives without requiring manual tuning or hyperparameter search.

### 4.2 MULTI-OBJECTIVE ADAPTIVE GENERATION OPTIMIZATION

To realize the adaptive weighting mechanism $m(x)$ introduced in Equation 4, we propose MAGO that dynamically balances competing objectives. The framework integrates three objectives:

$$m_{\text{MAGO}}(x) = \beta_1 \cdot S_{\text{accuracy}}(x) + \beta_2 \cdot S_{\text{efficiency}}(x) + \beta_3 \cdot S_{\text{calibration}}(x), \tag{5}$$

where $(\beta_1, \beta_2, \beta_3)$ are dynamically adapted weights that automatically balance the three competing objectives without manual tuning, and the three task-specific objectives are defined below:

**Accuracy Objective.** The accuracy objective $S_{\text{accuracy}}(x)$ measures the correctness of responses generated under different reasoning modes:

$$S_{\text{accuracy}}(x) = \mathbb{E}_{(c,a) \sim \pi_\theta(c,a|x)}[\mathbb{I}(\phi(a) = y^*)], \tag{6}$$

where $\mathbb{I}(\cdot)$ is the indicator function, $y^*$ is the ground-truth answer, and $\phi(a)$ extracts the final answer from the response sequence $a$. This function parses the generated response to identify the concluding numerical or textual answer, enabling direct comparison with the ground truth regardless of reasoning mode length.

**Efficiency Objective.** The efficiency objective $S_{\text{efficiency}}(x)$ captures the potential for computational savings through appropriate mode selection by measuring the expected response efficiency:

$$S_{\text{efficiency}}(x) = \mathbb{E}_{(c,a) \sim \pi_\theta(c,a|x)} \left[ 1 - \frac{|a|}{T_{\max}} \right], \tag{7}$$

where $|a|$ denotes the token length of the generated response sequence $a$, and $T_{\max}$ represents the maximum allowed sequence length. The normalization term $\frac{|a|}{T_{\max}}$ measures the relative computational cost, and subtracting from 1 converts this to an efficiency score where values approaching 1 indicate highly efficient responses. This expectation is computed by sampling responses and calculating their average normalized efficiency.

**Calibration Objective.** The decision calibration objective addresses a critical challenge in hybrid reasoning: ensuring that the model's mode selection decisions are well-calibrated with its problem-solving capabilities. Specifically, when the model chooses the short reasoning mode, it should be confident that it can solve the problem correctly without extended reasoning. Conversely, when it selects the think mode, this should indicate that the problem requires more elaborate reasoning for solution. Poor calibration occurs when the model overconfidently chooses short mode for difficult problems or unnecessarily defaults to think mode for simple problems it could solve directly.

The decision calibration objective $S_{\text{calibration}}(x)$ ensures that mode selection decisions align with the model's actual capability on the specific input by measuring decision calibration quality:

$$S_{\text{calibration}}(x) = 1 - \mathbb{E}_{(c,a)\sim\pi_\theta(c,a|x)}[|P_{\text{model}}(\text{correct}|x,c) - \mathbb{I}(\phi(a) = y^*)|]. \tag{8}$$

To compute the model's confidence estimate, we first extract the raw confidence score from the final answer tokens. Let $L_{\text{answer}}$ denote the logits over the answer vocabulary at the final token position. The raw confidence score is defined as:

$$\text{RawConf}(a) = \max(\text{softmax}(L_{\text{answer}})), \tag{9}$$

which represents the model's highest probability assignment among all possible answer tokens. We then discretize this continuous confidence score into predefined intervals:

$$b = \text{Bin}(\text{RawConf}(a)) = \lfloor \text{RawConf}(a) \times N_{\text{bins}} \rfloor, \tag{10}$$

where $N_{\text{bins}}$ is the number of confidence bins (e.g., 5 or 10).

The model's calibrated confidence estimate is then computed using statistical calibration based on historical performance:

$$P_{\text{model}}(\text{correct}|x,c) = \text{HistoricalAccuracy}(c,b), \tag{11}$$

where $\text{HistoricalAccuracy}(c,b)$ returns the empirical accuracy for mode $c$ in confidence bin $b$:

$$\text{HistoricalAccuracy}(c,b) = \frac{\sum_{t\in\mathcal{H}(c,b)} \mathbb{I}(\text{correct}_t)}{|\mathcal{H}(c,b)|}, \tag{12}$$

where $\mathcal{H}(c,b)$ represents the set of historical samples with mode $c$ and confidence bin $b$, and $\mathbb{I}(\text{correct}_t)$ indicates whether sample $t$ produced the correct answer.

The historical statistics are maintained using exponential decay to prioritize recent performance:

$$\text{HistoricalAccuracy}_{t+1}(c,b) = \lambda \cdot \text{HistoricalAccuracy}_t(c,b) + (1-\lambda) \cdot \mathbb{I}(\text{correct}_{t+1}), \tag{13}$$

where $\lambda \in (0,1)$ is the decay factor. This approach leverages the model's intrinsic confidence distribution while correcting for systematic overconfidence or underconfidence patterns through empirical calibration, requiring no additional neural components while providing more reliable confidence estimates than raw token probabilities. More details can be found in Appendix A.5.

### 4.3 PARETO FRONTIER

The Pareto frontier mechanism provides the mathematical foundation for dynamic weight adaptation in MAGO to address *Challenge #2* mentioned in previous sections. We formalize the multi-objective optimization problem as maintaining an evolving set of weight configurations $\mathcal{F}_t = \{\boldsymbol{\beta}^{(1)}, \boldsymbol{\beta}^{(2)}, ..., \boldsymbol{\beta}^{(k)}\}$, where each $\boldsymbol{\beta}^{(i)} = [\beta_1^{(i)}, \beta_2^{(i)}, \beta_3^{(i)}]$ represents a distinct combination of weights for the three competing objectives (accuracy, efficiency, and calibration). By maintaining a diverse set of non-dominated weight combinations, the Pareto optimization framework avoids the cone entrapment problem that constrains fixed-weight approaches to narrow regions of the objective space, enabling principled adaptation to varying task requirements.

At each iteration $t$, we evaluate the performance of weight configurations using the training batch $\mathcal{B}_t$. For a given weight vector $\boldsymbol{\beta}^{(i)}$, we define the objective vector based on batch-level performance:

$$\mathbf{S}_t(\boldsymbol{\beta}^{(i)}) = \left[\frac{1}{|\mathcal{B}_t|} \sum_{x \in \mathcal{B}_t} S_{\text{accuracy}}(x), \frac{1}{|\mathcal{B}_t|} \sum_{x \in \mathcal{B}_t} S_{\text{efficiency}}(x), \frac{1}{|\mathcal{B}_t|} \sum_{x \in \mathcal{B}_t} S_{\text{calibration}}(x)\right]_{\boldsymbol{\beta}^{(i)}}, \quad (14)$$

where $|\mathcal{B}_t|$ denotes the batch size, each component represents the average performance of the corresponding objective over the current batch, evaluated under the policy $\pi_\theta$ trained with weight configuration $\boldsymbol{\beta}^{(i)}$.

The Pareto frontier is maintained as the set of non-dominated weight configurations:

$$\mathcal{F}_t = \{\boldsymbol{\beta}^{(i)} \mid \nexists \boldsymbol{\beta}^{(j)} \in \mathcal{S}_t : \mathbf{S}_t(\boldsymbol{\beta}^{(j)}) \succ \mathbf{S}_t(\boldsymbol{\beta}^{(i)})\}, \quad (15)$$

where $\mathcal{S}_t$ represents the set of all evaluated weight configurations up to iteration $t$, superscripts $(i)$ and $(j)$ index different weight vectors in the frontier, and $\succ$ denotes Pareto dominance relation.

To address objective correlations that lead to cone entrapment, we introduce a correlation-aware weight selection mechanism. For each training batch $\mathcal{B}_t$ at iteration $t$, we compute the empirical correlation matrix between the three objectives:

$$\mathbf{C}_t[i, j] = \frac{\sum_{x \in \mathcal{B}_t} (S^{(i)}(x) - \mu_t^{(i)})(S^{(j)}(x) - \mu_t^{(j)})}{\sqrt{\sum_{x \in \mathcal{B}_t} (S^{(i)}(x) - \mu_t^{(i)})^2 \sum_{x \in \mathcal{B}_t} (S^{(j)}(x) - \mu_t^{(j)})^2}}, \quad (16)$$

where $S^{(i)}(x)$ denotes the $i$-th objective function (accuracy, efficiency, or calibration) evaluated on input $x$, and $\mu_t^{(i)} = \frac{1}{|\mathcal{B}_t|} \sum_{x \in \mathcal{B}_t} S^{(i)}(x)$ represents the batch mean of objective $i$. This correlation structure guides the selection of weight combinations from the current frontier, ensuring that highly correlated objectives receive balanced attention while conflicting objectives maintain proper balance.

The weight selection process employs a correlation-adaptive scoring function $\Psi_t(\boldsymbol{\beta})$ that evaluates the quality of each weight configuration in the current Pareto frontier and penalizes configurations leading to high correlation between conflicting objectives:

$$\Psi_t(\boldsymbol{\beta}) = \sum_{i=1}^{3} \beta_i \hat{S}_t^{(i)} - \beta_{\text{corr}} \sum_{i<j} |\mathbf{C}_t[i, j]| \cdot |\beta_i - \beta_j|, \quad (17)$$

where $\hat{S}_t^{(i)}$ represents the moving average of the $i$-th objective performance over recent iterations, and $\beta_{\text{corr}} > 0$ is a hyperparameter controlling the penalty strength for correlated objectives. The first term rewards weight configurations that emphasize well-performing objectives, while the second term $|\beta_i - \beta_j|$ penalizes unbalanced weight allocations when objectives $i$ and $j$ are highly correlated, encouraging more uniform distribution across correlated objectives. The optimal weight vector for the current iteration is selected as:

$$\boldsymbol{\beta}_t^* = \arg \max_{\boldsymbol{\beta} \in \mathcal{F}_t} \Psi_t(\boldsymbol{\beta}). \quad (18)$$

To prevent premature convergence and ensure frontier diversity, we employ an exploration mechanism that generates new candidate solutions through guided perturbation:

$$\boldsymbol{\beta}^{\text{new}} = \boldsymbol{\beta}_t^* + \epsilon_t \cdot \mathbf{d}, \quad (19)$$

where $\boldsymbol{\beta}_t^*$ is the currently selected optimal weight vector, $\mathbf{d}$ is sampled uniformly from the constraint surface $\{\mathbf{d} \in \mathbb{R}^3 : \|\mathbf{d}\|_2 = 1, \sum_{i=1}^{3} d_i = 0\}$ to preserve weight normalization, and $\epsilon_t$ is scaled based on the current frontier diversity measure:

$$\epsilon_t = \epsilon_0 \cdot \exp\left(-\frac{D(\mathcal{F}_t)}{D_{\text{target}}}\right), \quad (20)$$

where $\epsilon_0 > 0$ is the base exploration rate hyperparameter, $D_{\text{target}} > 0$ is the target diversity threshold hyperparameter, and $D(\mathcal{F}_t) = \frac{1}{|\mathcal{F}_t|^2} \sum_{\boldsymbol{\beta}^{(i)}, \boldsymbol{\beta}^{(j)} \in \mathcal{F}_t} \|\boldsymbol{\beta}^{(i)} - \boldsymbol{\beta}^{(j)}\|_2$ measures the average pairwise Euclidean distance among frontier solutions (Deb et al., 2002).

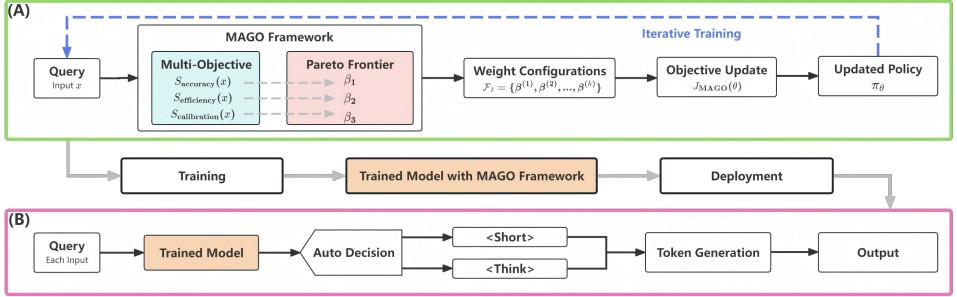

Figure 3: MAGO framework efficiently implements end-to-end integration from training to deployment and inference. (A) Training pipeline with multi-objective optimization and frontier updates. (B) Inference pipeline with learned adaptive mode selection.

The frontier update mechanism integrates newly evaluated candidate solutions and maintains non-dominance:

$$\mathcal{F}_{t+1} = \text{NonDominated}(\mathcal{F}_t \cup \{\boldsymbol{\beta}^{\text{new}}\}) \cap \text{DiversityFilter}(\cdot, \tau_{\text{div}}), \tag{21}$$

where $\text{DiversityFilter}(\cdot, \tau_{\text{div}})$ ensures minimum pairwise distance $\tau_{\text{div}}$ between frontier solutions to prevent clustering. In implementation, the number of frontier vectors $|\mathcal{F}_t|$ grows gradually in early training and stabilizes around 20–25, remaining below the upper bound $|\mathcal{F}_{\max}| = 30$. When the limit is reached, dominated or redundant vectors are pruned through cosine-similarity filtering to preserve representative diversity. More details about algorithm convergence can be found in Appendix A.6.

## 4.4 END-TO-END INTEGRATION

**Training and Deployment.** MAGO integrates into the hybrid reasoning training framework by replacing static weight parameters with dynamic multi-objective optimization (Figure 3A). At each training iteration, the system selects an optimal weight vector $\boldsymbol{\beta}_t^* = [\beta_1^*, \beta_2^*, \beta_3^*]$ from the current Pareto frontier $\mathcal{F}_t$ using correlation-aware selection (Eq. 18). The selected weights instantiate the adaptive weighting function $m_{\text{MAGO}}(x; \boldsymbol{\beta}_t^*) = \beta_1^* S_{\text{accuracy}}(x) + \beta_2^* S_{\text{efficiency}}(x) + \beta_3^* S_{\text{calibration}}(x)$, which determines the control token weight for the current batch in the training objective:

$$J(\theta) = \mathbb{E}_{x, a_i} \left[ \frac{1}{G} \sum_{k=1}^{G} \left( m_{\text{MAGO}}(x; \boldsymbol{\beta}_t^*) L_{k,0}(\theta) + \frac{1}{T_k} \sum_{j=1}^{T_k} L_{k,j}(\theta) - \beta D_{\text{KL}}[\pi_\theta(\cdot|x) \| \pi_{\text{ref}}(\cdot|x)] \right) \right]. \tag{22}$$

In training (Figure 3A), the framework iteratively performs policy updates using the selected weights, evaluates objective performance on the current batch, and maintains the Pareto frontier through non-dominated sorting in this closed-loop process. During deployment (Figure 3B), the trained model automatically selects between <short> and <think> modes with zero inference overhead through learned decision-making, followed by standard token generation. More details can be found in Appendix A.7.

**Training Process and Reward Design.** We use a minimal reward function in training which encourages efficiency while maintaining correctness:

$$r(a, y^*, c) = \begin{cases} 1.0, & \text{if } c = \text{<short>} \text{ and } \phi(a) = y^*, \\ 1.0 - \gamma, & \text{if } c = \text{<think>} \text{ and } \phi(a) = y^*, \\ -1.0, & \text{if } \phi(a) \neq y^*, \end{cases} \tag{23}$$

where $\phi(a)$ extracts the final answer and $0 < \gamma < 1$ creates preference for efficient correct responses (Fang et al., 2025). Additional details including the relative advantage computation and token-level loss formulations are provided in Appendix A.8.

**Framework Summary.** While MAGO framework introduces training overhead, this cost is amortized across millions of inference queries, yielding substantial operational savings. The framework

enables automatic adaptation to changing model capabilities and data characteristics, providing principled trade-offs between accuracy, efficiency, and calibration without manual tuning, operating entirely during training with zero additional inference parameters or computation.

# 5 EXPERIMENTS

## 5.1 IMPLEMENTATION DETAILS

**Experimental Setup.** We employ DeepSeek-R1-Distill-Qwen-1.5B (Guo et al., 2025) as the base model for hybrid reasoning training. To construct paired long-short response data for warm-up distillation, we leverage DeepSeek-R1-671B (Guo et al., 2025) to generate extended reasoning chains and Qwen2.5-Math-1.5B-Instruct (Yang et al., 2024a) for concise responses. The training corpus comprises approximately 40K samples aggregated from DeepScaleR (Luo et al., 2025), OpenR1 (Face, 2025), OpenThoughts-114K (Team, 2025), and additional open-source mathematical reasoning corpora (Jebali et al., 2024; Langlais et al., 2025). To demonstrate scalability, we conduct experiments across Qwen2.5 series backbones of varying sizes (1.5B, 7B, 14B, 32B parameters) (Yang et al., 2024a). All experiments are conducted on 4 to 8 NVIDIA H100 GPUs depending on model size.

**Training Configuration.** Training involves supervised fine-tuning (1 epoch) followed by MAGO reinforcement learning (600 steps), implemented using VeRL (Jiang et al., 2025) and Megatron (Shoeybi et al., 2019). We optimize using AdamW (AbuKaraki et al., 2024) with learning rate $1 \times 10^{-6}$, batch size 128, weight decay 0.01, and momentum $\beta = (0.9, 0.999)$. Context length is 16K during warm-up and 24K during reinforcement learning. MAGO hyperparameters: correlation penalty $\beta_{\text{corr}} = 0.1$, exploration rate $\epsilon_0 = 0.05$, diversity threshold $\tau_{\text{div}} = 0.2$, maximum frontier size $|\mathcal{F}_{\text{max}}| = 30$, calibration bins $N_{\text{bins}} = 10$, decay factor $\lambda = 0.95$ for historical accuracy, and reward preference $\gamma = 0.1$ favoring correct short responses.

**Evaluation Benchmarks and Baselines.** We evaluate on six benchmarks: AIME 2024 (Ji et al., 2025b), Minerva Algebra (Hendrycks et al., 2021), MATH-500 (Lightman et al., 2023), and GSM-8K (Cobbe et al., 2021) for mathematical reasoning, CommonsenseQA (Talmor et al., 2019) and MedQA-USMLE (Jin et al., 2021) for cross-domain generalization. All benchmarks report Pass@1 accuracy and token usage per query. We compare against three baseline categories: (1) *Base LLMs*: DeepSeek-R1-1.5B (Guo et al., 2025), Qwen2.5-1.5B-Instruct, and Qwen2.5-Math-1.5B-Instruct (Yang et al., 2024a); (2) *Shortened CoT*: Model Merging (Team et al., 2025) with coefficients (0.5, 0.6, 0.7) and CoT-Valve (Ma et al., 2025) with $\alpha \in \{4, 6, 8\}$; (3) *Hybrid Reasoning*: DeGRPO (Fang et al., 2025) with fixed $\alpha = 0.001$, random router, and Qwen2.5-7B router (Ong et al., 2024). Additional details are provided in Appendix A.9.

## 5.2 RESULT

**Multi-Objective Optimization Evaluation.** We first evaluate MAGO on the 1.5B backbone. Across mathematical reasoning benchmarks, MAGO yields $2.2\times$ to $3\times$ token-efficiency gains and $0.6\%$ to $9.4\%$ relative accuracy improvements over heuristic baselines, with consistent improvements across all evaluated tasks. Table 1 shows that MAGO achieves superior token efficiency (7,164 vs. 18,063+ baseline tokens on AIME) and competitive or superior accuracy on most benchmarks, including AIME (0.2741) and MATH-500 (0.8247), while remaining close to the best scores on Minerva Algebra (0.9483 vs. 0.9577) and GSM-8K (0.8469 vs. 0.8572). Unlike baseline methods that require dataset-specific hyperparameter tuning and router-based approaches that struggle with complex datasets, MAGO's Pareto optimization framework automatically calibrates reasoning strategies to achieve optimal efficiency–accuracy trade-offs, demonstrating the effectiveness of principled multi-objective optimization over heuristic approaches in hybrid reasoning systems. To validate scalability, we further apply MAGO to larger backbones (7B, 14B, and 32B). As model capacity increases, Pass@1 improves consistently across all benchmarks while average token usage per query decreases slightly, indicating that MAGO's Pareto optimization generalizes effectively to larger-scale models without increasing inference cost. More details can be found in Appendix A.10.

**Mode Collapse in RL.** Our Pareto optimization prevents mode collapse by maintaining balanced reasoning mode selection throughout training, avoiding the extreme preference for short responses that characterizes vanilla GRPO. Figure 4 (A) illustrates the *Mode Collapse* issue in standard GRPO,

Table 1: Comparison of MAGO against baseline reasoning methods on mathematical benchmarks.

| Models | Type | AIME 2024 | | Minerva Algebra | | MATH-500 | | GSM8K | |
|---|---|---|---|---|---|---|---|---|---|
| | | Pass@1 | #Tokens | Pass@1 | #Tokens | Pass@1 | #Tokens | Pass@1 | #Tokens |
| DeepSeek-R1-1.5B (Guo et al., 2025) | Base LLM | 0.2800 | 18063 | 0.9577 | 3029 | 0.8608 | 5675 | 0.8347 | 1919 |
| Q-1.5B (Yang et al., 2024a) | | 0.0200 | 1300 | 0.7771 | 933 | 0.5168 | 855 | 0.7022 | 466 |
| QMath-1.5B (Yang et al., 2024a) | | 0.1133 | 1128 | 0.9184 | 586 | 0.7604 | 721 | 0.8572 | 447 |
| Merging-0.5 (Team et al., 2025) | Short CoT | 0.1333 | 8636 | 0.9292 | 834 | 0.7740 | 1524 | 0.8332 | 601 |
| Merging-0.6 (Team et al., 2025) | | 0.1733 | 10615 | 0.9321 | 1091 | 0.7900 | 3000 | 0.8381 | 747 |
| Merging-0.7 (Team et al., 2025) | | 0.1667 | 15854 | 0.9398 | 1834 | 0.8108 | 4347 | 0.8458 | 1201 |
| CoT-Valve $\alpha = 8$ (Ma et al., 2025) | | 0.2000 | 10692 | 0.8079 | 1903 | 0.7060 | 3723 | 0.7726 | 773 |
| CoT-Valve $\alpha = 6$ (Ma et al., 2025) | | 0.1933 | 17245 | 0.9468 | 2656 | 0.8024 | 5167 | 0.7970 | 1009 |
| CoT-Valve $\alpha = 4$ (Ma et al., 2025) | | 0.2267 | 17722 | 0.9439 | 2965 | 0.8036 | 5820 | 0.8108 | 1396 |
| Router Random (Fang et al., 2025) | Hybrid | 0.1300 | 8093 | 0.9032 | 1736 | 0.7484 | 3096 | 0.8205 | 1086 |
| Router Q-7B (Ong et al., 2024) | | 0.1480 | 9296 | 0.9049 | 795 | 0.7781 | 2748 | 0.8587 | 563 |
| DeGRPO-Qwen-1.5B (Fang et al., 2025) | Hybrid | 0.2506 | 7262 | 0.9216 | 1228 | 0.8037 | 2644 | 0.8418 | 649 |
| MAGO-Qwen-1.5B (Ours) | Pareto | 0.2741 | 7164 | 0.9483 | 1174 | 0.8247 | 2578 | 0.8469 | 633 |
| MAGO-Qwen-7B (Ours) | Pareto | 0.2960 | 6890 | 0.9562 | 1102 | 0.8424 | 2426 | 0.8611 | 592 |
| MAGO-Qwen-14B (Ours) | | 0.3112 | 6724 | 0.9621 | 1041 | 0.8538 | 2368 | 0.8723 | 571 |
| MAGO-Qwen-32B (Ours) | | 0.3254 | 6587 | 0.9689 | 992 | 0.8652 | 2294 | 0.8834 | 552 |

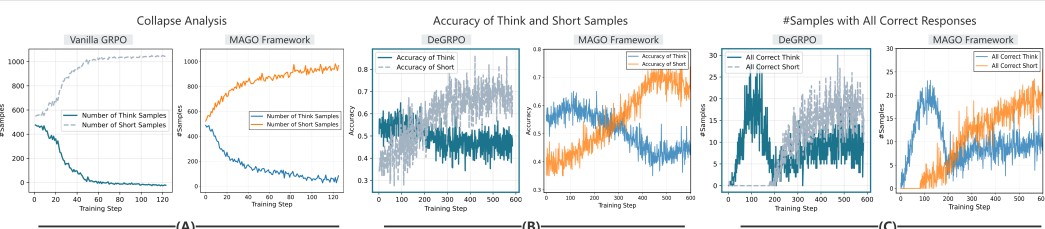

Figure 4: Training dynamics comparison between vanilla GRPO, DeGRPO, and MAGO frameworks. (A) Mode collapse analysis showing sample distribution over training steps. (B) Accuracy evolution for think and short reasoning modes during training. (C) Number of samples achieving correct responses for both reasoning modes.

Table 2: Cross-domain generalization on CommonsenseQA.

| Model | Accuracy (%) ↑ | Tokens / Query ↓ | Token Reduction |
|---|---|---|---|
| DeGRPO | 73.1 | 312 | - |
| CoT-Valve | 73.8 | 298 | 1.05 |
| **MAGO (ours)** | **74.9** | **152** | **2.05** |

where the model develops an excessive preference for short outputs during training, with the number of think samples dropping precipitously to near zero within 120 training steps. In contrast, the proposed framework demonstrates significantly more stable training dynamics, maintaining a balanced distribution between think and short samples throughout the process. The vanilla GRPO's rapid collapse to predominantly short-mode usage (below 10 think samples) indicates a failure to properly balance competing objectives of accuracy and efficiency. Our Pareto-based optimization prevents this catastrophic collapse by maintaining diverse weight configurations that ensure neither reasoning mode is abandoned, enabling adaptive strategy selection based on query complexity rather than converging to suboptimal modes.

**The U-Shape Learning Curve.** Figure 4 (B) reveals that our approach achieves more balanced training dynamics across 600 steps, with both reasoning modes converging smoothly after approximately 300 steps. While DeGRPO exhibits high volatility with fluctuations in accuracy for both modes, the proposed method demonstrates stable convergence patterns with reduced variance. The think mode maintains consistent performance around 0.6-0.7 accuracy while the short mode gradually improves from 0.4 to 0.5, contrasting sharply with DeGRPO's chaotic dynamics where accuracy fluctuates wildly between 0.3 and 0.8. Figure 4 (C) demonstrates superior sample efficiency, showing that the model quickly learns to activate short mode while ensuring correctness. The intersection point between think and short correct responses occurs later in training (around step 400), indicating more thorough exploration of reasoning mode trade-offs before settling on optimal strategies.

**Cross-Domain Generalization.** To evaluate the generalization ability of MAGO beyond mathematical reasoning, we perform additional experiments on CommonsenseQA, a benchmark that assesses everyday reasoning and contextual understanding. The objective is to examine whether

our proposed Pareto-based adaptive optimization, trained only on mathematical reasoning data, can effectively transfer to a different reasoning domain without further fine-tuning. The same inference settings described in Section 5.1 are adopted for all methods. Representative hybrid reasoning baselines, including DeGRPO and CoT-Valve, are used for comparison. The experimental results are presented in Table 2. All results are averaged over three random seeds to ensure stability. MAGO achieves 74.9% accuracy, outperforming DeGRPO and CoT-Valve by 1.8% and 1.1%, respectively, while reducing the average number of generated tokens from 312 to 152, corresponding to a $2.05\times$ improvement in efficiency. These findings demonstrate that MAGO's Pareto-based adaptive optimization generalizes effectively across reasoning domains and maintains a stable balance between accuracy and computational efficiency. We also evaluate MAGO on MedQA-USMLE (Jin et al., 2021), a medical question answering benchmark, where MAGO achieves over $2.0\times$ efficiency improvement while maintaining competitive accuracy. More details can be found in Appendix A.18.

**Computational Complexity.** We analyze the computational and memory complexity introduced by the multi-objective optimization process. Let $|\mathcal{B}|$ denote the batch size, $M = 3$ the number of objectives, and $|\mathcal{F}_t|$ the number of maintained frontier vectors, with an upper bound $|\mathcal{F}_{\max}| = 30$. For space consideration, more details are provided in Appendix A.19.

## 6 RELATED WORK

**Reasoning (Hybrid and Efficient).** Recent hybrid reasoning advances combine multiple paradigms for efficiency. Chain-of-thought and program-aided reasoning integrate natural language with code (Gao et al., 2022; Ranaldi et al., 2024), while self-refinement methods iteratively improve chains (Madaan et al., 2023; Ji et al., 2025a). Tree-of-thoughts structures reasoning as search (Yao et al., 2023; Pandey et al., 2025), adaptive frameworks select strategies by complexity (Zhou et al., 2023; Tu et al., 2025), and multi-path reasoning aggregates diverse chains (Zhu et al., 2024; Zhang et al., 2024c). Compression (Omri et al., 2025; Han et al., 2024) and selective generation (Jo et al., 2022; Yang et al., 2024b) reduce tokens while maintaining accuracy. However, these lack principled frameworks for jointly optimizing strategy selection and efficiency across diverse distributions.

**Effective Reasoning (Single Methods).** Single-paradigm optimizations enhance reasoning without hybridization. Prompt compression preserves semantics with 20x ratios (Jiang et al., 2023), knowledge distillation transfers capabilities to smaller models (Shridhar et al., 2023), and speculative decoding accelerates inference (Leviathan et al., 2022). Structured pruning removes redundant steps (Tao et al., 2023; Men et al., 2024), early-exit uses confidence thresholds (Tang et al., 2023; Xu et al., 2025), token-level optimization skips steps (Lee et al., 2024), and cache-based approaches reuse patterns (Yang et al., 2025a;b). These optimize singular objectives, missing opportunities.

**Multi-Objective Optimization (MOO).** MOO in language models balances competing goals. Pareto-optimal solutions identify accuracy-efficiency trade-offs (Mukherjee et al., 2024; Huang et al., 2024), weighted scalarization combines objectives (Yang et al., 2024c; Li & Ma, 2018), and RL optimizes multiple rewards (Zhang et al., 2024b). Constraint-based methods ensure safety (Zhang et al., 2024a; Peng et al., 2025), dynamic adjustment adapts priorities (Low & Kumar, 2024; Krishna & Vali, 2025), preference learning captures values (Dai et al., 2023; Shen et al., 2025), and evolutionary algorithms handle trade-offs (Bai et al., 2023; Li et al., 2024). However, MOO in inference mode selection remains underexplored, missing context-aware optimization opportunities.

## 7 CONCLUSION

We present MAGO, a multi-objective adaptive generation optimization framework that integrates Pareto frontier maintenance with correlation-aware weight selection for hybrid reasoning in LLMs. Our framework combines three competing objectives (accuracy, efficiency, and calibration) through dynamic weight adaptation using Pareto frontier maintenance and correlation-aware selection. This principled approach eliminates hyperparameter tuning while preventing the mode collapse observed in existing reinforcement learning methods. Experiments show that MAGO delivers $2.2\times$ to $3\times$ token-efficiency gains along with $0.6\%$ to $9.4\%$ relative accuracy improvements over heuristic methods on mathematical reasoning tasks. Cross-domain evaluation on CommonsenseQA and MedQA further confirms the framework's transferability beyond mathematics without additional fine-tuning.

## ETHICS STATEMENT

This work adheres to the ICLR Code of Ethics. Our research focuses on developing multi-objective optimization for adaptive reasoning in large language models. We identify the following ethical considerations:

**Privacy.** No personally identifiable information is collected or processed.

**Environmental Impact.** We report detailed computational requirements in Appendix A.9.

**Potential Harms.** Our optimization framework could potentially be applied to harmful applications. We emphasize the importance of responsible deployment and adherence to AI safety guidelines.

## REPRODUCIBILITY STATEMENT

To facilitate reproduction of our results:

**Code.** Complete implementation including training scripts and evaluation code will be released upon paper acceptance. For review purposes, we provide pseudocode in Appendix.

**Experimental Details.** Hyperparameters and experimental setup are fully specified in Appendix A.9. Hardware specifications are provided in Appendix A.9.

**Data.** We use publicly available datasets: AIME 2024, Minerva Algebra, MATH-500, and GSM-8K for mathematical reasoning evaluation; CommonsenseQA and MedQA-USMLE for cross-domain evaluation; DeepScaleR, OpenR1, and OpenThoughts-114K for training.

## ACKNOWLEDGEMENT

This research was funded by Zhejiang Province Philosophy and Social Sciences Planning Project (No.25GXSZ045YB) and Zhejiang Province Sino-Foreign Cooperative Education Research Center, Zhejiang Provincial Education Science Planning Project (No.2025SCG214).

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

## A    APPENDIX

All appendices are provided in the supplementary text.

