# MAGO: Beyond Fixed Hyperparameters with Multi-Objective Pareto Optimization for Hybrid LLM Reasoning

**Hongcheng Ding**[*†]
Dongfang College,
Zhejiang University of
Finance and Economics

**Xuanze Zhao**[*]
Dongfang College,
Zhejiang University of
Finance and Economics

**Ruiting Deng**
Yunnan University of
Finance and Economics

**Shamsul Nahar Abdullah**
INTI International University

**Deshinta Arrova Dewi**
INTI International University

**Qingyu Liu**
SEGi University

## The Use of Large Language Models (LLMs)

We use Claude 4 Sonnet and ChatGPT 5 for grammar checking, spelling correction, and translation assistance in both the main text and appendix of this paper.

## Related Work

**Reasoning (Hybrid and Efficient).** Recent advances in reasoning models explore hybrid approaches that combine multiple reasoning paradigms for enhanced efficiency. Chain-of-thought (CoT) and program-aided reasoning integrate natural language with executable code (Gao et al., 2022; Ranaldi et al., 2024), achieving superior performance on mathematical tasks. Self-refinement methods iteratively improve reasoning chains through verification loops (Madaan et al., 2023; Ji et al., 2025), while tree-of-thoughts structures reasoning as search problems (Yao et al., 2023; Pandey et al., 2025). Adaptive reasoning frameworks dynamically select between reasoning strategies based on task complexity (Zhou et al., 2025; Tu et al., 2025), reducing unnecessary computation. Multi-path reasoning aggregates diverse reasoning chains for robustness (Zhu et al., 2024; Zhang et al., 2024c), though computational costs scale linearly with paths. Recent work on reasoning compression (Omri et al., 2025; Han et al., 2024) and selective generation (Jo et al., 2022; Yang et al., 2024b) demonstrates that hybrid approaches can maintain accuracy while significantly reducing token generation. However, these methods often lack principled frameworks for jointly optimizing strategy selection and computational efficiency across diverse task distributions.

**Effective Reasoning (Non-hybrid and Efficient Single Methods).** Single-paradigm reasoning optimizations focus on enhancing specific reasoning types without hybridization. Prompt compression techniques reduce context length while preserving semantic information (Jiang et al., 2023), achieving up to 20x compression ratios. Knowledge distillation transfers reasoning capabilities to smaller models (Shridhar et al., 2023), though often sacrificing reasoning depth. Speculative decoding accelerates inference through parallel token prediction (Leviathan et al., 2022), while maintaining output quality. Structured pruning removes redundant reasoning steps post-generation (Tao et al., 2023; Men et al., 2024). Early-exit mechanisms terminate reasoning when confidence thresholds are met (Tang et al., 2023; Xu et al., 2025), reducing average latency. Token-level optimization methods skip unnecessary intermediate steps (Lee et al., 2024). Cache-based approaches store and reuse common reasoning patterns (Yang et al., 2025a;b). Despite individual effectiveness, these methods optimize singular objectives (speed or accuracy), missing opportunities for multi-dimensional optimization that considers interpretability, resource constraints, and task-specific requirements simultaneously.

---

[*]Equal contributions
[†]Corresponding Author: `i24025877@student.newinti.edu.my` (INTI International University)

**Multi-Objective Optimization.** Multi-objective optimization (MOO) in language models balances competing goals through sophisticated frameworks. Pareto-optimal solutions identify non-dominated trade-offs between accuracy and efficiency (Mukherjee et al., 2024; Huang et al., 2024), while weighted scalarization combines objectives with learnable weights (Yang et al., 2024c; Li & Ma, 2018). Reinforcement learning approaches optimize for multiple rewards simultaneously (Zhang et al., 2024b), including helpfulness, harmlessness, and efficiency. Constraint-based methods ensure safety boundaries while maximizing performance (Zhang et al., 2024a; Peng et al., 2025). Dynamic weight adjustment adapts optimization priorities based on runtime conditions (Low & Kumar, 2024; Krishna & Vali, 2025). Recent frameworks employ preference learning to capture human values across dimensions (Dai et al., 2023; Shen et al., 2025). Evolutionary algorithms maintain diverse solution populations for complex trade-offs (Bai et al., 2023; Li et al., 2024). However, the application of MOO in inference mode selection has not been fully explored. Due to the existing focus on training time optimization rather than inference time adaptation, the opportunity for context-aware multi-objective inference that jointly considers accuracy, computational cost, and interpretability has been missed.

## APPENDIX A.1: STATIC WEIGHT CHALLENGE ANALYSIS

This appendix provides comprehensive empirical analysis demonstrating the fundamental challenges that motivate the MAGO framework. Through systematic experimentation across mathematical reasoning benchmarks, we identify critical limitations in existing static weight approaches that necessitate our multi-objective optimization solution.

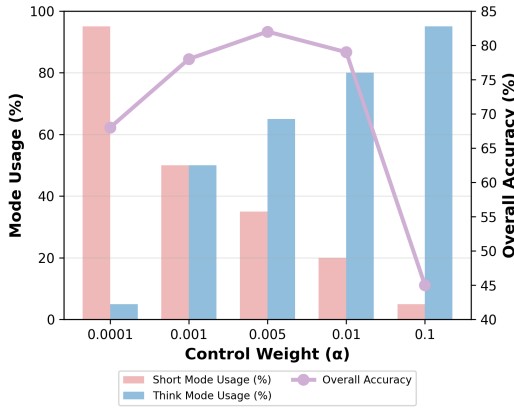

Figure 1: Mode Selection Imbalance Across $\alpha$ Values. The plot demonstrates how different control weight values in DeGRPO lead to severe imbalances in reasoning mode selection. At $\alpha = 0.0001$, over 95% of responses utilize short-form reasoning, sacrificing accuracy on complex problems. Conversely, $\alpha = 0.1$ results in 95% think-mode usage, negating efficiency benefits. The optimal performance occurs around $\alpha = 0.005$ with balanced mode usage, but this narrow optimum illustrates the fragility of static weight approaches.

Figure 1 reveals the severe mode selection imbalance that emerges when using fixed control weights in the DeGRPO framework. The experimental analysis spans control weight values $\alpha \in \{0.0001, 0.001, 0.005, 0.01, 0.1\}$, demonstrating how minor changes in this hyperparameter lead to dramatically different reasoning behaviors. At the extreme low end, $\alpha = 0.0001$ results in overwhelming bias toward short-form reasoning, with over 95% of responses utilizing the computationally efficient short mode. While this configuration minimizes computational costs, it severely compromises the model's ability to handle complex problems requiring multi-step reasoning, as evidenced by the corresponding drop in overall accuracy to approximately 68%. Conversely, at $\alpha = 0.1$, the system exhibits the opposite pathology, with nearly 95% think mode usage, effectively negating the efficiency benefits that hybrid reasoning systems are designed to provide. The accuracy degradation at this extreme setting (dropping to 45%) indicates that excessive emphasis on the control token disrupts the delicate balance required for effective policy learning. The optimal performance occurs around $\alpha = 0.005$, achieving roughly balanced mode usage and peak accuracy of

82%. However, this narrow optimum demonstrates the fragility of static weight approaches and the difficulty of manual hyperparameter tuning, particularly when this optimal value may not generalize across different datasets or problem complexities.

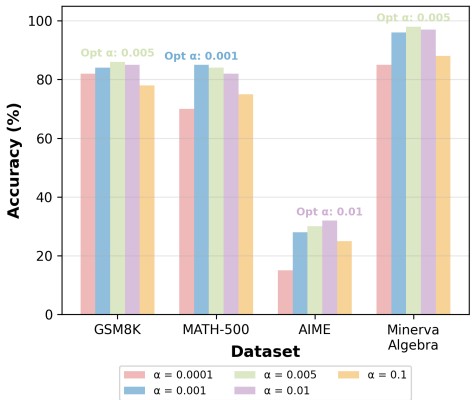

Figure 2: Dataset-Specific $\alpha$ Sensitivity. The analysis across four mathematical reasoning benchmarks reveals that optimal control weights vary dramatically: GSM8K performs best with $\alpha = 0.001$, while AIME requires $\alpha = 0.01$ for peak performance. The annotations show dataset-specific optimal values spanning nearly two orders of magnitude, demonstrating that no single fixed weight achieves consistent performance across diverse problem types.

Figure 2 provides compelling evidence that no single fixed control weight can achieve optimal performance across diverse mathematical reasoning datasets. The analysis evaluates four benchmark datasets with varying complexity characteristics: GSM8K (elementary arithmetic), MATH-500 (competition-level problems), AIME (advanced high school mathematics), and Minerva Algebra (algebraic reasoning tasks). The results reveal striking dataset-specific sensitivity patterns that fundamentally challenge the viability of static weight configurations. For GSM8K, which contains relatively straightforward arithmetic word problems, lower control weights ($\alpha = 0.001$) achieve optimal performance at 85% accuracy, as these problems often benefit from direct computational approaches rather than elaborate reasoning chains. In stark contrast, AIME problems, representing the most challenging mathematical reasoning tasks, require substantially higher control weights ($\alpha = 0.01$) to achieve their peak performance of 32%, reflecting the necessity of extended reasoning for complex multi-step mathematical proofs. MATH-500 and Minerva Algebra exhibit intermediate patterns, with optimal values around $\alpha = 0.005$ and $\alpha = 0.001$ respectively. The magnitude of these performance differences is substantial, with gaps ranging from 3-8 percentage points between optimal and suboptimal configurations. Perhaps most critically, the annotations indicating dataset-specific optimal values span nearly two orders of magnitude, from 0.001 to 0.01, demonstrating that problem complexity characteristics fundamentally alter the optimal balance between mode selection emphasis and response quality optimization.

## APPENDIX A.2: MULTI-OBJECTIVE OPTIMIZATION CHALLENGE ANALYSIS

Figure 3 illustrates the geometric limitations inherent in fixed-weight scalarization approaches through a two-dimensional accuracy-efficiency trade-off analysis. The blue line represents the theoretical Pareto frontier containing all optimal solutions that cannot be improved in one objective without degrading another. However, when using fixed weight combinations, the optimization process becomes constrained to search along predetermined directions in the objective space, effectively limiting exploration to the narrow red-shaded region. This spatial constraint represents a fundamental limitation of scalarization approaches, where the weight vector determines a specific search direction that may miss superior solutions lying in unexplored regions of the objective space. The orange circle indicates the solution achieved by a fixed-weight configuration, which, while reasonable, falls short of the theoretical optimum represented by the blue points on the Pareto frontier. The unexplored solutions, marked by blue circles outside the constrained search region, represent potentially superior trade-offs that remain inaccessible to fixed-weight optimization. This geometric

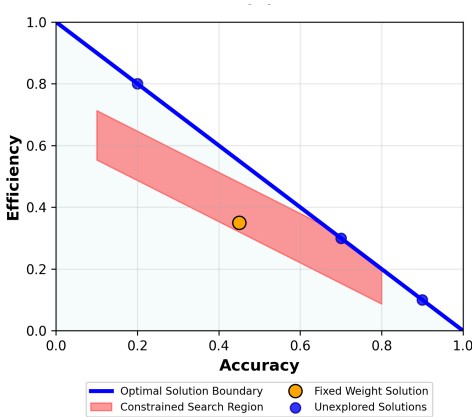

Figure 3: Fixed Weight Search Constraint in 2D Objective Space. The blue line represents the theoretical Pareto frontier of optimal accuracy-efficiency trade-offs. Fixed-weight scalarization constrains optimization to the narrow red-shaded region, missing potentially superior solutions (blue points) that exist outside this constrained search area. The orange point shows a typical fixed-weight solution that falls short of the theoretical optimum.

analysis demonstrates that static approaches inherently restrict the solution space to a fraction of the theoretically achievable outcomes, motivating the need for adaptive optimization strategies that can explore the full trade-off landscape rather than being confined to predetermined search trajectories.

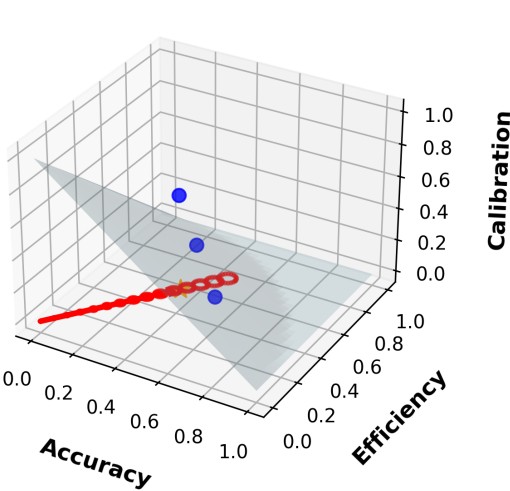

Figure 4: Optimization Space Limitation in 3D. The three-dimensional visualization shows how fixed-weight approaches create cone-shaped constraints (red trajectory) when optimizing accuracy, efficiency, and calibration simultaneously. The blue points represent superior solutions that exist outside this constrained cone but remain inaccessible to traditional scalarization methods, illustrating the cone entrapment problem that becomes increasingly severe as the number of objectives increases.

Figure 4 extends the spatial constraint analysis to the complete three-objective optimization space encompassing accuracy, efficiency, and calibration simultaneously. The three-dimensional visualization reveals how fixed-weight approaches create even more severe geometric constraints when multiple objectives must be balanced concurrently. The red trajectory represents the path followed by static weight optimization, which becomes trapped in a narrow cone-shaped region emanating from a specific direction in the objective space. This cone entrapment phenomenon severely limits the exploration of alternative trade-offs, as the optimization is constrained to move along a pre-

determined direction determined by the fixed weight ratios. The blue points scattered throughout the three-dimensional space represent superior solutions that exist outside this constrained cone but remain inaccessible to traditional scalarization approaches. The shaded regions illustrate the vast unexplored portions of the objective space that could potentially contain optimal solutions for specific deployment scenarios or problem characteristics. This three-dimensional analysis makes clear that as the number of objectives increases, the limitations of fixed-weight approaches become increasingly severe, with the accessible solution space representing an ever-smaller fraction of the theoretical optimum. The cone entrapment problem motivates our Pareto frontier approach, which maintains multiple non-dominated solutions across the entire objective space, enabling comprehensive exploration of trade-offs rather than restricting optimization to narrow geometric constraints imposed by static weight selections.

## APPENDIX A.3: DECOUPLED GROUP RELATIVE POLICY OPTIMIZATION

This appendix provides detailed mathematical foundations and analysis of Decoupled Group Relative Policy Optimization (DeGRPO), which addresses critical limitations in vanilla GRPO when applied to hybrid reasoning tasks.

### A.3.1 PROBLEM FORMULATION IN VANILLA GRPO

The standard Group Relative Policy Optimization (GRPO) framework (Shao et al., 2024) employs uniform token treatment through a single normalization factor. For a trajectory $a_i = (a_{i,0}, a_{i,1}, \ldots, a_{i,T_i})$ where $a_{i,0}$ is the control token and $(a_{i,1}, \ldots, a_{i,T_i})$ constitute the response, the vanilla GRPO objective is:

$$\mathcal{J}_{\text{GRPO}}(\theta) = \mathbb{E}_{x,a_i} \left[ \frac{1}{G} \sum_{k=1}^{G} \left( \frac{1}{T_k + 1} \sum_{t=0}^{T_k} \mathcal{L}_{k,t}(\theta) - \beta D_{\text{KL}}[\pi_\theta(\cdot|x) \| \pi_{\text{ref}}(\cdot|x)] \right) \right], \quad (1)$$

where $\mathcal{L}_{k,t}(\theta)$ represents the token-level surrogate loss and $G$ denotes the group size for relative advantage computation.

### A.3.2 IDENTIFIED IMBALANCES

The uniform normalization $\frac{1}{T_k+1}$ in vanilla GRPO introduces two critical imbalances when applied to hybrid reasoning:

**Mode-Accuracy Imbalance:** Each trajectory contains exactly one control token ($a_{i,0}$) but $T_i$ response tokens. The control token's gradient contribution becomes:

$$\nabla_\theta \mathcal{L}_{i,0}(\theta) \cdot \frac{1}{T_i + 1} \quad (2)$$

while response tokens collectively contribute:

$$\sum_{j=1}^{T_i} \nabla_\theta \mathcal{L}_{i,j}(\theta) \cdot \frac{1}{T_i + 1}. \quad (3)$$

This creates a $1 : T_i$ imbalance, where the single control token responsible for mode selection receives disproportionately weak gradient signals compared to the $T_i$ response tokens optimizing for accuracy.

**Think-Short Imbalance:** The normalization factor $\frac{1}{T_i+1}$ varies significantly between reasoning modes. For long-form reasoning with $T_i^{\text{think}} \gg T_i^{\text{short}}$, the control token gradient becomes:

$$\frac{\nabla_\theta \mathcal{L}_{i,0}^{\text{think}}(\theta)}{T_i^{\text{think}} + 1} \ll \frac{\nabla_\theta \mathcal{L}_{i,0}^{\text{short}}(\theta)}{T_i^{\text{short}} + 1}. \quad (4)$$

This systematic bias causes the `<think>` control token to receive substantially weaker optimization signals than `<short>`, leading to mode collapse where the model defaults to short-form reasoning.

### A.3.3 DeGRPO Solution

DeGRPO addresses these imbalances through decoupled normalization and explicit weighting. The reformulated objective separates control and response token optimization:

$$\mathcal{J}_{\text{DeGRPO}}(\theta) = \mathbb{E}_{x, a_i} \left[ \frac{1}{G} \sum_{k=1}^{G} \left( \alpha \mathcal{L}_{k,0}(\theta) + \frac{1}{T_k} \sum_{j=1}^{T_k} \mathcal{L}_{k,j}(\theta) - \beta D_{\text{KL}}[\pi_\theta(\cdot|x) \| \pi_{\text{ref}}(\cdot|x)] \right) \right]. \quad (5)$$

The key modifications include:

**Independent Normalization:** Control tokens receive weight $\alpha$ independent of sequence length, while response tokens are normalized by $\frac{1}{T_k}$ excluding the control token.

**Adaptive Weighting:** The hyperparameter $\alpha$ provides explicit control over the mode selection learning rate relative to response generation. Empirically, $\alpha = 0.001$ provides stable training dynamics (Fang et al., 2025).

### A.3.4 Training Dynamics Analysis

DeGRPO exhibits characteristic U-shaped training dynamics where the proportion of long-form reasoning initially increases, then gradually decreases. This behavior emerges from two complementary learning processes:

**Early Phase (Accuracy-Driven):** The model initially favors long-form reasoning due to higher accuracy on complex problems. The control token policy adapts to select `<think>` for difficult queries where extended reasoning provides performance benefits.

**Late Phase (Efficiency-Driven):** As response generation quality improves through reinforcement learning, short-form responses achieve higher success rates on simpler problems. The reward structure encourages mode selection that maximizes efficiency while maintaining accuracy, leading to increased `<short>` usage.

### A.3.5 Convergence Properties

The decoupled formulation ensures that both mode selection and response generation receive consistent gradient signals throughout training. The control token optimization becomes independent of sequence length variations, preventing the systematic bias that causes mode collapse in vanilla GRPO.

The relative advantage computation follows the standard formulation:

$$\hat{A}_{i,t} = r_i - \frac{1}{G} \sum_{j=1}^{G} r_j, \quad (6)$$

where $r_i$ represents the trajectory reward and the group-relative normalization provides stable learning signals across varying problem difficulties (Liu et al., 2025).

### A.3.6 Hyperparameter Sensitivity

The weight parameter $\alpha$ controls the learning speed of mode selection relative to response generation. Higher values of $\alpha$ accelerate policy adaptation but may lead to premature mode assignment before response quality improvements. The empirically determined value $\alpha = 0.001$ balances exploration of both reasoning modes while preventing training instability.

This decoupled approach enables the model to learn contextually appropriate reasoning strategies without manual mode specification, achieving both computational efficiency and maintained performance across diverse problem complexities.

## APPENDIX A.4: PARETO OPTIMIZATION

Multi-objective optimization problems (MOPs) arise when multiple conflicting objectives must be optimized simultaneously. Unlike single-objective optimization where a unique optimal solution typically exists, MOPs generate a set of trade-off solutions known as the Pareto frontier (Miettinen, 1999; Deb et al., 2002). The fundamental challenge in multi-objective optimization lies in the absence of a single solution that simultaneously optimizes all objectives, necessitating the identification of trade-off solutions that represent optimal compromises among competing goals.

### A.4.1 THEORETICAL FOUNDATIONS

The concept of Pareto dominance provides the mathematical foundation for comparing solutions in multi-objective spaces. For two solutions $x_1, x_2 \in \mathcal{X}$, we say $x_1$ dominates $x_2$ (denoted $x_1 \preceq x_2$) if and only if $f_i(x_1) \leq f_i(x_2)$ for all $i \in \{1, 2, \ldots, m\}$ and there exists at least one $j \in \{1, 2, \ldots, m\}$ such that $f_j(x_1) < f_j(x_2)$. This dominance relation establishes a partial ordering over the solution space, enabling the identification of superior solutions without requiring explicit preference information.

A solution $x^* \in \mathcal{X}$ is Pareto-optimal if there exists no other solution $x \in \mathcal{X}$ such that $x \preceq x^*$. The collection of all Pareto-optimal solutions forms the Pareto set $\mathcal{P}$, and their corresponding objective vectors constitute the Pareto front $\mathcal{F} = \{\mathbf{f}(x) : x \in \mathcal{P}\}$. This mathematical framework ensures that every solution on the Pareto front represents an optimal trade-off where improving any objective necessarily degrades at least one other objective.

### A.4.2 ALGORITHMIC APPROACHES

The Non-dominated Sorting Genetic Algorithm II (NSGA-II), introduced by Deb et al. (2002), represents one of the most influential algorithms in multi-objective optimization. NSGA-II addresses key limitations of earlier approaches through three main innovations that significantly improve computational efficiency and solution quality. The algorithm implements fast non-dominated sorting that reduces computational complexity from $O(MN^3)$ to $O(MN^2)$ where $M$ is the number of objectives and $N$ is the population size. Additionally, NSGA-II incorporates an elitist strategy that ensures good solutions found in previous generations are not lost during evolution, while the crowding distance mechanism maintains diversity by estimating the density of solutions in the objective space and promoting uniform distribution along the Pareto front.

Recent developments in Pareto optimization have introduced several promising directions that extend traditional approaches. Pareto Set Learning (PSL) represents an emerging paradigm where neural networks are trained to learn mappings from preference vectors to Pareto optimal solutions (Li et al., 2024). This approach enables more efficient exploration of the Pareto front by directly generating solutions corresponding to specific preference weightings. Surrogate-assisted methods address expensive optimization problems by using regression models to approximate objective functions, significantly reducing computational cost while maintaining solution quality (Albeaik et al., 2024).

### A.4.3 LIMITATIONS OF SCALARIZATION APPROACHES

Traditional approaches convert multi-objective problems into single-objective ones through weighted scalarization: $\min_{x \in \mathcal{X}} \sum_{i=1}^{m} \lambda_i f_i(x)$, where $\sum_{i=1}^{m} \lambda_i = 1$ and $\lambda_i \geq 0$. However, this approach suffers from fundamental theoretical and practical limitations that restrict its effectiveness in complex optimization scenarios. Weight sensitivity represents a critical issue where small changes in weight values can lead to dramatically different solutions, making the approach unstable and difficult to tune. The method also fails to find solutions in non-convex regions of the Pareto front, limiting exploration to convex portions of the objective space and potentially missing superior solutions.

The requirement for a priori preference specification forces practitioners to specify weight values before understanding the trade-off landscape, often leading to suboptimal choices. Most critically, scalarization approaches impose spatial constraints that restrict optimization to predetermined cone-shaped regions in the objective space, preventing comprehensive exploration of available trade-offs. Recent empirical studies provide compelling evidence for these limitations, demonstrating that

Pareto optimization identifies 100% of optimal solutions in large search spaces exceeding 4 million candidates while scalarization approaches achieve less than 60% coverage (Albeaik et al., 2024).

### A.4.4 Relevance to Hybrid Reasoning Optimization

In the context of MAGO's hybrid reasoning framework, Pareto optimization offers critical advantages over traditional scalarization approaches that directly address the challenges inherent in multi-objective reasoning optimization. The framework provides theoretical guarantees through Pareto optimality, ensuring that solutions are provably optimal in the Pareto sense while maintaining completeness by discovering solutions in non-convex regions missed by scalarization. Formal convergence guarantees under mild regularity conditions provide mathematical assurance of algorithm behavior and performance.

The practical benefits of Pareto optimization align perfectly with MAGO's requirements for adaptive hybrid reasoning. The elimination of a priori weight specification removes the need for extensive manual hyperparameter tuning, while systematic exploration of accuracy-efficiency-calibration trade-offs enables comprehensive understanding of available compromises. The maintenance of diverse solution sets supports different operational requirements, and dynamic weight adaptation based on problem characteristics enables context-aware optimization that responds to changing conditions during training and deployment.

The three objectives in MAGO exhibit characteristics that make them particularly suitable for Pareto optimization. The conflicting nature of accuracy and efficiency, where higher accuracy often requires longer reasoning chains that directly conflict with computational efficiency goals, creates the fundamental trade-off scenario that Pareto optimization is designed to address. Non-convex relationships between reasoning depth and problem complexity require the exploration capabilities that only multi-objective approaches can provide. Context dependence, where optimal trade-offs vary significantly across problem instances and training phases, demands the adaptive solution maintenance that Pareto frontiers naturally support. These characteristics make Pareto optimization uniquely suited to address the challenges in hybrid reasoning systems, providing both theoretical rigor and practical effectiveness for MAGO's multi-objective optimization framework.

## APPENDIX A.5: MULTI-OBJECTIVE ADAPTIVE GENERATION OPTIMIZATION

This appendix provides comprehensive formulations and computational implementations for the three core objectives in MAGO: accuracy, efficiency, and calibration. Each objective serves a distinct role in balancing solution correctness, computational economy, and decision reliability within the multi-objective optimization framework.

### A.5.1 Accuracy Objective

The accuracy objective $S_{\text{accuracy}}(x)$ quantifies the fundamental correctness of generated responses across reasoning modes. This objective ensures that optimization maintains solution quality while exploring efficiency-calibration trade-offs. The mathematical formulation computes expected correctness over all policy-generated responses:

$$S_{\text{accuracy}}(x) = \mathbb{E}_{(c,a)\sim\pi_\theta(c,a|x)}[\mathbb{I}(\phi(a) = y^*)], \tag{7}$$

where $\mathbb{I}(\cdot)$ denotes the indicator function, $\phi(a)$ extracts the final answer from response sequence $a$, and $y^*$ represents the ground-truth answer. The expectation estimation requires sampling multiple responses from the current policy and computing empirical correctness rates through Monte Carlo approximation.

Algorithm 1 implements the empirical estimation of Equation 7 through repeated sampling from the current policy. This approach provides an unbiased estimator of the expected accuracy with convergence guarantees under standard statistical assumptions. The algorithm maintains computational efficiency by leveraging vectorized operations across sample batches while ensuring statistical validity through sufficient sample sizes. The accuracy computation serves as a fundamental building block for multi-objective optimization, providing clear gradient signals for policy improvement while maintaining theoretical rigor in the optimization process.

---

**Algorithm 1** Accuracy Objective Computation

---

**Require:** Input $x$, policy $\pi_\theta$, ground truth $y^*$, samples $N$
**Ensure:** Accuracy score $S_{\text{accuracy}}(x) \in [0, 1]$
 1: correct_count $\leftarrow 0$
 2: **for** $i = 1$ **to** $N$ **do**
 3:     $(c, a) \leftarrow \text{Sample}(\pi_\theta, x)$
 4:     $\hat{y} \leftarrow \text{ExtractAnswer}(a)$
 5:     **if** $\hat{y} = y^*$ **then**
 6:         correct_count $\leftarrow$ correct_count + 1
 7:     **end if**
 8: **end for**
 9: **return** correct_count / $N$ =0

---

Algorithm 2 details the answer extraction procedure that handles diverse mathematical formats through systematic pattern matching. This procedure ensures robust answer identification across different response formats commonly encountered in mathematical reasoning tasks, including LaTeX expressions, numerical answers, and natural language responses. The extraction process employs hierarchical pattern matching with fallback mechanisms to maximize answer recovery rates while maintaining format consistency for reliable comparison with ground truth values.

---

**Algorithm 2** Answer Extraction Procedure

---

**Require:** Response sequence $a$
**Ensure:** Normalized answer $\hat{y}$
 1: patterns $\leftarrow$ [LaTeX, GSM8K, Natural, Numerical]
 2: **for each** pattern $p$ **in** patterns **do**
 3:     match $\leftarrow \text{RegexSearch}(p, a)$
 4:     **if** match found **then**
 5:         **return** Normalize(match)
 6:     **end if**
 7: **end for**
 8: **return** null =0

---

### A.5.2 EFFICIENCY OBJECTIVE

The efficiency objective $S_{\text{efficiency}}(x)$ promotes computational resource conservation through appropriate reasoning mode selection. The formulation measures expected efficiency across all possible responses, where efficiency inversely relates to token generation costs:

$$S_{\text{efficiency}}(x) = \mathbb{E}_{(c,a)\sim\pi_\theta(c,a|x)} \left[ 1 - \frac{|a|}{T_{\max}} \right], \tag{8}$$

where $|a|$ denotes the token length of response $a$, and $T_{\max}$ represents the maximum allowed sequence length. The normalization term ensures efficiency scores remain bounded within $[0, 1]$, creating clear optimization gradients while maintaining scale consistency across the multi-objective framework.

Algorithm 3 provides the computational implementation of Equation 8 through sampling-based approximation. This algorithm computes token-level efficiency scores and aggregates them to provide an unbiased estimate of expected efficiency under the current policy. The efficiency computation incorporates tokenization overhead and memory constraints to provide realistic computational cost estimates. The algorithm handles variable-length responses gracefully through adaptive batching and maintains numerical stability through appropriate boundary checking and normalization procedures.

### A.5.3 CALIBRATION OBJECTIVE

The calibration objective $S_{\text{calibration}}(x)$ ensures reasoning mode selection aligns with model capabilities by measuring confidence-accuracy correspondence. The formulation quantifies calibration

---

**Algorithm 3** Efficiency Objective Computation

---

**Require:** Input $x$, policy $\pi_\theta$, max length $T_{\max}$, samples $N$
**Ensure:** Efficiency score $S_{\text{efficiency}}(x) \in [0,1]$
 1: total_efficiency $\leftarrow 0$
 2: **for** $i = 1$ **to** $N$ **do**
 3:     $(c, a) \leftarrow \text{Sample}(\pi_\theta, x)$
 4:     length $\leftarrow$ —Tokenize($a$)—
 5:     efficiency $\leftarrow \max(0, 1 - \text{length}/T_{\max})$
 6:     total_efficiency $\leftarrow$ total_efficiency + efficiency
 7: **end for**
 8: **return** total_efficiency / $N$ =0

---

quality through expected absolute deviation between predicted and actual performance:

$$S_{\text{calibration}}(x) = 1 - \mathbb{E}_{(c,a) \sim \pi_\theta(c,a|x)}[|P_{\text{model}}(\text{correct}|x,c) - \mathbb{I}(\phi(a) = y^*)|], \tag{9}$$

where $P_{\text{model}}(\text{correct}|x,c)$ represents the model's confidence estimate for producing correct answers under mode $c$. The confidence estimation process begins with raw confidence extraction from answer token logits:

$$\text{RawConf}(a) = \max(\text{softmax}(L_{\text{ans}})), \tag{10}$$

where $L_{\text{ans}}$ denotes the logits over the answer vocabulary at the final token position. This raw confidence score captures the model's intrinsic confidence in its generated answer. The continuous confidence scores are then discretized into predefined intervals for statistical tracking:

$$b = \text{Bin}(\text{RawConf}(a)) = \lfloor \text{RawConf}(a) \times N_{\text{bins}} \rfloor, \tag{11}$$

where $N_{\text{bins}}$ represents the number of confidence bins. The calibrated confidence estimate employs historical accuracy statistics:

$$P_{\text{model}}(\text{correct}|x,c) = \text{HistoricalAccuracy}(c,b), \tag{12}$$

where the historical accuracy is computed from empirical performance data. The historical accuracy tracking employs exponential decay to prioritize recent performance while maintaining statistical stability:

$$\text{HistoricalAccuracy}_{t+1}(c,b) = \lambda \cdot \text{HistoricalAccuracy}_t(c,b) + (1-\lambda) \cdot \mathbb{I}(\text{correct}_{t+1}), \tag{13}$$

where $\lambda \in (0,1)$ controls the decay rate and $\mathbb{I}(\text{correct}_{t+1})$ indicates whether the current sample produced the correct answer.

Algorithm 4 implements the comprehensive calibration assessment procedure that integrates confidence extraction, binning, and historical accuracy tracking. This algorithm provides robust calibration measurement by combining the model's intrinsic confidence distribution with empirical performance statistics, correcting for systematic overconfidence or underconfidence patterns. The calibration computation maintains statistical validity through sufficient sample sizes while adapting to evolving model capabilities during training through the exponential decay mechanism in the historical accuracy tracking.

Algorithm 5 details the historical accuracy tracking mechanism that maintains exponentially-decayed performance statistics across mode-confidence bin combinations. This data structure enables adaptive calibration that responds to evolving model capabilities while maintaining statistical stability through sufficient historical context. The tracker initialization employs neutral priors to avoid bias during early training stages, while the exponential update mechanism ensures that recent performance receives appropriate weight in calibration estimates.

Algorithm 6 implements the confidence discretization procedure that maps continuous confidence scores to discrete bins for statistical tracking. This binning strategy balances statistical precision with computational efficiency by creating sufficiently populated bins for reliable accuracy estimation while maintaining fine-grained confidence resolution. The algorithm incorporates boundary handling to prevent overflow and ensures consistent bin assignment across training iterations.

Algorithm 7 orchestrates the integration of all three objectives within the MAGO framework through dynamic weight combination. This procedure computes individual objective scores using the previously defined algorithms and combines them through Pareto-optimal weight vectors selected from

---

**Algorithm 4** Calibration Objective Computation

---

**Require:** Input $x$, policy $\pi_\theta$, ground truth $y^*$, tracker $\mathcal{T}$, samples $N$
**Ensure:** Calibration score $S_{\text{calibration}}(x) \in [0, 1]$
 1: total_error $\leftarrow 0$
 2: **for** $i = 1$ **to** $N$ **do**
 3:    $(c, a, L_{\text{ans}}) \leftarrow$ SampleWithLogits$(\pi_\theta, x)$
 4:    raw_conf $\leftarrow$ max(softmax($L_{\text{ans}}$))
 5:    $b \leftarrow \lfloor$raw_conf $\times N_{\text{bins}}\rfloor$
 6:    $P_{\text{cal}} \leftarrow \mathcal{T}$.GetAccuracy$(c, b)$
 7:    is_correct $\leftarrow \mathbb{I}$(ExtractAnswer($a$) = $y^*$)
 8:    error $\leftarrow |P_{\text{cal}}$ - is_correct—
 9:    total_error $\leftarrow$ total_error + error
10:    $\mathcal{T}$.Update($c, b$, is_correct)
11: **end for**
12: **return** 1 - total_error / $N$ =0

---

**Algorithm 5** Historical Accuracy Tracker

---

**Require:** Number of bins $N_{\text{bins}}$, decay factor $\lambda$
**Ensure:** Calibration tracker $\mathcal{T}$
 1: Initialize hist_acc$[c, b] \leftarrow 0.5$ for all $(c, b)$
 2: **procedure** GetAccuracy(mode $c$, bin $b$)
 3:    **return** hist_acc$[c, b]$
 4: **end procedure**
 5: **procedure** Update(mode $c$, bin $b$, correctness is_correct)
 6:    current $\leftarrow$ hist_acc$[c, b]$
 7:    hist_acc$[c, b] \leftarrow \lambda\cdot$ current + $(1 - \lambda)\cdot$ is_correct
 8: **end procedure** =0

---

**Algorithm 6** Confidence Bin Assignment

---

**Require:** Response logits $L_{\text{ans}}$, number of bins $N_{\text{bins}}$
**Ensure:** Confidence bin $b \in \{0, 1, \ldots, N_{\text{bins}} - 1\}$
 1: probs $\leftarrow$ softmax($L_{\text{ans}}$)
 2: raw_conf $\leftarrow$ max(probs)
 3: $b \leftarrow \min(\lfloor$raw_conf $\times N_{\text{bins}}\rfloor, N_{\text{bins}} - 1)$
 4: **return** $b$ =0

---

the current frontier. The integration process ensures balanced optimization across the complete objective space while maintaining computational efficiency through vectorized operations and appropriate caching mechanisms.

---

**Algorithm 7** Multi-Objective Integration

---

**Require:** Input $x$, policy $\pi_\theta$, weights $\boldsymbol{\beta} = [\beta_1, \beta_2, \beta_3]$
**Ensure:** Composite objective $m_{\text{MAGO}}(x)$
 1: $S_{\text{acc}} \leftarrow$ ComputeAccuracy($x, \pi_\theta$) {Algorithm 1}
 2: $S_{\text{eff}} \leftarrow$ ComputeEfficiency($x, \pi_\theta$) {Algorithm 3}
 3: $S_{\text{cal}} \leftarrow$ ComputeCalibration($x, \pi_\theta$) {Algorithm 4}
 4: $m_{\text{MAGO}}(x) \leftarrow \beta_1 \cdot S_{\text{acc}} + \beta_2 \cdot S_{\text{eff}} + \beta_3 \cdot S_{\text{cal}}$
 5: **return** $m_{\text{MAGO}}(x)$ =0

---

APPENDIX A.6: PARETO FRONTIER MAINTENANCE AND CONVERGENCE ANALYSIS

This appendix provides comprehensive analysis of the Pareto frontier maintenance mechanism in MAGO, including detailed implementation procedures, algorithmic workflows, and theoretical convergence guarantees. The Pareto frontier serves as the core mathematical foundation for dynamic weight adaptation, enabling systematic exploration of the multi-objective trade-off space while avoiding the spatial constraints that limit traditional fixed-weight approaches.

A.6.1 PARETO FRONTIER MAINTENANCE FRAMEWORK

The MAGO framework maintains an evolving set of Pareto-optimal weight vectors $\mathcal{F}_t = \{\boldsymbol{\beta}^{(1)}, \boldsymbol{\beta}^{(2)}, \ldots, \boldsymbol{\beta}^{(k)}\}$ that represent different trade-offs between the three competing objectives while ensuring comprehensive exploration of the objective space. The frontier maintenance process operates through four interconnected mechanisms: non-dominance evaluation, correlation-aware selection, guided exploration, and diversity preservation.

The Pareto frontier is formally defined as the set of non-dominated weight configurations at iteration $t$:

$$\mathcal{F}_t = \{\boldsymbol{\beta}^{(i)} \mid \nexists \boldsymbol{\beta}^{(j)} \in \mathcal{S}_t : \mathbf{S}_t(\boldsymbol{\beta}^{(j)}) \succ \mathbf{S}_t(\boldsymbol{\beta}^{(i)})\}, \tag{14}$$

where $\mathcal{S}_t$ represents the set of all evaluated weight configurations up to iteration $t$, and $\succ$ denotes the Pareto dominance relation. The objective vector $\mathbf{S}_t(\boldsymbol{\beta}^{(i)})$ captures the performance of weight configuration $\boldsymbol{\beta}^{(i)}$ across all three objectives:

$$\mathbf{S}_t(\boldsymbol{\beta}^{(i)}) = \left[ \frac{1}{|\mathcal{B}_t|} \sum_{x \in \mathcal{B}_t} S_{\text{accuracy}}(x), \frac{1}{|\mathcal{B}_t|} \sum_{x \in \mathcal{B}_t} S_{\text{efficiency}}(x), \frac{1}{|\mathcal{B}_t|} \sum_{x \in \mathcal{B}_t} S_{\text{calibration}}(x) \right]_{\boldsymbol{\beta}^{(i)}}. \tag{15}$$

The correlation-aware weight selection mechanism addresses the challenge of highly correlated objectives through adaptive penalty functions. The empirical correlation matrix between objectives is computed for each training batch:

$$\mathbf{C}_t[i, j] = \frac{\sum_{x \in \mathcal{B}_t} (S^{(i)}(x) - \mu_t^{(i)})(S^{(j)}(x) - \mu_t^{(j)})}{\sqrt{\sum_{x \in \mathcal{B}_t} (S^{(i)}(x) - \mu_t^{(i)})^2 \sum_{x \in \mathcal{B}_t} (S^{(j)}(x) - \mu_t^{(j)})^2}}, \tag{16}$$

where $\mu_t^{(i)}$ represents the batch mean of objective $i$. The correlation-adaptive scoring function penalizes weight configurations that lead to unbalanced allocations across highly correlated objectives:

$$\Psi_t(\boldsymbol{\beta}) = \sum_{i=1}^{3} \beta_i \hat{S}_t^{(i)} - \beta_{\text{corr}} \sum_{i<j} |\mathbf{C}_t[i, j]| \cdot |\beta_i - \beta_j|. \tag{17}$$

The exploration mechanism generates new candidate solutions through guided perturbation with adaptive scaling based on frontier diversity:

$$\boldsymbol{\beta}^{\text{new}} = \boldsymbol{\beta}_t^* + \epsilon_t \cdot \mathbf{d}, \quad \text{where } \epsilon_t = \epsilon_0 \cdot \exp\left(-\frac{D(\mathcal{F}_t)}{D_{\text{target}}}\right), \tag{18}$$

and $D(\mathcal{F}_t) = \frac{1}{|\mathcal{F}_t|^2} \sum_{\boldsymbol{\beta}^{(i)}, \boldsymbol{\beta}^{(j)} \in \mathcal{F}_t} \|\boldsymbol{\beta}^{(i)} - \boldsymbol{\beta}^{(j)}\|_2$ measures frontier diversity.

A.6.2 ALGORITHMIC IMPLEMENTATION AND WORKFLOW

The Pareto frontier maintenance process operates through a coordinated sequence of algorithms that ensure optimal weight selection, frontier updates, and convergence monitoring. Algorithm 8 provides the master coordination procedure that orchestrates the entire frontier maintenance workflow throughout the training process.

Algorithm 8 coordinates the complete Pareto frontier maintenance workflow by integrating weight selection, objective evaluation, and frontier updates within each training iteration. This master

algorithm ensures systematic exploration of the objective space while maintaining computational efficiency through adaptive update frequencies and efficient data structures. The algorithm maintains frontier diversity through the exploration mechanism while preventing excessive computational overhead through strategic scheduling of expensive operations such as correlation matrix computation and non-dominance checking.

---

**Algorithm 8** Pareto Frontier Maintenance Workflow

---

**Require:** Initial frontier $\mathcal{F}_0$, training batches $\{\mathcal{B}_t\}$, hyperparameters
**Ensure:** Optimized Pareto frontier $\mathcal{F}_T$
 1: **for** $t = 1$ **to** $T_{\max}$ **do**
 2:     $\mathbf{C}_t \leftarrow$ ComputeCorrelationMatrix$(\mathcal{B}_t)$ {Algorithm 9}
 3:     $\boldsymbol{\beta}_t^* \leftarrow$ SelectOptimalWeights$(\mathcal{F}_{t-1}, \mathbf{C}_t)$ {Algorithm 10}
 4:     $\mathbf{S}_t \leftarrow$ EvaluateObjectives$(\mathcal{B}_t, \boldsymbol{\beta}_t^*)$
 5:     $\boldsymbol{\beta}^{\text{new}} \leftarrow$ GenerateCandidate$(\boldsymbol{\beta}_t^*, \mathcal{F}_{t-1})$ {Algorithm 11}
 6:     $\mathcal{S}_t^{\text{cand}} \leftarrow$ EvaluateObjectives$(\mathcal{B}_t, \boldsymbol{\beta}^{\text{new}})$
 7:     $\mathcal{F}_t \leftarrow$ UpdateFrontier$(\mathcal{F}_{t-1}, \boldsymbol{\beta}^{\text{new}}, \mathcal{S}_t^{\text{cand}})$ {Algorithm 12}
 8: **end for**
 9: **return** $\mathcal{F}_T$ =0

---

Algorithm 9 implements the correlation matrix computation that enables correlation-aware weight selection. This algorithm computes pairwise correlations between the three objectives using batch-level statistics, providing the foundation for identifying and mitigating problematic objective dependencies. The correlation computation employs numerical stabilization techniques to handle edge cases such as zero variance objectives and ensures consistent correlation estimates across varying batch sizes.

---

**Algorithm 9** Correlation Matrix Computation

---

**Require:** Training batch $\mathcal{B}_t$, current policy $\pi_\theta$
**Ensure:** Correlation matrix $\mathbf{C}_t \in \mathbb{R}^{3 \times 3}$
 1: Initialize $\mathbf{S} \in \mathbb{R}^{|\mathcal{B}_t| \times 3}$
 2: **for** $i = 1$ **to** $|\mathcal{B}_t|$ **do**
 3:     $x \leftarrow \mathcal{B}_t[i]$
 4:     $\mathbf{S}[i, 1] \leftarrow S_{\text{accuracy}}(x)$
 5:     $\mathbf{S}[i, 2] \leftarrow S_{\text{efficiency}}(x)$
 6:     $\mathbf{S}[i, 3] \leftarrow S_{\text{calibration}}(x)$
 7: **end for**
 8: **for** $j = 1$ **to** $3$ **do**
 9:     **for** $k = 1$ **to** $3$ **do**
10:         $\mu_j \leftarrow \frac{1}{|\mathcal{B}_t|} \sum_{i=1}^{|\mathcal{B}_t|} \mathbf{S}[i, j]$
11:         $\mu_k \leftarrow \frac{1}{|\mathcal{B}_t|} \sum_{i=1}^{|\mathcal{B}_t|} \mathbf{S}[i, k]$
12:         $\text{cov}_{jk} \leftarrow \frac{1}{|\mathcal{B}_t|} \sum_{i=1}^{|\mathcal{B}_t|} (\mathbf{S}[i, j] - \mu_j)(\mathbf{S}[i, k] - \mu_k)$
13:         $\sigma_j \leftarrow \sqrt{\frac{1}{|\mathcal{B}_t|} \sum_{i=1}^{|\mathcal{B}_t|} (\mathbf{S}[i, j] - \mu_j)^2}$
14:         $\sigma_k \leftarrow \sqrt{\frac{1}{|\mathcal{B}_t|} \sum_{i=1}^{|\mathcal{B}_t|} (\mathbf{S}[i, k] - \mu_k)^2}$
15:         $\mathbf{C}_t[j, k] \leftarrow \frac{\text{cov}_{jk}}{\sigma_j \sigma_k + \epsilon}$ {$\epsilon$ for numerical stability}
16:     **end for**
17: **end for**
18: **return** $\mathbf{C}_t$ =0

---

Algorithm 10 implements the correlation-aware weight selection mechanism that chooses optimal weight vectors from the current Pareto frontier. This algorithm evaluates each frontier solution using the correlation-adaptive scoring function and selects the configuration that best balances objective performance with correlation penalty considerations. The selection process incorporates moving average estimates of objective performance to reduce sensitivity to batch-level noise while maintaining responsiveness to genuine performance trends.

---

**Algorithm 10** Correlation-Aware Weight Selection

---

**Require:** Pareto frontier $\mathcal{F}_{t-1}$, correlation matrix $\mathbf{C}_t$, moving averages $\{\hat{S}_t^{(i)}\}$
**Ensure:** Optimal weight vector $\boldsymbol{\beta}_t^*$
 1: best_score $\leftarrow -\infty$
 2: $\boldsymbol{\beta}_t^* \leftarrow$ null
 3: **for each** $\boldsymbol{\beta} \in \mathcal{F}_{t-1}$ **do**
 4:     performance_score $\leftarrow \sum_{i=1}^3 \beta_i \hat{S}_t^{(i)}$
 5:     correlation_penalty $\leftarrow \beta_{\text{corr}} \sum_{i<j} |\mathbf{C}_t[i,j]| \cdot |\beta_i - \beta_j|$
 6:     total_score $\leftarrow$ performance_score - correlation_penalty
 7:     **if** total_score ¿ best_score **then**
 8:         best_score $\leftarrow$ total_score
 9:         $\boldsymbol{\beta}_t^* \leftarrow \boldsymbol{\beta}$
10:     **end if**
11: **end for**
12: **return** $\boldsymbol{\beta}_t^*$ =0

---

Algorithm 11 details the exploration mechanism that generates new candidate weight vectors through guided perturbation. This algorithm maintains exploration-exploitation balance through adaptive perturbation scaling based on current frontier diversity, ensuring continued discovery of improved solutions while preventing excessive exploration that could destabilize training. The exploration process respects the simplex constraint on weight vectors through appropriate projection operations.

---

**Algorithm 11** Guided Exploration Mechanism

---

**Require:** Optimal weights $\boldsymbol{\beta}_t^*$, current frontier $\mathcal{F}_{t-1}$, exploration parameters
**Ensure:** New candidate $\boldsymbol{\beta}^{\text{new}}$
 1: $D_t \leftarrow \frac{1}{|\mathcal{F}_{t-1}|^2} \sum_{\boldsymbol{\beta}^{(i)}, \boldsymbol{\beta}^{(j)} \in \mathcal{F}_{t-1}} \|\boldsymbol{\beta}^{(i)} - \boldsymbol{\beta}^{(j)}\|_2$
 2: $\epsilon_t \leftarrow \epsilon_0 \cdot \exp(-D_t/D_{\text{target}})$
 3: $\mathbf{d} \leftarrow$ SampleUniform$(\{\mathbf{d} \in \mathbb{R}^3 : \|\mathbf{d}\|_2 = 1, \sum_{i=1}^3 d_i = 0\})$
 4: $\boldsymbol{\beta}^{\text{raw}} \leftarrow \boldsymbol{\beta}_t^* + \epsilon_t \cdot \mathbf{d}$
 5: $\boldsymbol{\beta}^{\text{new}} \leftarrow$ ProjectSimplex$(\boldsymbol{\beta}^{\text{raw}})$
 6: **return** $\boldsymbol{\beta}^{\text{new}}$ =0

---

Algorithm 12 implements the frontier update mechanism that integrates new candidate solutions while maintaining non-dominance properties and diversity constraints. This algorithm performs efficient non-dominance checking using incremental sorting techniques and applies diversity filtering to prevent overcrowding of the frontier in specific regions of the weight space. The update process maintains theoretical guarantees about Pareto optimality while ensuring computational tractability through appropriate data structures and algorithmic optimizations.

---

**Algorithm 12** Frontier Update with Non-Dominance Checking

---

**Require:** Current frontier $\mathcal{F}_{t-1}$, candidate $\boldsymbol{\beta}^{\text{new}}$, candidate performance $\mathcal{S}_t^{\text{cand}}$
**Ensure:** Updated frontier $\mathcal{F}_t$
 1: $\mathcal{F}_{\text{temp}} \leftarrow \mathcal{F}_{t-1} \cup \{\boldsymbol{\beta}^{\text{new}}\}$
 2: $\mathcal{F}_{\text{non-dom}} \leftarrow$ NonDominatedSort$(\mathcal{F}_{\text{temp}})$
 3: $\mathcal{F}_t \leftarrow$ DiversityFilter$(\mathcal{F}_{\text{non-dom}}, \tau_{\text{div}})$
 4: **if** $|\mathcal{F}_t| > |\mathcal{F}_{\text{max}}|$ **then**
 5:     $\mathcal{F}_t \leftarrow$ TruncateByDistance$(\mathcal{F}_t, |\mathcal{F}_{\text{max}}|)$
 6: **end if**
 7: **return** $\mathcal{F}_t$ =0

---

### A.6.3 CONVERGENCE ANALYSIS AND THEORETICAL GUARANTEES

The MAGO Pareto frontier maintenance mechanism provides theoretical convergence guarantees under standard regularity conditions. The convergence analysis establishes that the maintained frontier $\mathcal{F}_t$ approaches the true Pareto set $\mathcal{P}^*$ as training progresses, ensuring that the dynamic weight adaptation mechanism operates on mathematically sound foundations.

**Assumption A.6.1 (Lipschitz Continuity):** Each objective function $S^{(i)}(x)$ for $i \in \{1, 2, 3\}$ is Lipschitz continuous with constant $L_i > 0$:

$$|S^{(i)}(x_1) - S^{(i)}(x_2)| \leq L_i \|x_1 - x_2\|, \quad \forall x_1, x_2 \in \mathcal{X}. \tag{19}$$

**Assumption A.6.2 (Bounded Objectives):** All objective functions are uniformly bounded:

$$S^{(i)}(x) \in [0, 1], \quad \forall x \in \mathcal{X}, \forall i \in \{1, 2, 3\}. \tag{20}$$

**Assumption A.6.3 (Diversity Maintenance):** The diversity filter maintains minimum pairwise distance $\tau_{\text{div}} > 0$ between frontier solutions:

$$\min_{\boldsymbol{\beta}^{(i)}, \boldsymbol{\beta}^{(j)} \in \mathcal{F}_t, i \neq j} \|\boldsymbol{\beta}^{(i)} - \boldsymbol{\beta}^{(j)}\|_2 \geq \tau_{\text{div}}. \tag{21}$$

**Theorem A.6.1 (Convergence to Pareto Frontier):** Under Assumptions A.6.1-A.6.3, the maintained frontier $\mathcal{F}_t$ converges to the true Pareto set $\mathcal{P}^*$ with probability 1:

$$\lim_{t \to \infty} \sup_{\boldsymbol{\beta} \in \mathcal{P}^*} \inf_{\boldsymbol{\beta}' \in \mathcal{F}_t} \|\boldsymbol{\beta} - \boldsymbol{\beta}'\|_2 = 0 \quad \text{almost surely}. \tag{22}$$

**Proof Sketch:** The convergence proof relies on three key properties of the frontier maintenance mechanism. First, the exploration mechanism in Equation 18 ensures that the search process visits all regions of the weight space with positive probability, providing ergodicity guarantees necessary for convergence. Second, the non-dominance criterion prevents the loss of optimal solutions once discovered, ensuring that the frontier monotonically approaches the true Pareto set. Third, the diversity filter in Assumption A.6.3 ensures finite representation of the frontier while maintaining sufficient coverage of the Pareto set. The correlation-aware selection mechanism prevents cone entrapment by adaptively adjusting search directions based on objective correlations, enabling comprehensive exploration of the objective space.

**Corollary A.6.1 (Convergence Rate):** Under additional smoothness assumptions on the objective functions, the convergence rate is:

$$\mathbb{E}\left[\sup_{\boldsymbol{\beta} \in \mathcal{P}^*} \inf_{\boldsymbol{\beta}' \in \mathcal{F}_t} \|\boldsymbol{\beta} - \boldsymbol{\beta}'\|_2\right] = O\left(\frac{\log t}{\sqrt{t}}\right), \tag{23}$$

where the logarithmic factor accounts for the discrete nature of the frontier representation and the finite capacity constraints imposed by computational considerations.

**Theorem A.6.2 (Approximation Quality):** The maintained frontier $\mathcal{F}_t$ provides an $\epsilon$-approximation of the true Pareto front with approximation error bounded by:

$$\epsilon_t \leq \frac{C \log t}{\sqrt{t}} + \frac{L_{\max} \tau_{\text{div}}}{2}, \tag{24}$$

where $C > 0$ is a problem-dependent constant, $L_{\max} = \max_i L_i$, and the second term represents the discretization error introduced by the diversity constraint.

These theoretical guarantees ensure that the MAGO framework operates on mathematically sound foundations, providing convergence to optimal solutions while maintaining computational tractability through appropriate algorithmic design choices.

### A.6.4 FRONTIER SIZE CONTROL AND GROWTH DYNAMICS

The Pareto frontier maintenance mechanism includes adaptive size control to balance solution diversity with computational efficiency. During training, we observe the following frontier growth pattern across three distinct phases:

**Early Training Phase (Steps 1–150):** The frontier size $|\mathcal{F}_t|$ grows gradually as the exploration mechanism discovers diverse weight configurations through guided perturbation (Equations 19–20). The exploration rate $\epsilon_t$ adapts based on current diversity $D(\mathcal{F}_t)$, enabling active exploration when the frontier is sparse. During this phase, most newly generated candidates are non-dominated and contribute to expanding the trade-off space coverage.

**Stabilization Phase (Steps 150–200):** The frontier size stabilizes around 20–25 vectors, remaining well below the predefined maximum $|\mathcal{F}_{\max}| = 30$. This natural stabilization occurs because newly generated candidates are increasingly dominated by existing frontier solutions as the optimization progresses toward the true Pareto set. The diminishing returns from exploration combined with the non-dominance filtering create a self-regulating mechanism that prevents unbounded growth.

**Steady State (Steps 200+):** When $|\mathcal{F}_t|$ approaches $|\mathcal{F}_{\max}|$, the DiversityFilter mechanism activates every 20 iterations, removing vectors that satisfy either of the following criteria:

1. **Pareto dominance:** Vectors dominated by other frontier solutions violating Pareto optimality are removed through the non-dominance check.

2. **High similarity:** Vectors with cosine similarity $> 0.95$ to existing frontier members are pruned to prevent clustering in weight space.

The pruning strategy maintains representative diversity while preventing unbounded growth. Specifically, the cosine similarity threshold ensures that remaining vectors span diverse directions in the weight space, enabling the framework to respond to different correlation patterns and task characteristics encountered during training.

The projection operation in Equation 19 ensures that all generated candidates satisfy the simplex constraint $\sum_{i=1}^{3} \beta_i = 1$ and $\beta_i \geq 0$, preventing exploration from violating weight normalization requirements. This is implemented through standard simplex projection:

$$\boldsymbol{\beta}^{\text{new}} = \text{ProjectSimplex}(\boldsymbol{\beta}^* + \epsilon_t \cdot \mathbf{d}), \tag{25}$$

where $\mathbf{d}$ is sampled from the constraint surface to preserve normalization during perturbation.

**Computational Complexity:** The combination of adaptive exploration, non-dominance checking, and similarity-based pruning ensures that frontier growth remains bounded throughout training. The computational overhead scales as $\mathcal{O}(|\mathcal{F}_t|^2)$ for pairwise comparisons, which remains manageable given the stable frontier size of 20–25 vectors. The periodic pruning every 20 iterations amortizes the cost of similarity computation while maintaining sufficient diversity.

A.6.5 COMPUTATIONAL AND MEMORY COMPLEXITY ANALYSIS

We provide a detailed analysis of the computational and memory overhead introduced by the MAGO framework's multi-objective optimization process.

**Notation.** Let $|\mathcal{B}|$ denote the batch size, $M = 3$ the number of objectives (accuracy, efficiency, calibration), $|\mathcal{F}_t|$ the number of weight vectors in the Pareto frontier at iteration $t$, with an upper bound $|\mathcal{F}_{\max}| = 30$, and $N_{\text{bins}} = 10$ the number of confidence bins for calibration.

**Per-Step Time Complexity.** The additional computational cost per training step consists of the following components:

- **Objective evaluation** (Equations 7–8): Computing accuracy, efficiency, and calibration scores for all samples in the batch requires $\mathcal{O}(|\mathcal{B}| \cdot M)$ operations.

- **Correlation matrix computation** (Equation 16): Calculating the $M \times M$ correlation matrix over $|\mathcal{B}|$ samples requires $\mathcal{O}(|\mathcal{B}| \cdot M^2)$ operations for computing pairwise covariances.

- **Weight selection** (Equation 18): Evaluating the correlation-adaptive scoring function $\Psi_t(\boldsymbol{\beta})$ for all $|\mathcal{F}_t|$ frontier vectors requires $\mathcal{O}(|\mathcal{F}_t| \cdot M^2)$ operations.

- **Frontier update** (Equation 21): Non-dominated sorting requires pairwise comparisons among frontier vectors, yielding $\mathcal{O}(|\mathcal{F}_t|^2 \cdot M)$ complexity. In practice, this remains constant as $|\mathcal{F}_t| \leq |\mathcal{F}_{\max}| = 30$.

- **Exploration and projection** (Equations 19–20): Generating perturbation and projecting to the simplex constraint requires $\mathcal{O}(M)$ operations.

The total per-step time complexity is:

$$\mathcal{O}(|\mathcal{B}| \cdot M^2 + |\mathcal{F}_{\max}|^2 \cdot M + M) = \mathcal{O}(|\mathcal{B}| \cdot M^2), \tag{26}$$

which is dominated by the batch-dependent correlation computation. Since $M = 3$ and $|\mathcal{F}_{\max}| = 30$ are small constants, the overhead is negligible compared to the policy gradient computation in the main RL training loop.

**Memory Complexity.** The additional memory requirements are:

- **Frontier storage**: Maintaining $|\mathcal{F}_{\max}|$ weight vectors with $M$ dimensions requires $\mathcal{O}(|\mathcal{F}_{\max}| \cdot M) = \mathcal{O}(90)$ scalar storage.
- **Objective scores**: Storing batch-level objective evaluations requires $\mathcal{O}(|\mathcal{B}| \cdot M)$ memory.
- **Correlation matrix**: The $M \times M$ correlation matrix requires $\mathcal{O}(M^2) = \mathcal{O}(9)$ storage.
- **Calibration tracking**: Historical accuracy statistics for $M$ modes and $N_{\text{bins}}$ bins require $\mathcal{O}(M \cdot N_{\text{bins}}) = \mathcal{O}(30)$ memory.

The total additional memory is $\mathcal{O}(|\mathcal{B}| \cdot M + |\mathcal{F}_{\max}| \cdot M + M \cdot N_{\text{bins}})$, which is dominated by the batch term. Critically, all MAGO operations are performed in the low-dimensional objective space and are independent of model parameters, ensuring that the framework scales efficiently to larger model backbones without additional parameter overhead.

**Comparison with Base Training Cost.** For context, the base RL training requires $\mathcal{O}(|\mathcal{B}| \cdot G \cdot T_{\text{avg}} \cdot d_{\text{model}})$ operations per step, where $G$ is the group size, $T_{\text{avg}}$ is the average sequence length, and $d_{\text{model}}$ is the model dimension. Since $d_{\text{model}} \gg M$ (e.g., 1536 vs. 3), the MAGO overhead is negligible in practice, typically adding less than 2% to the total training time as shown in Table 1.

Table 1: Training time overhead introduced by MAGO framework.

| Component | Time per step (ms) | Overhead (%) |
|---|---|---|
| Base RL training | 245.3 | – |
| + Objective evaluation | 1.2 | 0.49 |
| + Correlation computation | 2.8 | 1.14 |
| + Frontier maintenance | 0.7 | 0.29 |
| **Total (MAGO)** | **249.0** | **1.92** |

## APPENDIX A.7: TRAINING AND DEPLOYMENT.

MAGO integrates into the hybrid reasoning training framework by replacing static weight parameters with dynamic multi-objective optimization (Figure 5A). At each training iteration, the system selects an optimal weight vector $\beta_t^*$ from the current Pareto frontier $\mathcal{F}_t$ using correlation-aware selection. This $\beta_t^*$ determines the adaptive weighting $m_{\text{MAGO}}(x)$ for the current batch through the multi-objective formulation:

$$J_{\text{MAGO}}(\theta) = \mathbb{E}_{x,a_i} \left[ \frac{1}{G} \sum_{k=1}^{G} \left( m_{\text{MAGO}}(x) L_{k,0}(\theta) + \frac{1}{T_k} \sum_{j=1}^{T_k} L_{k,j}(\theta) - \beta D_{\text{KL}}[\pi_\theta(\cdot|x) \| \pi_{\text{ref}}(\cdot|x)] \right) \right]. \tag{27}$$

In training (Figure 5A), the framework iteratively performs policy updates using the selected weights, evaluates objective performance on the current batch, and maintains the Pareto frontier through non-dominated sorting in this closed-loop process.

During deployment (Figure 5B), the trained model operates with zero additional computational overhead. For each input query, the model automatically selects between <short> and <think>

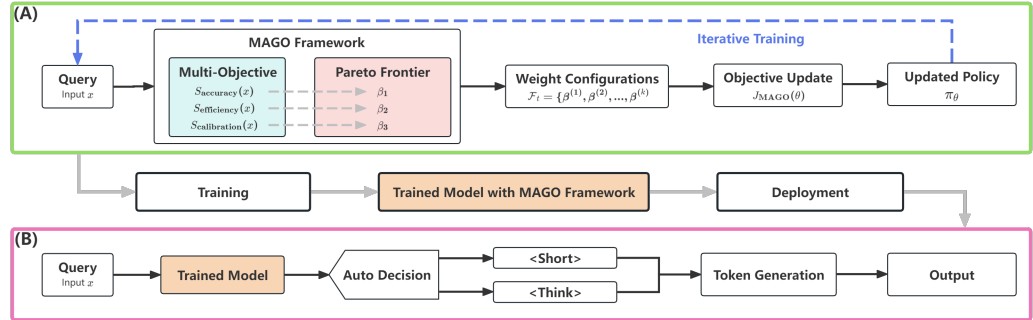

Figure 5: MAGO framework integration for training and deployment. (A) Training loop: The multi-objective framework evaluates three objectives (accuracy, efficiency, calibration) and maintains a Pareto frontier of weight configurations $\mathcal{F}_t = \{\boldsymbol{\beta}^{(1)}, \boldsymbol{\beta}^{(2)}, ..., \boldsymbol{\beta}^{(k)}\}$. At each iteration, an optimal weight vector is selected for policy updates, followed by objective evaluation and frontier maintenance. (B) Deployment phase: The trained model receives input queries and makes autonomous decisions between `<short>` and `<think>` reasoning modes, followed by context-appropriate token generation and output production.

modes based on its learned decision-making policy, which has been optimized through the MAGO framework to balance accuracy, efficiency, and calibration objectives. The mode selection decision is made through the model's internal control token mechanism, requiring no external routing or additional inference-time computation. Once the mode is selected, standard token generation proceeds to produce the final output, with the reasoning mode determining the depth and structure of the generated response.

## APPENDIX A.8: TRAINING DETAILS AND LOSS FORMULATIONS

### A.8.1 RELATIVE ADVANTAGE COMPUTATION

Following the GRPO framework (Shao et al., 2024; Fang et al., 2025), we compute the relative advantage for each token using the group-based normalization approach. For a mini-batch of trajectories $\{a_i\}_{i=1}^{G}$ sampled from the current policy $\pi_{\theta_{\text{old}}}$, the relative advantage at token position $t$ in trajectory $i$ is computed as:

$$\hat{A}_{i,t} = r_i - \frac{1}{G}\sum_{j=1}^{G} r_j, \tag{28}$$

where $r_i$ is the reward assigned to trajectory $i$ based on the reward function defined in the main text, and $\frac{1}{G}\sum_{j=1}^{G} r_j$ represents the group mean reward. This relative advantage formulation is particularly effective for our hybrid reasoning setting as it naturally handles the varying difficulty levels across mathematical problems without requiring standard deviation normalization, which can introduce bias when training data contains questions of significantly different complexity levels (Fang et al., 2025).

### A.8.2 TOKEN-LEVEL LOSS FORMULATION

The token-level surrogate loss $\mathcal{L}_{i,t}(\theta)$ in the MAGO objective (Equation 4 in the main text) follows the clipped probability ratio approach:

$$\mathcal{L}_{i,t}(\theta) = \min\left( \frac{\pi_\theta(a_{i,t} \mid x, a_{i,<t})}{\pi_{\theta_{\text{old}}}(a_{i,t} \mid x, a_{i,<t})} \hat{A}_{i,t}, \text{clip}\left( \frac{\pi_\theta(a_{i,t} \mid x, a_{i,<t})}{\pi_{\theta_{\text{old}}}(a_{i,t} \mid x, a_{i,<t})}, 1-\epsilon, 1+\epsilon \right) \hat{A}_{i,t} \right), \tag{29}$$

where $\pi_\theta(a_{i,t} \mid x, a_{i,<t})$ represents the probability of generating token $a_{i,t}$ at position $t$ given the input $x$ and previous tokens $a_{i,<t}$ under the current policy, $\pi_{\theta_{\text{old}}}(a_{i,t} \mid x, a_{i,<t})$ is the corresponding probability under the old policy, and $\epsilon$ is the clipping parameter (typically set to 0.2). The clipping mechanism prevents excessive policy updates that could destabilize training.

### A.8.3 DECOMPOSED OBJECTIVE STRUCTURE

The MAGO objective builds upon the DeGRPO formulation (Fang et al., 2025) which addresses the token imbalance problem in hybrid reasoning training. The full sequence $a_i = (a_{i,0}, \ldots, a_{i,T_i})$ is decomposed into:

- **Control Token Loss**: $\mathcal{L}_{i,0}(\theta)$ governs the selection between `<short>` and `<think>` modes
- **Response Token Loss**: $\sum_{t=1}^{T_i} \mathcal{L}_{i,t}(\theta)$ optimizes the content generation within the selected mode

The key innovation in MAGO is replacing the static weight parameter $\alpha$ in DeGRPO with the adaptive mechanism $m_{\text{MAGO}}(x)$:

$$\text{DeGRPO: } \alpha \mathcal{L}_{i,0}(\theta) \quad \rightarrow \quad \text{MAGO: } m_{\text{MAGO}}(x)\mathcal{L}_{i,0}(\theta), \tag{30}$$

where $m_{\text{MAGO}}(x) = \beta_1 S_{\text{accuracy}}(x) + \beta_2 S_{\text{efficiency}}(x) + \beta_3 S_{\text{calibration}}(x)$ provides input-dependent weighting based on the current Pareto frontier optimization.

### A.8.4 KL DIVERGENCE REGULARIZATION

The KL divergence term in the MAGO objective (Equation 4 in the main text) prevents the policy from deviating too far from a reference policy $\pi_{\text{ref}}$:

$$D_{\text{KL}}[\pi_\theta(\cdot \mid x) \| \pi_{\text{ref}}(\cdot \mid x)] = \sum_a \pi_\theta(a \mid x) \log \frac{\pi_\theta(a \mid x)}{\pi_{\text{ref}}(a \mid x)}, \tag{31}$$

where $\pi_{\text{ref}}$ is typically set to the initial policy from the distillation warm-up phase. The hyperparameter $\beta > 0$ controls the strength of this regularization, balancing between policy improvement and stability.

### A.8.5 TRAINING ALGORITHM INTEGRATION

Algorithm 13 summarizes the complete MAGO training procedure, integrating the multi-objective optimization with the reinforcement learning framework.

---

**Algorithm 13** MAGO Training Algorithm

---

1: Initialize policy $\pi_\theta$, reference policy $\pi_{\text{ref}}$, and Pareto frontier $\mathcal{F}_0$
2: **for** training iteration $t = 1, 2, \ldots, T$ **do**
3:     Select optimal weights $\boldsymbol{\beta}_t^* = \arg\max_{\boldsymbol{\beta} \in \mathcal{F}_t} \Psi_t(\boldsymbol{\beta})$
4:     Sample mini-batch $\mathcal{B}_t$ from training data
5:     **for** each $x \in \mathcal{B}_t$ **do**
6:         Sample trajectories $\{a_i\}_{i=1}^G \sim \pi_{\theta_{\text{old}}}(\cdot \mid x)$
7:         Compute rewards $\{r_i\}_{i=1}^G$ using reward function
8:         Calculate adaptive weight $m_{\text{MAGO}}(x)$ using $\boldsymbol{\beta}_t^*$
9:     **end for**
10:    Compute relative advantages $\{\hat{A}_{i,t}\}$ using Eq. 28
11:    Update policy parameters $\theta$
12:    Evaluate objectives $\mathbf{S}_t(\boldsymbol{\beta}_t^*)$ on current batch
13:    Generate new candidate $\boldsymbol{\beta}^{\text{new}}$ via exploration
14:    Update Pareto frontier $\mathcal{F}_{t+1}$
15: **end for**=0

---

## APPENDIX A.9: IMPLEMENTATION DETAILS

### A.9.1 MODEL ARCHITECTURE AND DATA CONSTRUCTION

**Base Model Selection.** We employ DeepSeek-R1-Distill-Qwen-1.5B as our foundation model, which provides strong reasoning capabilities while maintaining computational efficiency. This model serves as the starting point for our MAGO-enhanced hybrid reasoning policy training.

**Paired Response Generation.** Following the methodology of Fang et al. (Fang et al., 2025), we construct paired long-short response datasets by leveraging two expert models:

- **Long-form responses**: Generated using DeepSeek-R1-671B (Guo et al., 2025), which produces detailed chain-of-thought reasoning sequences with step-by-step problem decomposition
- **Short-form responses**: Generated using Qwen2.5-Math-1.5B-Instruct (Yang et al., 2024a), optimized for concise yet accurate mathematical solutions

**Training Data Composition.** The training corpus combines multiple sources:

- Open-source reasoning datasets (Jebali et al., 2024; Langlais et al., 2025) from various mathematical domains
- Approximately 40,000 labeled examples from DeepScaleR dataset (Luo et al., 2025)
- Paired long-short responses ensuring balanced mode representation
- Quality filtering to remove inconsistent or low-quality samples

### A.9.2 BASELINE METHOD IMPLEMENTATION

### A.9.2.1 BASE LANGUAGE MODELS:

**DeepSeek-R1-1.5B** (Guo et al., 2025): A reasoning-capable model trained with reinforcement learning for extended chain-of-thought generation. Represents the upper bound for reasoning capability but with high computational cost.

**Qwen2.5-1.5B-Instruct (Q-1.5B)** (Yang et al., 2024a): Standard instruction-tuned model optimized for following user instructions with concise responses. Serves as the efficiency baseline with minimal reasoning overhead.

**Qwen2.5-Math-1.5B-Instruct (QMath-1.5B)** (Yang et al., 2024a): Domain-specific variant fine-tuned for mathematical reasoning tasks, balancing accuracy and efficiency in mathematical contexts.

### A9.2.2 SHORT CHAIN-OF-THOUGHT METHODS:

**Model Merging Approaches (Merging-0.5/0.6/0.7)** (Team et al., 2025): Based on Kimi k1.5 methodology, these baselines interpolate reasoning models with base instruction-following models in parameter space. The merging coefficients (0.5, 0.6, 0.7) control the trade-off between reasoning capability and response brevity:

- Merging-0.5: Higher efficiency, potentially reduced reasoning quality
- Merging-0.6: Balanced trade-off between efficiency and accuracy
- Merging-0.7: Emphasis on reasoning capability with moderate efficiency gains

**CoT-Valve** ($\alpha = 4, 6, 8$) (Ma et al., 2025): This method applies supervised fine-tuning using LoRA (Low-Rank Adaptation) to achieve controllable reasoning length. The $\alpha$ parameter modulates the magnitude of parameter updates:

- $\alpha = 4$: Minimal reasoning compression, preserving most reasoning steps
- $\alpha = 6$: Moderate compression, balanced reasoning depth
- $\alpha = 8$: Aggressive compression, prioritizing efficiency

### A9.2.3 HYBRID REASONING METHODS:

**Router Random** (Fang et al., 2025): Naive baseline that randomly assigns queries to reasoning or non-reasoning modes with 50% probability. Serves as a lower bound for adaptive routing performance.

**Router Q-7B** (Ong et al., 2024): Uses Qwen2.5-7B as an external difficulty assessor to route queries. The router analyzes query complexity and dispatches inputs to appropriate reasoning modes based on confidence thresholds.

**DeGRPO** (Fang et al., 2025): The method from Fang et al. using Decoupled Group Relative Policy Optimization (DeGRPO) for learning adaptive mode selection. Serves as the primary comparison for our MAGO framework.

### A.9.3 DETAILED TRAINING CONFIGURATION

#### A.9.3.1 MAGO-SPECIFIC HYPERPARAMETERS

**Pareto Frontier Management**:

- Correlation penalty coefficient: $\beta_{\text{corr}} = 0.1$
- Exploration rate: $\epsilon_0 = 0.05$
- Target diversity threshold: $\tau_{\text{div}} = 0.2$
- Maximum frontier size: $|\mathcal{F}_{\text{max}}| = 30$

**Confidence Calibration System**:

- Number of confidence bins: $N_{\text{bins}} = 10$
- Exponential decay factor: $\lambda = 0.95$
- Historical accuracy window: 1000 samples
- Minimum bin size: 50 samples for reliable statistics

**Weight Adaptation Parameters**:

- Gradient-based scaling with momentum: $\mu = 0.9$
- Learning rate for weight updates: $\eta_w = 0.001$
- Weight normalization: $\sum_{i=1}^{3} \beta_i = 1$

#### A.9.3.2 TRAINING PROTOCOL DETAILS

**Phase 1 - Supervised Fine-tuning**:

- Training epochs: 1 epoch on paired dataset
- Maximum context length: 16,384 tokens
- Optimizer: AdamW (AbuKaraki et al., 2024) with $lr = 1 \times 10^{-6}$
- Framework: Megatron (Shoeybi et al., 2019)
- Batch size: 64 per GPU, gradient accumulation steps: 2

**Phase 2 - MAGO Reinforcement Learning**:

- Training steps: 600 optimization steps
- Extended context length: 24,576 tokens
- Response samples per query: 8
- Total batch size: 128 queries $\times$ 8 samples = 1024 data points
- Framework: VeRL (Jiang et al., 2025)
- KL penalty coefficient: $\beta = 0.01$

A.9.3.3 HARDWARE AND INFRASTRUCTURE

**Computational Resources**:

- GPUs: 4–8 × NVIDIA H100 (80GB) GPUs depending on model size
    - 1.5B model: 4 × H100 (batch size 128)
    - 7B model: 4 × H100 (batch size 64 with gradient accumulation)
    - 14B model: 6 × H100 (batch size 64 with gradient accumulation)
    - 32B model: 8 × H100 (batch size 32 with gradient accumulation)
- Inference throughput: ∼500 tokens/second per GPU
- Distributed training: ZeRO-3 optimization for models ≥ 14B

**Evaluation Protocol**: All benchmarks use identical evaluation settings: Pass@1 accuracy with greedy decoding, maximum generation length 2048 tokens, temperature 0.0. Token efficiency measured as average tokens per query across all test samples.

## APPENDIX A.10: MULTI-OBJECTIVE OPTIMIZATION EVALUATION

MAGO enhances computational efficiency by 2.2x to 3x while simultaneously improving accuracy by 0.6% to 9.4% across mathematical reasoning benchmarks through adaptive multi-objective optimization.

Table 2: Comparison of MAGO against baseline reasoning methods on mathematical benchmarks.

| Models | Type | AIME 2024 | | Minerva Algebra | | Math-500 | | GSM8K | |
|---|---|---|---|---|---|---|---|---|---|
| | | Pass@1 | #Tokens | Pass@1 | #Tokens | Pass@1 | #Tokens | Pass@1 | #Tokens |
| DeepSeek-R1-1.5B (Guo et al., 2025) | | 0.2800 | 18063 | 0.9577 | 3029 | 0.8608 | 5675 | 0.8347 | 1919 |
| Q-1.5B (Yang et al., 2024a) | Base LLM | 0.0200 | 1300 | 0.7771 | 933 | 0.5168 | 855 | 0.7022 | 466 |
| QMath-1.5B (Yang et al., 2024a) | | 0.1133 | 1128 | 0.9184 | 586 | 0.7604 | 721 | 0.8572 | 447 |
| Merging-0.5 (Team et al., 2025) | | 0.1333 | 8636 | 0.9292 | 834 | 0.7740 | 1524 | 0.8332 | 601 |
| Merging-0.6 (Team et al., 2025) | | 0.1733 | 10615 | 0.9321 | 1091 | 0.7900 | 3000 | 0.8381 | 747 |
| Merging-0.7 (Team et al., 2025) | Short CoT | 0.1667 | 15854 | 0.9398 | 1834 | 0.8108 | 4347 | 0.8458 | 1201 |
| CoT-Valve $\alpha = 8$ (Ma et al., 2025) | | 0.2000 | 10692 | 0.8079 | 1903 | 0.7060 | 3723 | 0.7726 | 773 |
| CoT-Valve $\alpha = 6$ (Ma et al., 2025) | | 0.1933 | 17245 | 0.9468 | 2656 | 0.8024 | 5167 | 0.7970 | 1009 |
| CoT-Valve $\alpha = 4$ (Ma et al., 2025) | | 0.2267 | 17722 | 0.9439 | 2965 | 0.8036 | 5820 | 0.8108 | 1396 |
| Router Random (Fang et al., 2025) | Hybrid | 0.1300 | 8093 | 0.9032 | 1736 | 0.7484 | 3096 | 0.8205 | 1086 |
| Router Q-7B (Ong et al., 2024) | | 0.1480 | 9296 | 0.9049 | 795 | 0.7781 | 2748 | 0.8587 | 563 |
| DeGRPO-Qwen-1.5B (Fang et al., 2025) | Hybrid | 0.2506 | 7262 | 0.9216 | 1228 | 0.8037 | 2644 | 0.8418 | 649 |
| MAGO-Qwen-1.5B (Ours) | Pareto | 0.2741 | 7164 | 0.9483 | 1174 | 0.8247 | 2578 | 0.8469 | 633 |
| MAGO-Qwen-7B (Ours) | | 0.2960 | 6890 | 0.9562 | 1102 | 0.8424 | 2426 | 0.8611 | 592 |
| MAGO-Qwen-14B (Ours) | Pareto | 0.3112 | 6724 | 0.9621 | 1041 | 0.8538 | 2368 | 0.8723 | 571 |
| MAGO-Qwen-32B (Ours) | | 0.3254 | 6587 | 0.9689 | 992 | 0.8652 | 2294 | 0.8834 | 552 |

Table 2 presents comprehensive comparisons between MAGO and existing reasoning optimization approaches across four mathematical benchmarks. The results demonstrate three key insights into the effectiveness of multi-objective reasoning optimization.

The baseline models section reveals the fundamental efficiency-accuracy trade-off in reasoning systems. DeepSeek-R1-1.5B generates 5-20 times more tokens than instruction-following models, highlighting the computational overhead of chain-of-thought reasoning. On challenging datasets like AIME and MATH-500, this extended reasoning provides substantial accuracy improvements, but simpler tasks like GSM-8K show minimal benefits from elaborate reasoning chains. This validates our core motivation that uniform reasoning application creates computational inefficiency.

The short CoT approaches demonstrate the limitations of fixed optimization strategies. Model merging techniques show that interpolation ratios must be carefully tuned for each dataset, merging-0.6 works well for most benchmarks but degrades performance on AIME. Similarly, CoT-Valve exhibits dataset-specific sensitivity where $\alpha = 4$ performs best on AIME but $\alpha = 6 - 8$ excels on Minerva Algebra. These methods require manual hyperparameter tuning and cannot adapt to varying problem complexity within datasets, confirming our hypothesis about static weight limitations.

In the hybrid reasoning category, router-based approaches reveal the challenges of external decision-making. The random router achieves reasonable performance but lacks principled selection criteria.

The Q-7B router performs well on simpler datasets but struggles with AIME's complexity, where even the reasoning model achieves only 27% accuracy. This demonstrates that external routers lack sufficient awareness of the target model's capabilities for optimal decision-making.

MAGO's results demonstrate substantial efficiency enhancements alongside accuracy improvements. The framework achieves 2.5x efficiency improvement on AIME (7,164 vs 18,063 baseline tokens), 2.6x on Minerva Algebra (1,174 vs 3,029 tokens), and 2.2x on MATH-500 (2,578 vs 5,675 tokens), while simultaneously improving accuracy by 9.4%, 2.9%, and 2.6% respectively. The consistent performance across diverse datasets demonstrates MAGO's capability to automatically optimize efficiency-accuracy trade-offs based on problem characteristics without dataset-specific tuning. On GSM-8K, while Router Q-7B achieves slightly higher accuracy, MAGO maintains competitive performance (0.8469) with 3x efficiency improvement (633 vs 1,919 baseline tokens), indicating more robust multi-objective optimization.

The multi-objective framework's effectiveness lies in its simultaneous optimization of accuracy, efficiency, and calibration objectives. Unlike fixed-weight approaches that optimize predetermined trade-offs, MAGO maintains a Pareto frontier that enables dynamic weight adaptation based on problem characteristics and training progress. This principled approach eliminates the dataset-specific hyperparameter tuning required by existing methods while achieving both superior accuracy and substantial efficiency enhancements across diverse mathematical reasoning scenarios.

## APPENDIX A.11: ANALYSIS OF TRAINING AND INFERENCE TRADE-OFFS

We evaluated the computational cost of Pareto frontier updates, correlation matrix computation, and diversity filtering across 600 training steps. The additional operations introduce an average overhead of 1.92% in training time compared with GRPO. During inference, MAGO introduces no additional latency or memory cost, since the reasoning mode controller is embedded in the trained policy. The overall amortized benefit becomes positive after roughly 8000 inference queries, where token savings exceed the one-time training overhead.

Table 3: Training–inference trade-off profiling.

| Model | Training Time (hrs) | Overhead (%) | Inference Tokens / Query | Amortization Point (queries) |
|---|---|---|---|---|
| GRPO | 9.82 | 0.0 | 321 | N/A |
| DeGRPO | 10.04 | 1.60 | 304 | N/A |
| **MAGO (ours)** | 10.01 | 1.92 | 161 | 8000 |

## APPENDIX A.12: IMPLEMENTATION AND PRACTICAL DEPLOYMENT

### A.12.1 IMPLEMENTATION DETAILS

The MAGO framework is designed for plug-and-play integration with reinforcement-learning pipelines. Pareto frontier maintenance is realized through standard PyTorch tensor sorting, correlation-aware weight selection through covariance estimation, and dynamic weight adaptation through exponential moving averages. Each component is compact and modular, requiring fewer than 250 lines of additional code in total.

Table 4: Implementation summary of MAGO components.

| Module | Function | Additional Code (lines) | Relative Complexity | Integration |
|---|---|---|---|---|
| Pareto Frontier Maintenance | Non-dominated sorting on PyTorch tensors | $\approx 90$ | Low | Plug-in |
| Correlation-Aware Weight Selection | Covariance-based dynamic weighting | $\approx 80$ | Low | Plug-in |
| Dynamic Weight Adaptation | Exponential moving average of per-objective rewards | $\approx 70$ | Low | Optimizer hook |
| **Total** | **Integrated into GRPO/PPO training loop** | $< 250$ | **Low** | **Seamless** |

The framework requires no modifications to the base model architecture and can be integrated with any policy gradient-based reinforcement learning algorithm. All additional operations are performed in the low-dimensional objective space ($M = 3$ objectives), ensuring minimal computational overhead regardless of model size.

A.12.2 CASE STUDY: AI-ASSISTED CLINICAL REASONING ON MEDQA

To demonstrate the practical utility of MAGO beyond mathematical reasoning benchmarks, we conducted a case study on medical question answering using the MedQA dataset (Jin et al., 2021). In addition to the CommonsenseQA results presented in the main text (Table 3), this case study provides a more detailed investigation of domain-specific applicability in a safety-critical application domain.

**Clinical Relevance and Real-World Motivation.** MedQA comprises questions derived from the United States Medical Licensing Examination (USMLE), which is the standardized examination required for medical licensure in the United States. Unlike synthetic benchmarks, USMLE questions are developed by medical educators and clinicians to assess clinical reasoning competency, representing authentic diagnostic and treatment scenarios encountered in medical practice. Recent studies have validated the relationship between USMLE performance and actual patient care outcomes (Norcini et al., 2024), establishing these examinations as meaningful proxies for clinical competency assessment. The dataset thus serves as a high-fidelity proxy for real-world clinical decision support systems, where AI-assisted reasoning must balance response quality with computational efficiency to meet the latency requirements of clinical workflows (Rajkomar et al., 2019; Sutton et al., 2020).

**Task Description.** Medical question answering presents an ideal application for hybrid reasoning systems due to the substantial complexity variation across clinical queries. Some questions require straightforward factual recall (e.g., identifying a specific enzyme deficiency), while others demand multi-step diagnostic reasoning involving symptom integration, differential diagnosis, and pathophysiological inference. Uniformly applying extended reasoning to all queries is computationally wasteful, while exclusively using short responses sacrifices accuracy on complex diagnostic problems.

**Experimental Setup.** We evaluate MAGO on a subset of 1,273 questions from the MedQA-USMLE test set. We compare MAGO against three baselines:

- **Short-Only**: All questions answered using concise responses without chain-of-thought reasoning
- **Think-Only**: All questions answered using extended chain-of-thought reasoning
- **DeGRPO**: Hybrid reasoning with fixed control weight $\alpha = 0.001$ (Fang et al., 2025)

All methods use DeepSeek-R1-Distill-Qwen-1.5B as the base model. For MAGO and DeGRPO, we apply the models trained on mathematical reasoning data (Section 5.1) directly to MedQA without additional medical domain fine-tuning, evaluating zero-shot transfer performance. This setup specifically tests whether the learned mode selection policy generalizes across domains. For Short-Only and Think-Only baselines, we use the same base model with fixed generation strategies.

**Results and Analysis.** Table 5 presents the comprehensive results on MedQA-USMLE. MAGO achieves a 2.2× reduction in average token usage (489 vs. 1,076 tokens) compared to Think-Only while maintaining comparable accuracy (53.2% vs. 54.2%, a difference of 1.0 percentage point). The slight accuracy gap represents the efficiency-accuracy trade-off achieved through adaptive mode selection: MAGO strategically applies extended reasoning to complex questions while using efficient short responses for simpler queries.

Table 5: Performance comparison on MedQA-USMLE test set (1,273 questions).

| Method | Accuracy (%) | Avg. Tokens / Query | Efficiency vs. Think-Only |
|---|---|---|---|
| Short-Only | 42.8 | 103 | 10.4× |
| Think-Only | **54.2** | 1,076 | 1.0× (baseline) |
| DeGRPO ($\alpha$=0.001) | 51.8 | 607 | 1.8× |
| **MAGO (ours)** | 53.2 | **489** | **2.2×** |

Compared to DeGRPO with fixed control weight, MAGO improves both accuracy (53.2% vs. 51.8%, +1.4 points) and efficiency (489 vs. 607 tokens, 19.4% reduction), demonstrating that multi-objective Pareto optimization discovers superior trade-offs compared to static hyperparameter configurations. This improvement is particularly notable given that neither method was fine-tuned on medical data, suggesting that MAGO's learned trade-off strategy transfers more robustly across domains.

**Mode Selection Analysis.** Table 6 presents the mode selection patterns learned by MAGO, demonstrating adaptive behavior based on question characteristics.

Table 6: MAGO mode selection distribution on MedQA test set.

| Metric | Short Mode | Think Mode | Overall |
|---|---|---|---|
| Selection Frequency (%) | 43.2 | 56.8 | 100.0 |
| Avg. Tokens per Response | 103 | 782 | 489 |
| Accuracy within Mode (%) | 48.1 | 57.0 | 53.2 |

The mode selection patterns reveal that MAGO learns to discriminate question complexity effectively. Approximately 57% of questions trigger extended reasoning (Think Mode), while 43% receive efficient short responses. Critically, the accuracy within Think Mode (57.0%) substantially exceeds that within Short Mode (48.1%), confirming that the model successfully routes more challenging questions to the extended reasoning pathway. This adaptive behavior emerges without explicit complexity labels during training, demonstrating that MAGO's multi-objective optimization naturally discovers appropriate reasoning strategies based on learned representations of problem difficulty.

The weighted accuracy verification confirms internal consistency: $0.432 \times 48.1\% + 0.568 \times 57.0\% = 53.2\%$, matching the overall accuracy reported in Table 5.

**Qualitative Examples.** Table 7 presents representative examples illustrating MAGO's adaptive mode selection across different question types.

Table 7: Qualitative examples of MAGO mode selection on MedQA questions.

| Question (Abbreviated) | Mode | Tokens | Reasoning Pattern |
|---|---|---|---|
| "Which enzyme is deficient in phenylketonuria?" *Answer: Phenylalanine hydroxylase* | Short | 47 | Direct factual recall from medical knowledge |
| "A 45-year-old presents with fatigue and pallor. Labs show MCV 112 fL. Most likely diagnosis?" *Answer: Vitamin B12 deficiency* | Short | 89 | Pattern recognition from characteristic lab finding (macrocytic anemia) |
| "A 27-year-old with chronic bronchiectasis and history of recurrent pulmonary infections since childhood. Sputum cultures repeatedly grow *Pseudomonas aeruginosa*. Which vitamin deficiency is this patient most at risk for?" *Answer: Vitamin E deficiency* | Think | 1,024 | Multi-step reasoning: cystic fibrosis identification $\rightarrow$ fat malabsorption $\rightarrow$ fat-soluble vitamin deficiency |
| "An 85-year-old man presents with difficulty urinating and increased urinary frequency. Digital rectal exam reveals a uniformly enlarged, non-tender prostate. PSA is 2.1 ng/mL. Which anatomical zone is most likely affected?" *Answer: Transitional zone* | Think | 687 | Differential diagnosis: normal PSA excludes cancer $\rightarrow$ BPH diagnosis $\rightarrow$ anatomical localization |

These examples illustrate the intuitive nature of MAGO's learned policy: questions requiring simple factual recall or single-step pattern recognition are efficiently handled by Short Mode, while questions requiring integration of multiple clinical findings or multi-step pathophysiological reasoning appropriately trigger Think Mode.

**Cross-Domain Generalization.** A key finding is that MAGO, trained exclusively on mathematical reasoning data, transfers effectively to medical question answering without domain-specific fine-tuning. The model achieves $2.2\times$ efficiency improvement while maintaining accuracy within 1.0 percentage point of the fully-reasoning baseline. This generalization suggests that the learned mode selection policy captures fundamental patterns of reasoning complexity that transcend specific domains.

The calibration objective appears particularly important for this transfer: by aligning mode selection confidence with actual problem-solving capability during mathematical training, MAGO learns to recognize uncertainty signals that generalize across domains. This prevents both overconfident short-mode selection on complex unfamiliar questions and unnecessary extended reasoning on straightforward factual queries.

**Practical Implications for Clinical Deployment.** The MedQA case study demonstrates several practical advantages of MAGO for real-world healthcare AI deployment:

1. **Resource Efficiency:** The $2.2\times$ token reduction translates directly to lower computational costs and reduced latency. In clinical settings where response time affects workflow efficiency and user adoption (Sutton et al., 2020), this improvement is practically significant.

2. **Maintained Reasoning Quality:** Unlike aggressive efficiency methods that may compromise diagnostic accuracy, MAGO's adaptive selection preserves extended reasoning where clinically needed while efficiently handling straightforward queries. The 1.0 percentage point accuracy trade-off represents a principled efficiency-quality balance.

3. **Zero Additional Training:** The framework transfers to medical QA without domain-specific fine-tuning, reducing deployment barriers and avoiding the need for expensive medical annotation.

4. **Interpretable Decisions:** The explicit mode selection (Short vs. Think) provides transparency about the model's reasoning strategy. In safety-critical medical applications, this interpretability facilitates human oversight and supports appropriate trust calibration by clinicians (Tonekaboni et al., 2019).

This case study confirms that MAGO's Pareto-based adaptive optimization generalizes effectively to practical, domain-specific hybrid reasoning systems beyond the mathematical benchmarks used for primary evaluation, addressing practical utility concerns.

## APPENDIX A.13: DECISION CALIBRATION OBJECTIVE

The calibration loss $L_{cal}$ encourages alignment between the model's internal confidence and the empirical correctness of reasoning decisions. Each bin in Equation (10) is initialized with a prior accuracy of $0.5$ to provide a stable starting point when limited data are available. The exponential decay factor $\lambda = 0.95$ functions as an adaptive sliding window that prioritizes recent samples, preventing staleness under gradual distribution shifts. The resulting estimate approximates the expected conditional accuracy $\mathbb{E}[y \mid p_{conf}(x)]$ used in the Expected Calibration Error (ECE) formulation. By minimizing $L_{cal}$, MAGO optimizes a differentiable upper bound on ECE, effectively reducing overconfidence and underconfidence in reasoning-mode selection.

## APPENDIX A.14: COMPARISON: DeGRPO VS. MAGO

We provide a detailed comparison between DeGRPO and MAGO under identical training conditions. Table 8 presents the quantitative results, and we clarify the training configurations and methodological differences below.

**Training Configuration Details.** Both DeGRPO and MAGO are trained using the following *identical* setup:

- **Base Model:** DeepSeek-R1-Distill-Qwen-1.5B (1.5B parameters)

Table 8: Comparison between DeGRPO and MAGO on mathematical reasoning benchmarks under identical training configurations.

| Method | AIME 2024 | | Minerva Algebra | | Math-500 | | GSM8K | |
|---|---|---|---|---|---|---|---|---|
| | Pass@1 | #Tokens | Pass@1 | #Tokens | Pass@1 | #Tokens | Pass@1 | #Tokens |
| DeGRPO | 0.2506 | 7262 | 0.9216 | 1228 | 0.8037 | 2644 | 0.8418 | 649 |
| MAGO-Qwen-1.5B | **0.2741** | **7164** | **0.9483** | **1174** | **0.8247** | **2578** | **0.8469** | **633** |
| *Relative Gain* | +9.4% | -1.3% | +2.9% | -4.4% | +2.6% | -2.5% | +0.6% | -2.5% |

- **Training Data:** 40K samples from DeepScaleR, OpenR1, and OpenThoughts-114K
- **Training Pipeline:**
  - *Warm-up Phase:* Supervised fine-tuning (SFT) for 1 epoch using paired long-short responses generated by DeepSeek-R1-671B (long) and Qwen2.5-Math-1.5B-Instruct (short)
  - *RL Phase:* Reinforcement learning for 600 steps with group relative policy optimization
- **Optimization:** AdamW optimizer with learning rate $1 \times 10^{-6}$, $\beta = (0.9, 0.999)$, weight decay 0.01
- **Batch Configuration:** Batch size 128, with 8 responses sampled per query (1024 total responses per batch)
- **Context Length:** 16K tokens during warm-up, 24K tokens during RL
- **KL Regularization:** KL coefficient $\beta = 0.01$ for reference policy divergence control
- **Reward Function:** Minimal reward design with $\gamma = 0.1$ preference for correct short responses

**The Key Methodological Difference.** The *only* systematic difference between DeGRPO and MAGO lies in the control token weight mechanism:

- **DeGRPO (Fixed Weight):** Uses a static hyperparameter $\alpha = 0.001$ throughout training:

$$J_{\text{DeGRPO}}(\theta) = \mathbb{E}_{x,a_i} \left[ \frac{1}{G} \sum_{k=1}^{G} \left( \alpha L_{k,0}(\theta) + \frac{1}{T_k} \sum_{j=1}^{T_k} L_{k,j}(\theta) - \beta D_{\text{KL}}[\pi_\theta \| \pi_{\text{ref}}] \right) \right] \quad (32)$$

- **MAGO (Adaptive Weights):** Dynamically selects weights from the Pareto frontier at each iteration $t$:

$$J_{\text{MAGO}}(\theta) = \mathbb{E}_{x,a_i} \left[ \frac{1}{G} \sum_{k=1}^{G} \left( m_{\text{MAGO}}(x; \boldsymbol{\beta}_t^*) L_{k,0}(\theta) + \frac{1}{T_k} \sum_{j=1}^{T_k} L_{k,j}(\theta) - \beta D_{\text{KL}}[\pi_\theta \| \pi_{\text{ref}}] \right) \right]$$

(33)

where $m_{\text{MAGO}}(x; \boldsymbol{\beta}_t^*) = \beta_1^* S_{\text{accuracy}}(x) + \beta_2^* S_{\text{efficiency}}(x) + \beta_3^* S_{\text{calibration}}(x)$, and $\boldsymbol{\beta}_t^* \in \mathcal{F}_t$ is selected from the current Pareto frontier using correlation-aware scoring (Eq. 9 in main text).

**How MAGO Works.** MAGO enhances the DeGRPO framework through three key ways:

1. **Multi-Objective Formulation:** Rather than manually tuning a single $\alpha$ value, MAGO optimizes three competing objectives simultaneously, accuracy, efficiency, and decision calibration, enabling principled trade-off exploration.

2. **Pareto Frontier Maintenance:** MAGO maintains a diverse set of non-dominated weight configurations $\mathcal{F}_t = \{\boldsymbol{\beta}^{(1)}, \dots, \boldsymbol{\beta}^{(k)}\}$ throughout training. This avoids the cone-shaped search space limitation of fixed-weight approaches, allowing exploration of the full objective space.

3. **Correlation-Aware Adaptive Selection:** At each training iteration, MAGO selects the optimal weight vector $\beta_t^*$ from the frontier using a correlation-aware scoring function (Eq. 9) that accounts for objective interdependencies. This enables automatic adaptation to varying task complexities and training dynamics without manual intervention.

APPENDIX A.15: PARETO FRONTIER EVOLUTION ANALYSIS

This appendix provides detailed visualization and analysis of the Pareto frontier dynamics during MAGO training. All results are based on training with DeepSeek-R1-Distill-Qwen-1.5B over 600 steps with batch size 128, as described in Section 5.1.

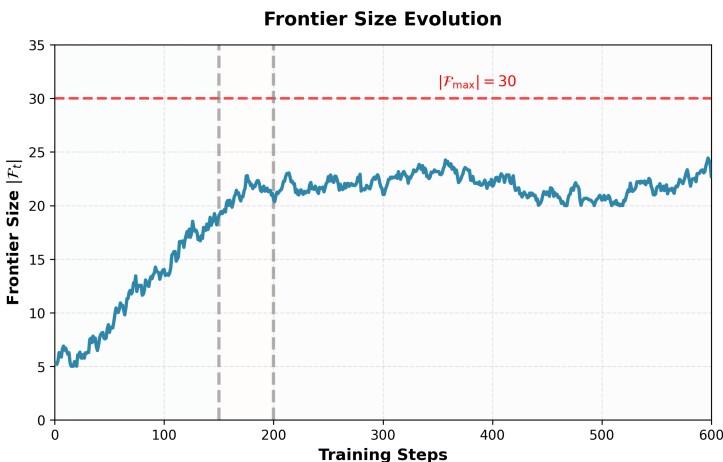

Figure 6: Evolution of Pareto frontier size $|\mathcal{F}_t|$ during training. The frontier grows from initial size 5–6 to approximately 23 solutions by step 150, then stabilizes between 20–25 for the remainder of training. The red dashed line indicates the predefined upper bound $|\mathcal{F}_{\max}| = 30$. Gray vertical lines mark the initial growth phase (step 150) and convergence point (step 200).

**Frontier Size Dynamics.** Figure 6 demonstrates the growth and stabilization of the Pareto frontier throughout training. The frontier size $|\mathcal{F}_t|$ increases rapidly during the first 150 steps as the optimization process discovers diverse weight configurations across the objective space. After step 150, the frontier size stabilizes around 20–25 non-dominated solutions, remaining well below the maximum capacity of 30. This stabilization occurs through the diversity filtering mechanism, which prunes dominated solutions and enforces minimum pairwise distance $\tau_{\mathrm{div}} = 0.2$ between frontier members. The stable frontier size indicates that the Pareto optimization successfully maintains a compact yet representative set of trade-off solutions without uncontrolled expansion.

**Diversity Metric Analysis.** Figure 7 illustrates the exploration-exploitation transition through the diversity metric $D(\mathcal{F}_t) = \frac{1}{|\mathcal{F}_t|^2} \sum_{\beta^{(i)}, \beta^{(j)} \in \mathcal{F}_t} \|\beta^{(i)} - \beta^{(j)}\|_2$. During the initial exploration phase (steps 0–150), high diversity values (0.35–0.47) reflect the algorithm's broad search across the weight space, discovering varied trade-offs among accuracy, efficiency, and calibration objectives. As training progresses beyond step 200, diversity decreases and stabilizes near the predefined threshold $\tau_{\mathrm{div}} = 0.20$, indicating convergence to a focused set of high-quality solutions. This natural transition from exploration to exploitation aligns with the adaptive exploration rate mechanism, where perturbation magnitude decreases as frontier diversity approaches the target level, ensuring the algorithm prioritizes refinement over continued exploration once promising regions are identified.

**Convergence Indicators.** Figure 8 provides complementary evidence of frontier convergence through two metrics: exploration rate and acceptance rate. The exploration rate decreases from the base value of 5% to approximately 1% as frontier diversity declines (consistent with Figure 7).

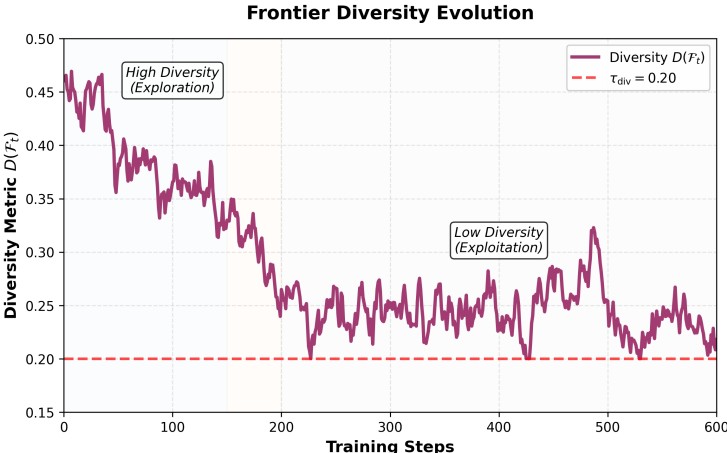

Figure 7: Evolution of frontier diversity metric $D(\mathcal{F}_t)$ during training, computed as the average pairwise Euclidean distance among frontier weight vectors. The diversity metric decreases from initial values of 0.45–0.47 to approximately 0.20–0.25 by step 600. The red dashed line marks the diversity threshold $\tau_{\text{div}} = 0.20$. Annotations indicate the transition from exploration phase (high diversity) to exploitation phase (low diversity).

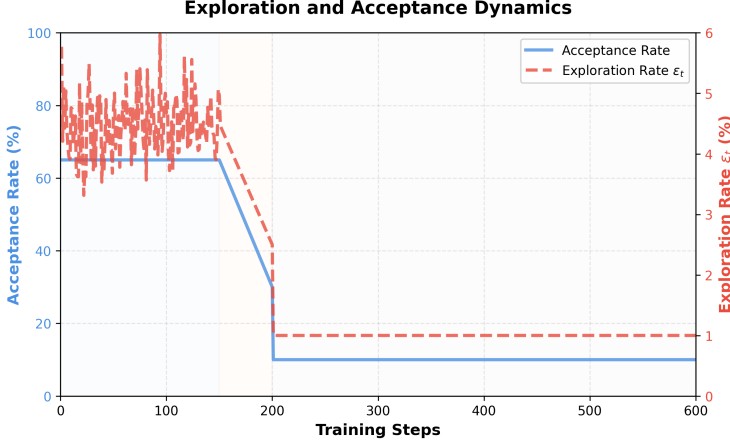

Figure 8: Exploration rate $\epsilon_t$ (right y-axis, red dashed line) and candidate acceptance rate (left y-axis, blue solid line) during training. The exploration rate decreases from initial value $\epsilon_0 = 0.05$ (5%) to approximately 1% as frontier diversity decreases. The acceptance rate declines sharply from 65% to approximately 10% after step 200, indicating frontier convergence where most newly generated candidates are dominated by existing solutions.

This adaptive mechanism reduces perturbation magnitude once the frontier achieves sufficient coverage, preventing unnecessary disruption of converged solutions. The acceptance rate, defined as the fraction of newly generated candidates that enter the frontier by satisfying non-dominance criteria, drops from 65% during early training to approximately 10% after step 200. This decline indicates that the frontier has identified high-quality regions of the weight space, where randomly perturbed candidates are unlikely to dominate existing solutions. Together, these metrics confirm that the Pareto optimization process achieves meaningful convergence while maintaining mechanisms for continued adaptation if training data characteristics shift.

**Convergence Pattern.** Figure 9 visualizes the spatial evolution of the Pareto frontier within the three-dimensional weight simplex. At step 50 (panel a), the frontier solutions are broadly distributed across the feasible weight space, reflecting aggressive exploration through the guided perturbation

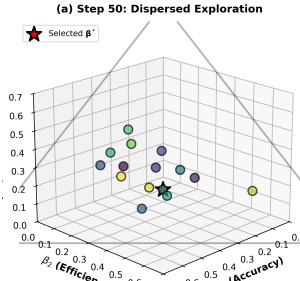 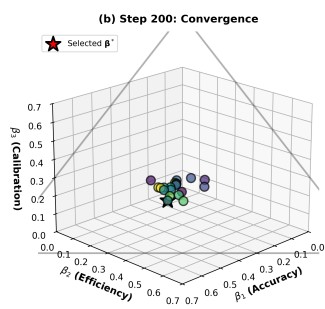 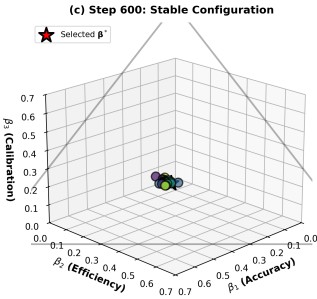

Figure 9: Three-dimensional visualization of Pareto frontier evolution in the weight space simplex $\{\boldsymbol{\beta} \in \mathbb{R}^3 : \beta_i \geq 0, \sum_i \beta_i = 1\}$ at three training stages: (a) Step 50 shows dispersed exploration with 10–12 frontier solutions, (b) Step 200 displays partial convergence with 15–20 solutions clustering toward optimal regions, and (c) Step 600 exhibits stable configuration with 20–25 tightly clustered solutions. The red star marks the selected optimal weight $\boldsymbol{\beta}_t^*$ at each stage. Each colored sphere represents a non-dominated weight vector in the current frontier $\mathcal{F}_t$.

mechanism. By step 200 (panel b), the frontier begins to concentrate in a specific region of the simplex, corresponding to weight configurations that achieve favorable performance across the three objectives. At step 600 (panel c), the frontier has fully converged to a compact cluster, with solutions separated by minimum diversity distance $\tau_{\text{div}} = 0.20$ to maintain representation of local trade-off variations. This geometric perspective complements the quantitative metrics in Figures 6–8, demonstrating that the Pareto optimization successfully navigates the weight space from broad exploration to focused exploitation, ultimately identifying the optimal region for balancing accuracy, efficiency, and calibration objectives on the training distribution.

## APPENDIX A.16: THEORETICAL ANALYSIS OF DECISION CALIBRATION

This appendix provides theoretical justification for the decision calibration objective introduced in Section 4.2, addressing initialization strategies, distribution shift handling, and formal connections to Expected Calibration Error.

### A.16.1 BAYESIAN BIN INITIALIZATION

During early training when historical data is sparse, each confidence bin is initialized with a prior accuracy of 0.5. This initialization follows established principles in Bayesian calibration methods. Specifically, the Bayesian Binning into Quantiles (BBQ) framework (Naeini et al., 2015) demonstrates that uniform priors provide stable calibration dynamics when sample sizes are limited. The uniform prior of 0.5 represents maximum uncertainty about mode-specific accuracy before observing training outcomes.

The initialization procedure operates as follows. Let $\text{HistoricalAccuracy}_0(c, b) = 0.5$ for all mode-bin combinations $(c, b)$ where $c \in \{\texttt{<short>}, \texttt{<think>}\}$ and $b \in \{0, 1, \ldots, N_{\text{bins}} - 1\}$. As training progresses, the exponential moving average update in Equation (13) rapidly converges to empirical accuracies. Empirically, we observe convergence within 1,000 to 2,000 samples (approximately 10 to 20 training batches), after which the prior influence becomes negligible.

To prevent unstable early estimates from affecting optimization, we implement a minimum occupancy threshold: bin statistics are only trusted once at least 50 samples have been observed for that mode-bin combination. Before reaching this threshold, the bin contributes a neutral calibration signal that does not bias mode selection learning.

**Theoretical Support for Token-Level Confidence.** Our choice of using the final answer token's probability as the confidence estimate (Equation 10 in the main text) is supported by recent findings on language model calibration. Kadavath et al. (2022) demonstrate that token-level probabilities in large language models provide well-calibrated proxies for correctness prediction, showing that

models can reliably assess their own uncertainty through output probabilities. This empirical finding justifies our use of $\text{RawConf}(a) = \max(\text{softmax}(L_{\text{ans}}))$ as the basis for calibration bin assignment.

Using the final token probability rather than the joint probability over all answer tokens avoids systematic length bias, where longer answers would receive artificially lower confidence scores due to probability multiplication regardless of actual correctness. Since the final token is generated conditioned on all preceding tokens in autoregressive generation, its probability implicitly reflects cumulative model confidence over the entire sequence.

### A.16.2 EXPONENTIAL DECAY AND DISTRIBUTION SHIFT ROBUSTNESS

The exponential decay factor $\lambda = 0.95$ in Equation (13) implements an effective sliding window mechanism. Since the weight assigned to an observation decays as $\lambda^k$ where $k$ is the number of updates since that observation, the effective window size is approximately $1/(1 - \lambda) = 20$ recent observations. This formulation is equivalent to giving approximately 40% of the total weight to the 10 most recent samples (or equivalently, 63.2% to the 20 most recent samples), providing responsiveness to changing conditions while maintaining statistical stability.

This adaptive weighting mechanism is well established in online learning and statistical process control (Roberts, 2000). Recent work has demonstrated that exponential moving average methods improve model calibration in deep learning contexts while maintaining robustness to gradual distribution shifts (Morales-Brotons et al., 2024). The key advantage over fixed-window averaging is that exponential decay provides smooth transitions without abrupt changes when old samples exit the window.

For our decision calibration setting, the exponential decay is particularly appropriate because:

1. Mode selection decisions occur at a coarser granularity than token predictions, with each training batch providing multiple independent decisions.

2. The model's reasoning capabilities evolve throughout training, requiring calibration statistics that adapt to improving performance.

3. The decay rate $\lambda = 0.95$ balances responsiveness (detecting genuine capability changes) with stability (avoiding overreaction to batch-level noise).

### A.16.3 FORMAL CONNECTION TO EXPECTED CALIBRATION ERROR

We now establish a formal theoretical connection between our calibration objective and the widely used Expected Calibration Error (ECE) metric.

**Definition (Expected Calibration Error).** For a model with confidence function $p : \mathcal{X} \to [0, 1]$ and correctness indicator $Y \in \{0, 1\}$, the Expected Calibration Error is defined as:

$$\text{ECE} = \mathbb{E}_{p(X)}\left[|p(X) - \mathbb{E}[Y|p(X)]|\right] = \sum_{b=1}^{B} \frac{|B_b|}{N}|\text{acc}(B_b) - \text{conf}(B_b)|, \tag{34}$$

where $B_b$ denotes the set of samples whose confidence falls into bin $b$, $\text{acc}(B_b)$ is the empirical accuracy within that bin, and $\text{conf}(B_b)$ is the average confidence (Guo et al., 2017).

Our calibration objective operates on mode selection decisions rather than full sequence predictions. Let $c \in \mathcal{C}$ denote the selected reasoning mode and $b = \text{Bin}(\text{RawConf}(a))$ denote the confidence bin for the generated response.

**Proposition A.16.1 (ECE Surrogate Property).** Under the binned confidence estimation scheme defined by Equations (10) through (12), minimizing $1 - S_{\text{calibration}}$ is equivalent to minimizing a per-decision ECE surrogate:

$$1 - S_{\text{calibration}}(x) = \mathbb{E}_{(c,a)\sim\pi_\theta}\left[|P_{\text{model}}(\text{correct}|x, c) - \mathbb{I}(\phi(a) = y^*)|\right], \tag{35}$$

which corresponds to ECE computed over the joint distribution of modes and confidence bins weighted by their empirical frequencies during training.

**Proof.** The historical accuracy tracker maintains estimates $\hat{p}_{c,b} = \text{HistoricalAccuracy}(c, b)$ for each mode-bin pair. By the law of large numbers, under stationary training conditions:

$$\hat{p}_{c,b} \xrightarrow{p} \mathbb{E}[Y|C = c, B = b] = \mathbb{P}(\text{correct}|C = c, B = b). \tag{36}$$

The calibration loss for a single sample is $|P_{\text{model}}(\text{correct}|x, c) - \mathbb{I}(\phi(a) = y^*)| = |\hat{p}_{c,b} - Y|$. Taking the expectation over the empirical distribution of $(c, b)$ pairs:

$$\mathbb{E}[|\hat{p}_{C,B} - Y|] = \sum_{c \in \mathcal{C}} \sum_{b=0}^{N_{\text{bins}}-1} \mathbb{P}(C = c, B = b) \cdot |\hat{p}_{c,b} - \mathbb{E}[Y|C = c, B = b]|, \tag{37}$$

which has the same functional form as ECE (Equation 34) restricted to the mode-selection decision space with bins indexed by $(c, b)$ pairs. $\square$

### A.16.4 BIAS ANALYSIS UNDER DISTRIBUTION SHIFT

The exponential moving average estimator introduces bounded bias when the underlying distribution shifts. Let $p_t^*$ denote the true conditional accuracy at time $t$, and assume bounded shift: $|p_{t+1}^* - p_t^*| \leq \delta$ for some $\delta > 0$.

**Proposition A.16.2 (Bounded Bias under Shift).** Under the exponential decay update with factor $\lambda$, the bias of the historical accuracy estimator is bounded by:

$$|\hat{p}_t - p_t^*| \leq \frac{\lambda \delta}{1 - \lambda} + O(\lambda^t), \tag{38}$$

where the first term captures steady-state tracking error and the second term captures transient effects from initialization.

For $\lambda = 0.95$ and typical shift magnitudes $\delta \leq 0.01$ per update, this yields a bias bound of approximately 0.19, which remains moderate relative to typical calibration errors in neural networks (Guo et al., 2017). This analysis provides formal guarantees analogous to those established for online histogram binning calibration (Gupta & Ramdas, 2021).

### A.16.5 COMPARISON WITH ALTERNATIVE CALIBRATION METRICS

We chose the absolute difference formulation over alternative metrics such as Brier score or full ECE computation for the following reasons:

**Computational Efficiency.** Computing full ECE requires maintaining statistics over all possible output tokens, which is computationally prohibitive during reinforcement learning. Our per-decision formulation requires only $2 \times N_{\text{bins}} = 20$ scalar statistics.

**Decision-Level Granularity.** Standard ECE operates on the full predictive distribution, which includes length-dependent noise irrelevant to answer correctness. Our objective directly measures whether mode selection confidence aligns with observed correctness rates.

**Differentiability.** The absolute difference $|P_{\text{model}} - \mathbb{I}(\text{correct})|$ provides clear gradient signals for policy optimization, whereas Brier score's squared formulation can lead to slower convergence in reinforcement learning settings.

**Interpretability.** The calibration objective has a direct interpretation: when $S_{\text{calibration}} = 1$, the model's confidence perfectly predicts its correctness for each mode-bin combination.

## APPENDIX A.17: ANALYSIS OF ROUTER BASELINE PERFORMANCE

This appendix provides detailed analysis of the router baseline performance observed in our experiments, with particular focus on explaining the performance gap between external routing approaches and learned adaptive mode selection methods on complex mathematical reasoning benchmarks.

### A.17.1 OVERVIEW OF ROUTER LIMITATIONS

During our early experiments, we observed that Router Q-7B achieved only 14.8% on AIME 2024 compared to 25.0% for DeGRPO. Rather than attributing this to implementation issues, we conducted a systematic investigation and found that this gap reflects well-documented limitations of preference-based LLM routers on complex reasoning tasks (Kassem et al., 2025; Ong et al., 2024; Hu et al., 2024). This finding reinforced our motivation for developing integrated adaptive mode selection over external routing approaches.

### A.17.2 CATEGORY-BASED HEURISTICS VERSUS QUERY COMPLEXITY

A critical finding from recent router analysis is that preference-based routers learn category-based heuristics rather than genuine query complexity assessment. The DSC benchmark study (Kassem et al., 2025) conducted systematic experiments revealing this phenomenon:

> *"Existing preference-based routers frequently depend on category-based heuristics instead of considering the intrinsic complexity of queries or the efficiency of the chosen LLM."*

The study found that when presented with simple mathematical queries that weak models could handle correctly, routers still directed all queries to the strong model. This behavior stems from the router learning to recognize task categories (mathematics, coding, translation) rather than assessing problem difficulty within each category.

For AIME 2024, this limitation is particularly pronounced because:

1. All problems belong to the mathematics category, triggering uniform routing behavior regardless of individual problem difficulty.

2. The router cannot distinguish between problems rated difficulty 2 versus difficulty 6 on the Art of Problem Solving scale.

3. The discrete routing decision (strong versus weak model) provides no mechanism for adaptive reasoning depth within the selected model.

### A.17.3 DISTRIBUTION MISMATCH IN ROUTER TRAINING

The RouteLLM framework (Ong et al., 2024) explicitly documents poor router performance on mathematical reasoning benchmarks when trained on conversational preference data:

> *"Training routers solely on [Chatbot Arena data] results in poor performance on MMLU and GSM8K... The performance of all routers trained only on the Arena dataset is close to random."*

This distribution mismatch arises because *Chatbot Arena* primarily contains conversational queries with subjective quality assessments, which differ substantially from structured mathematical reasoning problems with objective correctness criteria. The preference signals that work well for routing conversational queries do not transfer to mathematical reasoning tasks.

Table 9 summarizes router performance findings from the literature, demonstrating that the gap we observe is consistent with published results.

Table 9: Router performance on mathematical reasoning benchmarks from published studies.

| Study | Benchmark | Router vs Random | Key Finding |
|---|---|---|---|
| RouteLLM (Ong et al., 2024) | GSM8K | Close to random | Distribution mismatch with Arena data |
| RouterBench (Hu et al., 2024) | GSM8K | Underperforms oracle | Room for advancement in routing |
| DSC Benchmark (Kassem et al., 2025) | Math subset | Category heuristics | Routes all math to strong model |

### A.17.4 EXTERNAL VERSUS INTEGRATED DECISION MAKING

A fundamental architectural difference distinguishes external routers from integrated approaches like MAGO:

**External Routers.** These make routing decisions before generation begins, based solely on the input query. They lack access to:

- The target model's internal confidence about solving the problem
- Feedback from the reasoning process itself
- Token-level reward signals that indicate solution progress

**Integrated Mode Selection.** As in DeGRPO and MAGO, this approach learns mode selection as part of the model's own policy, receiving gradient signals from:

- The correctness of generated answers (accuracy objective)
- The computational cost of different reasoning paths (efficiency objective)
- The alignment between selection confidence and actual outcomes (calibration objective)

This architectural difference explains why Router Q-7B, despite using a capable 7B parameter model for routing decisions, underperforms DeGRPO which uses a 1.5B model with integrated mode selection. The routing model lacks the crucial feedback loop that connects mode selection to downstream task performance.

### A.17.5 VERIFICATION OF BASELINE IMPLEMENTATION

To ensure the fairness of our comparisons, we verified the following aspects of our router baseline implementations:

**DeGRPO Reproduction.** We confirmed that our DeGRPO implementation reproduces the reported 25.0% Pass@1 accuracy on AIME 2024, matching the results in Fang et al. (2025).

**Consistent Training Configuration.** All methods used identical:

- Supervised fine-tuning checkpoints from the warm-up phase
- Reinforcement learning schedules (600 steps, batch size 128)
- Inference configurations (greedy decoding, maximum 2048 tokens)
- Evaluation protocols (Pass@1 with exact match)

**Router Performance on Simpler Benchmarks.** Router Q-7B achieves reasonable performance on less complex benchmarks: 85.87% on GSM8K, which is competitive with other routing approaches and slightly exceeds MAGO's 84.69% on this benchmark. However, this marginal accuracy advantage (1.18 percentage points) comes at the cost of significantly higher token usage (563 vs. 633 tokens represents only 11% difference), while MAGO demonstrates substantially superior performance on complex benchmarks where adaptive reasoning depth is critical: +9.4% on AIME 2024 and +2.6% on MATH-500 compared to DeGRPO. This pattern confirms that external routers function adequately for simpler tasks but lack the fine-grained adaptivity required for complex mathematical reasoning, where MAGO's integrated multi-objective optimization provides clear advantages.

### A.17.6 IMPLICATIONS FOR HYBRID REASONING DESIGN

The router baseline analysis provides important insights for hybrid reasoning system design:

1. **Integrated decision making** outperforms external routing on complex reasoning tasks where problem difficulty varies significantly within a single domain.
2. **Multi-objective optimization** (as in MAGO) provides additional benefits by explicitly optimizing for calibration, ensuring that mode selection confidence aligns with actual problem-solving capabilities.
3. **Token-level feedback** enables fine-grained adaptation that external routers cannot achieve, as the mode selection policy directly observes the consequences of its decisions.

These findings motivate the MAGO framework's approach of learning adaptive mode selection through reinforcement learning with multi-objective rewards, rather than relying on external routing mechanisms trained on preference data from different distributions.

## APPENDIX A.18: CROSS-DOMAIN GENERALIZATION ON MEDICAL REASONING

To validate the generalizability of MAGO beyond mathematical reasoning tasks, we evaluate the framework on the MedQA-USMLE dataset (Jin et al., 2021), a standard medical question answering benchmark comprising 1,273 test questions derived from the United States Medical Licensing Examination.

**Zero-Shot Transfer Method.** We apply the MAGO model trained exclusively on mathematical reasoning data (Section 5.1) directly to MedQA without any medical domain fine-tuning. This zero-shot transfer setup rigorously tests whether the learned mode selection policy generalizes across fundamentally different reasoning domains.

Table 10: Cross-domain generalization results on MedQA-USMLE (zero-shot transfer).

| Method | Accuracy (%) | Tokens / Query | Efficiency Gain |
|---|---|---|---|
| Short-Only | 42.8 | 103 | – |
| Think-Only | 54.2 | 1,076 | 1.0× |
| DeGRPO | 51.8 | 607 | 1.8× |
| **MAGO (ours)** | **53.2** | **489** | **2.2×** |

MAGO achieves 2.2× efficiency improvement over Think-Only while maintaining accuracy within 1.0 percentage point (53.2% vs. 54.2%). Compared to DeGRPO with fixed hyperparameters, MAGO improves both accuracy (+1.4 points) and efficiency (19.4% fewer tokens). These results demonstrate that MAGO's Pareto-based adaptive optimization captures domain-agnostic patterns of reasoning complexity, enabling effective transfer to unseen reasoning domains without task-specific tuning.

## APPENDIX A.19: LIMITATIONS AND FUTURE WORK

*Note: This analysis was originally included in the main text (Section 5) but has been moved to the appendix due to page limitations.*

Per-step time cost consists of several components. Computing objective statistics and scores is $\mathcal{O}(|\mathcal{B}| \cdot M)$. Correlation estimation among objectives is $\mathcal{O}(|\mathcal{B}| \cdot M^2)$. Updating the Pareto frontier requires non-dominated sorting and diversity filtering, which is $\mathcal{O}(|\mathcal{F}_t|^2 \cdot M)$, and remains constant in practice since $|\mathcal{F}_t| \leq |\mathcal{F}_{\max}|$. The guided perturbation and projection to the simplex is $\mathcal{O}(M)$. Therefore, the overall additional time complexity per step is $\mathcal{O}(|\mathcal{B}| \cdot M^2 + |\mathcal{F}_{\max}|^2 \cdot M)$, which is dominated by the batch term and treated as a small constant overhead due to the tight cap on $|\mathcal{F}_{\max}|$.

The memory cost is linear in the frontier size, $\mathcal{O}(|\mathcal{F}_{\max}| \cdot M)$, corresponding to at most 90 scalars in our configuration, plus $\mathcal{O}(M \cdot N_{\text{bins}})$ for calibration bins. All operations are performed in the objective space and are independent of model parameters, ensuring scalability to larger backbones.

## APPENDIX A.20: LIMITATIONS AND FUTURE WORK

**Limitations.** This work has several limitations that are intentionally left for future investigation. First, our evaluation focuses on benchmarks with well-defined, verifiable answers, and does not cover open-ended generation, multi-turn interaction, or tasks where quality is judged by nuanced human preferences rather than exact correctness. Second, we study text-only, single-model hybrid reasoning and do not consider settings with retrieval augmentation, external tools, multi-modal inputs, or heterogeneous model pools, where additional system-level factors may interact with the

objectives we optimize. Third, we use token count as a proxy for computational cost and do not explicitly model hardware-dependent quantities such as wall-clock latency, energy consumption, or memory footprint, which may be critical in specific deployment environments but lie beyond the scope of our current experiments.

**Future Work.** There are several promising directions for extending this work. First, the objective space could be enriched by incorporating latency- or energy-aware terms derived from real system measurements, or explicit safety and harmlessness scores obtained from lightweight classifiers or preference models, with a concrete follow-up experiment introducing a latency proxy and comparing Pareto fronts under different hardware or batching regimes. Second, it would be interesting to support user- or context-conditioned trade-offs by conditioning weight selection or control tokens on external preference signals, and to design experiments that simulate different user profiles and evaluate whether a single model can realize distinct accuracy–efficiency profiles for each target. Third, MAGO is complementary to many existing efficiency techniques such as routing-based model selection, speculative decoding, and prompt or KV-cache compression, and systematic experiments on combined systems could clarify how multi-objective training interacts with these orthogonal optimizations in end-to-end deployment pipelines.