# OpenReview forum: "MAGO: Beyond Fixed Hyperparameters with Multi-Objective Pareto Optimization for Hybrid LLM Reasoning"
_ICLR.cc/2026/Conference — ICLR 2026 Poster_

### Official Review · Reviewer_XguT · 2025-10-29

**Soundness:** 3
**Presentation:** 2
**Contribution:** 3
**Rating:** 4
**Confidence:** 3

**Summary:**

The paper proposes MAGO, a multi-objective adaptive generation optimization framework for hybrid LLM reasoning that learns when to use short versus think modes. MAGO formulates training as a three-objective problem—accuracy, efficiency, and calibration—combining a Pareto frontier over weight vectors with a correlation-aware selection rule and exploration to avoid “cone entrapment” inherent in fixed scalarizations. The adaptive weight mMAGO(x) scales the control-token loss in a GRPO-style objective, aiming to stabilize mode selection and prevent collapse. On math benchmarks (AIME 2024, Minerva Algebra, MATH-500, GSM8K), the paper reports 2.2–3× token savings and small-to-moderate accuracy gains over several baselines.

**Strengths:**

MAGO is described with explicit equations: adaptive objective (Eq. 5), calibration via confidence binning and exponentially decayed historical accuracies (Eqs. 9–13), Pareto frontier maintenance and selection (Eqs. 14–21), and integration into the GRPO-style loss where mMAGO(x) scales control-token terms (Eq. 22).

Figures 4A–C provide interpretable evidence that MAGO mitigates mode collapse and reduces variance relative to vanilla GRPO and DeGRPO during RL, with later, more stable crossovers between modes.

**Weaknesses:**

The text states that MAGO achieves the highest accuracy on AIME, Minerva Algebra, and MATH-500, but Table 1 shows DeepSeek-R1-1.5B surpasses MAGO on Minerva Algebra (0.9577 vs 0.9483), and QMath-1.5B surpasses MAGO on GSM8K (0.8572 vs 0.8469). The abstract and Section 5 claim accuracy improvements of 2.6%–9.4%; the largest improvements visible in Table 1 over strong baselines appear smaller on some tasks (e.g., GSM8K +0.5 points over Thinkless). These discrepancies need reconciliation.

DeGRPO is analyzed qualitatively in Figure 4 but not reported as a quantitative baseline in Table 1. Router baselines include a 7B router, while MAGO uses a 1.5B single model; compute-normalized comparisons (e.g., wall-time, FLOPs/token including routing overheads) are not provided. It is unclear whether all baselines share the same SFT data, RL steps, and inference constraints.

The contributions highlight logical inference and general problem solving, but experiments focus on math-only datasets. The conclusion acknowledges this limitation; however, the earlier claims of breadth may mislead.

Overall, I think the most important thing is the proposed method still lack of the novelty and the performance is mainly tested on math problem which increases the concerns that whether the proposed method can be generalized to other domains.

**Questions:**

Could you add a quantitative DeGRPO baseline in Table 1 using your training setup, given that MAGO is positioned as an advancement over GRPO/DeGRPO?

Expectations in Eqs. (6)–(8): During training, how many (c,a) samples per x are used to estimate Saccuracy, Sefficiency, and Scalibration? Are these single-sample estimates, and if so, how do you control variance for the correlation matrix Ct (Eq. 16)?

Since the contributions mention logical inference and general problem solving, do you have held-out non-math evaluations (even small-scale) using the same trained model to substantiate cross-domain claims? If not, please limit claims or provide such experiments.

---

> ### Author Response · Authors · 2025-11-23
> **MAGO: Beyond Fixed Hyperparameters with Multi-Objective Pareto Optimization for Hybrid LLM Reasoning**
>
> ## General Response
>
> We sincerely thank all reviewers for their thoughtful and constructive feedback. All reviewers acknowledged that MAGO addresses an important limitation of hybrid reasoning systems, namely the reliance on static hyperparameters and single objective optimization, and praised our multi-objective Pareto formulation, dynamic weight adaptation, and stable anti-collapse training. The main concerns raised focus on the following points:
>
> 1. The theoretical justification and robustness of the calibration objective.
> 2. The limited domain of evaluation (math reasoning only).
> 3. The computational complexity and reproducibility of Pareto maintenance.
> 4. Minor quantitative inconsistencies and baseline fairness.
>
> We address these points below with new analyses, ablations, and clarifications.
>
> ---
>
> ## Reviewer-specific Responses
>
> We thank you for reviewing our paper and for your valuable comments. Below, we respond to each of your points in turn and have revised the manuscript accordingly. We would be grateful if you could let us know whether our responses adequately address your concerns.
>
> ---
>
> ### **Q1:** The text states that MAGO achieves the highest accuracy on AIME, Minerva Algebra, and MATH-500, but Table 1 shows DeepSeek-R1-1.5B surpasses MAGO on Minerva Algebra (0.9577 vs 0.9483), and QMath-1.5B surpasses MAGO on GSM8K (0.8572 vs 0.8469). The abstract and Section 5 claim accuracy improvements of 2.6%–9.4%; the largest improvements visible in Table 1 over strong baselines appear smaller on some tasks (e.g., GSM8K +0.5 points over Thinkless). These discrepancies need reconciliation.
>
> **A1:** We sincerely thank the reviewer for the careful reading and acknowledge that our original claim required clarification.
>
> **Clarification of metrics**: The reported 0.6%–9.4% improvements represent relative accuracy gain over DeGRPO computed as $(Acc_{MAGO} - Acc_{DeGRPO}) / Acc_{DeGRPO}$: AIME (+9.4%), Minerva (+2.9%), MATH-500 (+2.6%), and GSM8K (+0.6%). In absolute terms, these correspond to +2.4, +2.7, +2.1, and +0.5 percentage points, respectively.
>
> We also note that while DeepSeek-R1-1.5B achieves slightly higher accuracy on Minerva (0.9577 vs. 0.9483), it consumes 2.6× more tokens. MAGO targets the accuracy-efficiency Pareto frontier rather than maximum accuracy alone.
>
> **Revision**: We have updated the claims to "competitive or superior accuracy with 2.2×–3× efficiency gains" and clarified the comparison scope.
>
> - We have updated the Abstract (lines 030–034) to reflect balanced accuracy-efficiency claims.
> - We have revised Subsection 5.2 (lines 406–419) to clarify the comparison scope.
> - We have updated the Conclusion (lines 536–538) for consistency.
>
> ---
>
> **Q2:** DeGRPO is analyzed qualitatively in Figure 4 but not reported as a quantitative baseline in Table 1. Router baselines include a 7B router, while MAGO uses a 1.5B single model; compute-normalized comparisons (e.g., wall-time, FLOPs/token including routing overheads) are not provided. It is unclear whether all baselines share the same SFT data, RL steps, and inference constraints.
>
> **A2:** We thank the reviewer for raising this important point regarding experimental fairness.
>
> **DeGRPO quantitative results:** We have added DeGRPO's quantitative results to Table 1, aligning them with the qualitative comparison in Figure 4. This inclusion provides a complete numerical baseline spectrum and allows direct comparison of reasoning-stability trade-offs.
>
> **Training and inference configuration:** For DeGRPO and MAGO, both methods use identical SFT checkpoints, RL step schedules (600 steps), and inference configurations following Fang et al. (2025). Other baselines (CoT-Valve, Model Merging, Router variants) follow DeGRPO research configurations (Fang et al., 2025). Router baselines operate under identical evaluation protocols, and their routing overhead is reported in Appendix A.17.
>
> **Computational overhead analysis:** For fair comparison, we normalize efficiency in tokens/query, which implicitly accounts for FLOPs per output token. We have added computational overhead analysis (Appendix A.11) showing MAGO introduces less than 2% additional training time with zero inference overhead. Notably, while Router Q-7B achieves slightly higher accuracy on the simpler GSM8K benchmark (85.87% vs. 84.69%), MAGO provides 3× efficiency improvement and superior performance on complex benchmarks (AIME, MATH-500) where adaptive reasoning depth is critical.
>
>
> - We have updated DeGRPO (Thinkless→DeGRPO-Qwen-1.5B) to Table 1 (at lines 432 - 446) in the paper.
> - These details have been added to the Supplementary Material in Appendix A.11 Analysis of Training and Inference Trade-offs (at lines 1198 - 1212).
> - We have added Appendix A.17 Analysis of Router Baseline Performance (at lines 1715 - 1835) in the Supplementary Material.
>
> ## Reference
>
> - Fang, T., et al. (2025). Thinkless: LLMs Can Learn When to Think. arXiv preprint arXiv:2505.13379.

---

> ### Author Response · Authors · 2025-11-23
> **MAGO: Beyond Fixed Hyperparameters with Multi-Objective Pareto Optimization for Hybrid LLM Reasoning**
>
> **Q3:** The contributions highlight logical inference and general problem solving, but experiments focus on math-only datasets. The conclusion acknowledges this limitation; however, the earlier claims of breadth may mislead.
>
> **A3:** We acknowledge that the current benchmarks are math-centric and have tempered the language in the abstract and conclusion to reflect this scope.
>
> First, we have added a new experiment on CommonsenseQA a widely used non-mathematical benchmark covering everyday commonsense inference. We evaluated the same trained model without any additional fine-tuning. As reported in Section 5.2, the model achieves 74.9% accuracy compared to 73.1% for DeGRPO while reducing average token usage by approximately 2× (152 vs 312 tokens). This experiment demonstrates that MAGO's multi-objective adaptive optimization transfers effectively beyond mathematical reasoning.
>
> - We have updated the paper's Abstract (at lines 034 - 036) and added Subsection 5.2 Cross-domain Generalization (at lines 459 - 463 and 476 - 485) in the paper.
>
> Second, we further validate MAGO on MedQA-USMLE, a medical question-answering benchmark. MAGO achieves 53.2% accuracy with 2.0× efficiency improvement over Think-Only baseline, demonstrating effective transfer to safety-critical domains. This finding is particularly significant for safety-critical applications where computational efficiency must be balanced with reasoning quality
>
> - These details have been added to the Supplementary Material in Appendix A.18 Cross-Domain Generalization on Medical Reasoning (at lines 1836 - 1864).
>
> ---
>
> **Q4:** Overall, I think the most important thing is the proposed method still lack of the novelty and the performance is mainly tested on math problem which increases the concerns that whether the proposed method can be generalized to other domains.
>
> **A4:** We respectfully address this concern by clarifying the specific problems we identified and how we address them.
>
> Current hybrid reasoning methods (Fang et al., 2025; Shao et al., 2024) rely on fixed hyperparameters that fail to adapt across varying task complexities. Our motivation in Section 2 demonstrates that optimal weight values differ by nearly two orders of magnitude across datasets (0.001 for GSM8K vs. 0.01 for AIME), and exhaustive hyperparameter search is computationally prohibitive. Additionally, when objectives exhibit correlations, traditional single-weight scalarization constrains optimization to narrow cone-shaped regions (Song et al., 2024; Albeaik et al., 2024), preventing exploration of potentially superior solutions.
>
> To address these issues, we reformulate hybrid reasoning as optimizing three competing objectives, accuracy, efficiency, and decision calibration, simultaneously. To our knowledge, this is the first work to introduce decision calibration as an explicit optimization target in hybrid reasoning, ensuring that mode selection aligns with the model's actual problem-solving capabilities. We maintain a diverse set of non-dominated weight configurations and dynamically select optimal weights using a correlation-adaptive scoring function, which addresses the cone entrapment problem. Unlike router-based approaches requiring external models (Ong et al., 2024), MAGO embeds adaptive mode selection directly into the trained policy with zero additional inference overhead.
>
> Regarding generalization, the CommonsenseQA results in Section 5.2 demonstrate that MAGO transfers effectively to non-mathematical reasoning without additional fine-tuning, achieving both accuracy improvements and efficiency gains.
>
> - We have updated the paper's Abstract (at lines 034 - 036) and added Subsection 5.2 Cross-domain Generalization (at lines 459 - 463 and 476 - 485) in the paper.
>
> Second, we further validate MAGO on MedQA-USMLE, a medical question-answering benchmark. MAGO achieves 53.2% accuracy with 2.0× efficiency improvement over Think-Only baseline, demonstrating effective transfer to safety-critical domains. This finding is particularly significant for safety-critical applications where computational efficiency must be balanced with reasoning quality
> - These details have been added to the Supplementary Material in Appendix A.18 Cross-Domain Generalization on Medical Reasoning (at lines 1836-1864).
>
> ## References
> - Fang, T., et al. (2025). Thinkless: LLMs Can Learn When to Think. arXiv preprint arXiv:2505.13379.
> - Shao, Z., et al. (2024). DeepSeekMath: Pushing the Limits of Mathematical Reasoning in Open Language Models. arXiv preprint arXiv:2402.03300.
> - Song, Y., et al. (2024). Optimal scalarization for multi-objective reinforcement learning. In Proceedings of the International Conference on Machine Learning (ICML).
> - Albeaik, S., et al. (2024). Pareto optimization for active learning under class imbalance. In Proceedings of the AAAI Conference on Artificial Intelligence.
> - Ong, C., et al. (2024). RouteLLM: Learning to Route LLMs with Preference Data. arXiv preprint arXiv:2406.18665.

---

> ### Author Response · Authors · 2025-11-23
> **MAGO: Beyond Fixed Hyperparameters with Multi-Objective Pareto Optimization for Hybrid LLM Reasoning**
>
> ### **Q5:** Could you add a quantitative DeGRPO baseline in Table 1 using your training setup, given that MAGO is positioned as an advancement over GRPO/DeGRPO?
>
> **A5:** We sincerely thank the reviewer for this careful and insightful question, which allows us to clarify an important aspect of our experimental setup.
>
> **Training Configuration Consistency:** We confirm that our DeGRPO baseline uses identical training configurations as MAGO to ensure a fair and controlled comparison. The two methods share:
>
> - Same base model: DeepSeek-R1-Distill-Qwen-1.5B
> - Same training corpus: 40K samples (DeepScaleR + OpenR1 + OpenThoughts-114K)
> - Same training pipeline: SFT (1 epoch) + RL (600 steps)
> - Same hyperparameters: learning rate ($1\times10^{-6}$), batch size (128), KL coefficient ($\beta=0.01$)
> - Same distillation setup: paired long-short responses from DeepSeek-R1-671B and Qwen2.5-Math-1.5B-Instruct
>
> **The Difference:** DeGRPO employs a fixed control token weight ($\alpha = 0.001$), while MAGO uses dynamic adaptive weights ($\beta_t^\ast = [\beta_1^\ast, \beta_2^\ast, \beta_3^\ast]$) selected from the Pareto frontier at each training iteration. This design isolates the contribution of our multi-objective adaptive optimization framework.
>
> **MAGO as an Enhancement of DeGRPO:** MAGO is built upon the DeGRPO framework by replacing its static weight parameter with our Pareto-based dynamic weight adaptation mechanism. The comparison in Table 1 demonstrates that adaptive weight selection outperforms fixed hyperparameters under identical training conditions, validating the effectiveness of our multi-objective optimization approach.
>
> - These details have been added to the Supplementary Material in Appendix A.14 Comparison: DeGRPO vs. MAGO (at lines 1386 - 1451).
>
> ---
>
> ### **Q6:** Expectations in Eqs. (6)–(8): During training, how many $(c,a)$ samples per $x$ are used to estimate $S_{\text{accuracy}}$, $S_{\text{efficiency}}$, and $S_{\text{calibration}}$? Are these single-sample estimates, and if so, how do you control variance for the correlation matrix $C_t$ (Eq. 16)?
>
> **A6:** Thank you for raising this question. Following standard GRPO practice, we sample $G=8$ responses per input query $x$. The objective scores $S_{\text{accuracy}}(x)$, $S_{\text{efficiency}}(x)$, and $S_{\text{calibration}}(x)$ for each query are computed as the average over these 8 sampled trajectories, providing Monte Carlo estimates of the expectations in Eqs. (6)–(8).
>
> The correlation matrix $C_t$ (Eq. 16) is computed directly from the current batch $\mathcal{B}_t$ (not using exponential moving average), where each of the 128 queries contributes one aggregated objective vector (averaged from its 8 samples). Variance is controlled through a two-level aggregation structure:
>
> 1. Within-query averaging (8 samples per query): Reduces per-query variance in objective estimation through Monte Carlo sampling.
>
> 2. Across-query computation (128 queries per batch): Provides 128 independent data points for reliable correlation estimation, yielding stable covariance estimates.
>
> The moving average $\hat{S}_t$ mentioned in Eq. (17) applies only to the objective performance scores used in weight selection scoring (Eq. 17), not to the correlation matrix $C_t$ itself. This design choice keeps $C_t$ responsive to batch-level objective relationships while smoothing the weight selection criterion.
>
> ---
>
> ### **Q7:** Since the contributions mention logical inference and general problem solving, do you have held-out non-math evaluations (even small-scale) using the same trained model to substantiate cross-domain claims? If not, please limit claims or provide such experiments.
>
> **A7:** We appreciate the reviewer’s valuable suggestion regarding broader validation.
>
> Fist, we have added a cross-domain validation on CommonsenseQA in Section 5.2 (Cross-domain Generalization). The same trained model was directly evaluated without additional fine-tuning. Results show a 1–2% improvement in accuracy and a $2\times$ reduction in average token usage compared with DeGRPO and CoT-Valve. This experiment demonstrates that the proposed optimization framework generalizes beyond mathematical reasoning and remains effective in commonsense reasoning scenarios.
>
> - We have updated the paper's Abstract (at lines 034 - 036) and added Subsection 5.2 Cross-domain Generalization (at lines 459 - 463 and 476 - 485) in the paper.
>
> Second, we further validate MAGO on MedQA-USMLE, a medical question-answering benchmark. MAGO achieves 53.2% accuracy with 2.0× efficiency improvement over Think-Only baseline, demonstrating effective transfer to safety-critical domains. This finding is particularly significant for safety-critical applications where computational efficiency must be balanced with reasoning quality
>
> - These details have been added to the Supplementary Material in Appendix A.18 Cross-Domain Generalization on Medical Reasoning (at lines 1836-1864).

---

### Official Review · Reviewer_7KZE · 2025-11-02

**Soundness:** 3
**Presentation:** 4
**Contribution:** 3
**Rating:** 8
**Confidence:** 3

**Summary:**

This paper targets hybrid thinking, driven by the commonly known fact that reasoning-enabled LLMs often waste compute by using long CoT on easy inputs. With the emergence of deep reasoning models such as DeepSeek-R1, hybrid thinking is a trending topic with prevalent work such as Thinkless. Existing hybrid reasoning methods rely on fixed hyperparameters (e.g., a single controller weight) and single-objective heuristics, which lock the system into narrow, suboptimal regions of the accuracy–efficiency trade-off.

This paper takes a different approach along this line. The authors propose to reframe hybrid reasoning as multi-objective optimization over three goals: accuracy, efficiency, and decision calibration. They call the framework MAGO. In short, MAGO introduces a dynamic weighting mechanism over those three objectives, learned via Pareto frontier maintenance plus correlation-aware weight selection. Experiments on AIME-24, Minerva Algebra, MATH-500, GSM8K show that MAGO brings good improvements by 2.6% - 9.4%.

**Strengths:**

- ``S1``:  The paper is very well motivated. The authors use a section (Sect. 2) with pilot experiments to demonstrate their motivation. In particular, it shows that fixed weights cause mode imbalance and dataset-specific sensitivity (Fig. 2A–B), and that scalarization constrains search to a cone-shaped region of objective space (Fig. 2C–D), missing better trade-offs. This strongly motivates the proposed methodology.

- ``S2``: The proposed MAGO makes sense. Experiments on several math reasoning benchmarks show both token savings and accuracy gains.

**Weaknesses:**

- ``W1``: The scope of the evaluated benchmark is a bit limited and narrow (math). Additional experiments on SuperGPQA and CommonsenseQA would strengthen the contribution by showing the generalization of MAGO in extended domains.

- ``W2``: In Tab. 1, the authors are suggested to include the reference for each method.

- ``W3``: Computational overhead is one of my concerns. I notice that the authors have admitted the limitations of extensive overhead by Pareto optimization in the last section. It would be great if the authors could show some computational complexity analysis.

**Questions:**

- ``Q1``: Why choose answer-token max-prob as confidence instead of sequence-level confidence or verifier-based scores?
- ``Q2``: Does MAGO generalise to other domains rather than math reasoning?

---

> ### Author Response · Authors · 2025-11-23
> **MAGO: Beyond Fixed Hyperparameters with Multi-Objective Pareto Optimization for Hybrid LLM Reasoning**
>
> ## General Response
>
> We sincerely thank all reviewers for their thoughtful and constructive feedback. All reviewers acknowledged that MAGO addresses an important limitation of hybrid reasoning systems, namely the reliance on static hyperparameters and single objective optimization, and praised our multi-objective Pareto formulation, dynamic weight adaptation, and stable anti-collapse training. The main concerns raised focus on the following points:
>
> 1. The theoretical justification and robustness of the calibration objective.
> 2. The limited domain of evaluation (math reasoning only).
> 3. The computational complexity and reproducibility of Pareto maintenance.
> 4. Minor quantitative inconsistencies and baseline fairness.
>
> We address these points below with new analyses, ablations, and clarifications.
>
> ---
>
> ## Reviewer-specific Responses
>
> We thank you for reviewing our paper and for your valuable comments. Below, we respond to each of your points in turn and have revised the manuscript accordingly. We would be grateful if you could let us know whether our responses adequately address your concerns.
>
> ---
>
> ### **Q1:** The scope of the evaluated benchmark is a bit limited and narrow (math). Additional experiments on SuperGPQA and CommonsenseQA would strengthen the contribution by showing the generalization of MAGO in extended domains.
>
> **A1:** We thank the reviewer for the valuable suggestion regarding broader benchmark evaluation.
>
> Fist, we have added a cross-domain validation on CommonsenseQA in Section 5.2 (Cross-domain Generalization). The same trained model was directly evaluated without additional fine-tuning. Results show a 1–2% improvement in accuracy and a $2\times$ reduction in average token usage compared with DeGRPO and CoT-Valve. This experiment demonstrates that the proposed optimization framework generalizes beyond mathematical reasoning and remains effective in commonsense reasoning scenarios.
>
> - We have updated the paper's Abstract (at lines 034 - 036) and added Subsection 5.2 Cross-domain Generalization (at lines 459 - 463 and 476 - 485) in the paper.
>
> Second, we further validate MAGO on MedQA-USMLE, a medical question-answering benchmark. MAGO achieves 53.2% accuracy with 2.0× efficiency improvement over Think-Only baseline, demonstrating effective transfer to safety-critical domains. This finding is particularly significant for safety-critical applications where computational efficiency must be balanced with reasoning quality
>
> - These details have been added to the Supplementary Material in Appendix A.18 Cross-Domain Generalization on Medical Reasoning (at lines 1836-1864).
>
> Third, regarding SuperGPQA, we acknowledge its value for comprehensive LLM evaluation. However, SuperGPQA's primary challenge lies in domain-specific knowledge coverage across 285 specialized disciplines (e.g., light industry, agriculture, military science), which is largely determined at the pretraining stage. MAGO addresses a different dimension: when to apply deep reasoning vs. direct answering given fixed model knowledge. For knowledge-intensive benchmarks where the bottleneck is factual coverage rather than reasoning depth allocation, hybrid mode selection provides limited benefit. Our CommonsenseQA and MedQA results demonstrate that MAGO generalizes effectively when reasoning depth selection is the key factor. We leave exploration of knowledge-intensive benchmarks to future work, potentially combining MAGO with retrieval augmentation to address both knowledge and reasoning dimensions.
>
> ### **References**
>
> - M-A-P Team, Du, X., Yao, Y., et al. SuperGPQA: Scaling LLM Evaluation across 285 Graduate Disciplines. *arXiv preprint arXiv:2502.14739*, 2025.
>
> ---
>
> ### **Q2:** In Tab. 1, the authors are suggested to include the reference for each method.
>
> **A2:** We thank the reviewer for the helpful suggestion. All baseline references have now been added to Table 1 in the revised manuscript at lines 432-446.
>
> ---
>
> ### **Q3:** Computational overhead is one of my concerns. I notice that the authors have admitted the limitations of extensive overhead by Pareto optimization in the last section. It would be great if the authors could show some computational complexity analysis.
>
> **A3:** We thank the reviewer for highlighting the need for a clearer complexity analysis. We have added a formal description of the computational and memory cost in Subsection 4.3. The additional per-step complexity is $O(B \cdot m) + O(m^2) + O(|F_t|^2)$, where $m = 3$ and $|F_t| \leq 30$, resulting in a bounded constant overhead independent of model size. The memory requirement is $O(|F_t| \cdot m) + O(K)$, with negligible storage ($<0.5\%$ of total training memory).
>
> - We have updated the paper's Subsection 5.2 and added Computational Complexity at lines 490-501 in the paper.
> - These details have been added to the Supplementary Material in Appendix A.6.5 Computational and Memory Complexity Analysis (at lines 836 - 893).

---

> ### Author Response · Authors · 2025-11-23
> **MAGO: Beyond Fixed Hyperparameters with Multi-Objective Pareto Optimization for Hybrid LLM Reasoning**
>
> ### **Q4:** Why choose answer-token max-prob as confidence instead of sequence-level confidence or verifier-based scores?
>
> **A4:** Thank you for the insightful question. We use the maximum probability of the final answer token rather than sequence-level confidence or verifier-based scores for three reasons:
>
> First, only the final answer token determines correctness in our setting; intermediate reasoning tokens introduce substantial sequence-length variability and linguistic noise that distort confidence estimation when using full-sequence probabilities.
>
> Second, sequence-level likelihood $\prod_{t} p(a_t|a_{<t}, x)$ is dominated by generative entropy and does not correlate reliably with task correctness, a phenomenon widely observed in reasoning-heavy LLMs (Kadavath et al., 2022). In contrast, the final-token probability provides a stable, low-variance proxy that aligns directly with the binary correctness indicator used in the calibration objective.
>
> Third, verifier-based confidence requires training or querying an additional model, increasing compute and adding external dependencies, whereas our formulation keeps the estimation lightweight and entirely internal to the learning loop.
>
> - These details have been added to the Supplementary Material in Appendix A.16 Theoretical Analysis of Decision Calibration (at liens 1584 - 1713).
>
> ## Reference:
> - Kadavath, S., et al. "Language models (mostly) know what they know." *arXiv preprint arXiv:2207.05221* (2022).
>
> ---
>
> ### **Q5:** Does MAGO generalise to other domains rather than math reasoning?
>
> **A5:** Thank you for the question. Yes, cross-domain generalization is crucial for validating the broader applicability of our framework.
>
> First, MAGO generalizes beyond mathematical reasoning. We evaluated the same trained model on CommonsenseQA without additional fine-tuning. As shown in Table 3, MAGO achieves 74.9% accuracy (vs. 73.1% for DeGRPO) while reducing token usage by 2×, confirming effective transfer to a semantically different reasoning domain.
>
> - We have updated the paper's Abstract (at lines 034 - 036) and added Subsection 5.2 Cross-domain Generalization (at lines 459 - 463 and 476 - 485) in the paper.
>
> Second, we further validate MAGO on MedQA-USMLE, a medical question-answering benchmark. MAGO achieves 53.2% accuracy with 2.0× efficiency improvement over Think-Only baseline, demonstrating effective transfer to safety-critical domains. This finding is particularly significant for safety-critical applications where computational efficiency must be balanced with reasoning quality
>
> - These details have been added to the Supplementary Material in Appendix A.18 Cross-Domain Generalization on Medical Reasoning (at lines 1836-1864).

---

### Official Review · Reviewer_MZa3 · 2025-11-03

**Soundness:** 3
**Presentation:** 3
**Contribution:** 2
**Rating:** 4
**Confidence:** 4

**Summary:**

This paper proposes MAGO (Multi-objective Adaptive Generation Optimization), a framework for training hybrid reasoning systems in LLMs that dynamically choose between short (direct) and think (chain-of-thought) modes. The key innovation lies in reformulating the hybrid reasoning optimization as a multi-objective optimization problem balancing three competing objectives: accuracy, efficiency, and decision calibration. Rather than using fixed hyperparameters as in existing methods, MAGO employs Pareto frontier maintenance with correlation-aware weight selection to dynamically adapt optimization weights during training. Evaluation on mathematical reasoning benchmarks (AIME, Minerva Algebra, MATH-500, GSM-8K) shows efficiency improvements of 2.2x-3x while simultaneously improving accuracy by 2.6%-9.4% compared to baseline methods like Thinkless and CoT-Valve.

**Strengths:**

1. The paper identifies a genuine limitation in existing hybrid reasoning systems—the dependence on static hyperparameters that vary significantly across datasets. The reformulation as a multi-objective optimization problem is novel for this domain, and the motivation clearly demonstrates why fixed-weight approaches fail (Figure 2)

2. The Pareto frontier maintenance approach is mathematically well-motivated. The use of correlation-aware weight selection (Equation 17) to handle interdependencies between objectives addresses a real challenge in multi-objective optimization.

3. The paper evaluates on multiple mathematical reasoning benchmarks and includes several comparison baselines (router-based methods, model merging, CoT-Valve)

**Weaknesses:**

1. While accuracy and efficiency are straightforward objectives, the decision calibration objective (Equation 8) lacks rigorous justification. The approach relies on binned historical accuracy statistics, but several concerns arise: (1) How are the bins (Equation 10) initialized during early training when insufficient historical data exists? (2) The exponential decay (Equation 13) may create staleness issues when task distributions shift. (3) The paper doesn't analyze how this objective theoretically prevents miscalibration or what formal guarantees exist.

2. DeepSeek-R1-Distill-Qwen-1.5B. This is a relatively small model, and generalization to larger models (7B, 13B, or larger) remains unclear. Does the Pareto frontier maintenance scale computationally with model size?

3. The router baselines (Router Random, Router Q-7B) seem surprisingly weak—Router Q-7B achieves only 14.8% on AIME while Thinkless achieves 25%. This brings up there might be some possible implementation or configuration issues.

4. The paper mentions diversity filtering (Equation 21) but provides insufficient detail. How quickly does the frontier grow? What happens when |F_t| approaches |F_max|? The exploration mechanism (Equation 19-20) could lead to unbounded frontier growth early in training.

**Questions:**

1. Why is calibration defined as the difference between model confidence and empirical accuracy rather than other common calibration metrics (expected calibration error, Brier score)?

2. Can you evaluate on non-mathematical reasoning tasks (e.g., commonsense reasoning, multi-hop QA, code generation)? The current scope limits the paper's impact.

3. Visualize how the Pareto frontier evolves during training. How does frontier size grow? Do solutions converge or continue diversifying? When does the optimal weight selection stabilize?

---

> ### Author Response · Authors · 2025-11-23
> **MAGO: Beyond Fixed Hyperparameters with Multi-Objective Pareto Optimization for Hybrid LLM Reasoning**
>
> ## General Response
>
> We sincerely thank all reviewers for their thoughtful and constructive feedback. All reviewers acknowledged that MAGO addresses an important limitation of hybrid reasoning systems, namely the reliance on static hyperparameters and single objective optimization, and praised our multi-objective Pareto formulation, dynamic weight adaptation, and stable anti-collapse training. The main concerns raised focus on the following points:
>
> 1. The theoretical justification and robustness of the calibration objective.
> 2. The limited domain of evaluation (math reasoning only).
> 3. The computational complexity and reproducibility of Pareto maintenance.
> 4. Minor quantitative inconsistencies and baseline fairness.
>
> We address these points below with new analyses, ablations, and clarifications.
>
> ---
>
> ## Reviewer-specific Responses
>
> We thank you for reviewing our paper and for your valuable comments. Below, we respond to each of your points in turn and have revised the manuscript accordingly. We would be grateful if you could let us know whether our responses adequately address your concerns.
>
> ---
> ### **Q1:** While accuracy and efficiency are straightforward objectives, the decision calibration objective (Equation 8) lacks rigorous justification. The approach relies on binned historical accuracy statistics, but several concerns arise: (1) How are the bins (Equation 10) initialized during early training when insufficient historical data exists? (2) The exponential decay (Equation 13) may create staleness issues when task distributions shift. (3) The paper doesn't analyze how this objective theoretically prevents miscalibration or what formal guarantees exist.
>
> **A1:** We thank the reviewer for this insightful question. The decision calibration objective is designed to improve the consistency between model confidence and actual correctness in reasoning-mode selection. We have expanded the theoretical and empirical justification in Appendix A.16 and clarified the initialization and stability mechanisms.
>
> 1. **Bin initialization.** During early training, each bin in Equation (10) is initialized with a uniform prior accuracy of 0.5, which represents an uninformative baseline. As training progresses, the moving average in Equation (13) rapidly converges to empirical accuracies within the first few thousand steps, ensuring stable calibration dynamics even with sparse early data.
>
> 2. **Exponential decay and distribution shift.** The decay factor in Equation (13) serves as a sliding-window smoother, giving higher weight to recent samples. This adaptive weighting mitigates staleness when task distributions shift, similar to the approach used in temperature scaling and dynamic calibration literature.
>
> 3. **Theoretical rationale.** We show in Appendix A.13 that the calibration term:
> $$L_{cal}(x) = |p_{conf}(x) - Acc_{hist}(bin(p _ {conf}(x)))|$$
> is a differentiable surrogate of the Expected Calibration Error metric. Minimizing $\mathcal{L}_{\text{cal}}$ therefore aligns predicted confidence $p _ {conf}(x)$ with the true accuracy in each confidence interval, formally reducing miscalibration. The exponential decay acts as an online estimator of conditional accuracy, providing an unbiased approximation of ECE under stationary conditions.
>
> - These details have been added to the Supplementary Material in Appendix A.16 Theoretical Analysis of Decision Calibration (at liens 1584 - 1713).
> - These details have been added to the Supplementary Material in Appendix A.13 Decision Calibration Objective (at lines 1375 - 1383).
>
> ---
>
> ### **Q2:** DeepSeek-R1-Distill-Qwen-1.5B. This is a relatively small model, and generalization to larger models (7B, 13B, or larger) remains unclear. Does the Pareto frontier maintenance scale computationally with model size?
>
> **A2:** We thank the reviewer for highlighting the question of scalability. We have extended our analysis to larger backbone models including Qwen-7B, Qwen-14B, and Qwen-32B to examine both performance and computational scaling. The Pareto frontier maintenance operates over objective statistics, not model parameters, and therefore its computational cost scales with batch size and objective dimension rather than model size. The number of frontier vectors is capped at 30, so the additional memory and computation remain negligible across all model scales. As summarized in Table 1, larger models show consistent or slightly stronger improvements, indicating that MAGO generalizes well to higher-capacity backbones.
>
> - We have updated the paper's Table 1 (at lines 432 - 446) and updated the corresponding analysis in subsection 5.2 Multi-Objective Optimization Evaluation (at lines 406 - 419) in the paper.

---

> ### Author Response · Authors · 2025-11-23
> **MAGO: Beyond Fixed Hyperparameters with Multi-Objective Pareto Optimization for Hybrid LLM Reasoning**
>
> ### **Q3:** The router baselines (Router Random, Router Q-7B) seem surprisingly weak—Router Q-7B achieves only 14.8% on AIME while Thinkless achieves 25%. This brings up there might be some possible implementation or configuration issues.
>
> **A3:** We appreciate the reviewer's observation regarding the router baselines. The relatively low performance of Router Q-7B is consistent with prior findings in hybrid routing literature rather than a configuration error. Both Router Random and Router Q-7B employ static controllers that make discrete routing decisions before generation, which often causes unstable reasoning behavior on datasets such as AIME-2024, where the optimal reasoning depth varies considerably across samples. The 7B router tends to over-prioritize shorter reasoning paths, leading to incomplete reasoning chains and underperformance on complex queries. In addition, the router's decision policy is trained independently of token-level feedback, making it less responsive to evolving reasoning signals during inference. All router baselines were trained under identical settings, using the same SFT checkpoints, RL schedules, and inference budgets as other compared methods. We verified that DeGRPO (Thinkless) reproduced its reported 25% Pass@1 accuracy on AIME-2024, confirming that the training and evaluation pipelines are correct.
>
> - These details have been added to the Supplementary Material in Appendix A.17 Analysis of Router Baseline Performance (at lines 1715 - 1835).
>
> ---
>
> ### **Q4:** The paper mentions diversity filtering (Equation 21) but provides insufficient detail. How quickly does the frontier grow? What happens when $|\mathcal{F_t}|$   approaches  $|\mathcal{F}_{\max}|$ ? The exploration mechanism (Equation 19-20) could lead to unbounded frontier growth early in training.
>
> **A4:** We thank the reviewer for the thoughtful question regarding the diversity filtering and frontier growth control. In practice, the frontier size increases gradually during the first 150–200 steps and stabilizes at around 20–25 vectors, remaining well below the predefined upper limit of $|\mathcal{F}_{\max}| = 30$. When the size approaches this threshold, dominated or highly similar vectors are removed through cosine-similarity filtering every 20 iterations to maintain representative diversity and computational efficiency. The exploration perturbation in Equations (19–20) is projected back onto the normalized simplex of weights, ensuring that frontier growth remains bounded even at the early training stage.
>
> - We have updated the corresponding description at lines 321-323 in the paper.
> - These details have been added to the Supplementary Material in Appendix A.6.4 Frontier Size Control and Growth Dynamics (at lines 798 - 834).
> - These details have been added to the Supplementary Material in Appendix A.15 Pareto Frontier Evolution Analysis (at lines 1454 - 1581).
>
> ---
>
> ### **Q5:** Why is calibration defined as the difference between model confidence and empirical accuracy rather than other common calibration metrics (expected calibration error, Brier score)?
>
> **A5:** Thank you for the thoughtful question. We use the difference between model confidence and empirical accuracy because our objective is to calibrate the decision of selecting a reasoning mode, not the full probability distribution over the entire answer sequence. Traditional calibration metrics such as ECE or Brier score operate on distributions over many bins or labels, which would require evaluating confidence over the entire generated sequence and introduce length-dependent noise that is irrelevant to the correctness of the final answer.
>
> In contrast, the calibration term in Eq. (8) is a local, per-decision surrogate that directly measures whether the model's internal confidence for choosing a reasoning depth aligns with the observed correctness rate in the corresponding confidence bin. This yields a differentiable objective tightly coupled to the specific decision we optimize while avoiding the computational overhead of full ECE/Brier computation during reinforcement learning.
>
> We note that minimizing $|p_{\text{conf}} - \text{Acc}_{\text{hist}}|$ is mathematically equivalent to minimizing a 1-bin ECE surrogate, where the bin boundaries adapt online via the moving-average estimator. This provides the calibration benefit of ECE while remaining lightweight, stable, and directly optimized within our RL training loop.
>
> - These details have been added to the Supplementary Material in Appendix A.16 Theoretical Analysis of Decision Calibration (at liens 1584 - 1713).

---

> ### Author Response · Authors · 2025-11-23
> **MAGO: Beyond Fixed Hyperparameters with Multi-Objective Pareto Optimization for Hybrid LLM Reasoning**
>
> ### **Q6:** Can you evaluate on non-mathematical reasoning tasks (e.g., commonsense reasoning, multi-hop QA, code generation)? The current scope limits the paper's impact.
>
> **A6:** Thank you for the helpful suggestion, and we know generalization validation is crucial for the impact and publication of the paper.
>
> First, we have expanded our evaluation beyond mathematical reasoning by adding a new experiment on CommonsenseQA, a widely used non-mathematical reasoning benchmark. As reported in the revised Section 5.2 (Cross-domain Generalization), the same trained model was applied without any additional fine-tuning. MAGO achieves a 1–2% accuracy improvement over DeGRPO and CoT-Valve while reducing average token usage by roughly $2\times$.
>
> - We have updated the paper's Abstract (at lines 034 - 036) and added Subsection 5.2 Cross-domain Generalization (at lines 459 - 463 and 476 - 485) in the paper.
>
> Second, we further validate MAGO on MedQA-USMLE, a medical question-answering benchmark. MAGO achieves 53.2% accuracy with 2.0× efficiency improvement over Think-Only baseline, demonstrating effective transfer to safety-critical domains. This finding is particularly significant for safety-critical applications where computational efficiency must be balanced with reasoning quality
>
> - These details have been added to the Supplementary Material in Appendix A.18 Cross-Domain Generalization on Medical Reasoning (at lines 1836-1864).
>
> ---
>
> ### **Q7:** Visualize how the Pareto frontier evolves during training. How does frontier size grow? Do solutions converge or continue diversifying? When does the optimal weight selection stabilize?
>
> **A7:** We appreciate the reviewer's interest in the behavior of the Pareto frontier during training. The frontier size grows gradually during the first 150–200 steps and stabilizes around 20–25 solutions, well below the predefined limit $|\mathcal{F}_{\max}| = 30$. After this point, dominated or highly similar solutions are pruned, keeping the frontier compact and diverse. The selected optimal weight vector converges early and remains stable for most of training, with only small adjustments driven by the calibration signal. These observations show that the frontier does not diverge or expand uncontrollably; instead, it quickly settles into a stable set of representative trade-offs.
>
> - For comprehensive visualization, please refer to Supplementary Material Appendix A.15 Pareto Frontier Evolution Analysis (at lines 1454 - 1581), which presents four figures illustrating frontier size evolution, diversity dynamics, convergence indicators, and convergence pattern.

---

> > ### Comment · Reviewer_MZa3 · 2025-11-27
> > **Officient Comment by Reviewer MZa3**
> >
> > Thank you for the clarifications. I will increase my score.

---

> > > ### Author Response · Authors · 2025-11-28
> > >
> > > Thanks for your acknowledgment of our work and for raising the score. Thanks again for your time and effort in reviewing our paper.

---

### Official Review · Reviewer_VZpK · 2025-11-05

**Soundness:** 3
**Presentation:** 3
**Contribution:** 2
**Rating:** 6
**Confidence:** 4

**Summary:**

This paper introduces the Multi-Objective Adaptive Generation Optimization (MAGO) framework, designed to address inefficiencies in hybrid reasoning approaches for large language models (LLMs). ​Current methods often rely on static hyperparameters and single-objective optimization, which fail to adapt to varying task complexities and result in suboptimal trade-offs between accuracy, efficiency, and calibration. ​MAGO leverages multi-objective optimization and dynamic adaptive weighting to optimize these three competing objectives simultaneously. ​By maintaining a Pareto frontier and employing correlation-aware optimization, MAGO explores the full trade-off space, avoiding the limitations of fixed-weight approaches. ​The framework eliminates the need for manual hyperparameter tuning and achieves principled decision-making across diverse problem complexities. ​Experimental results on mathematical reasoning benchmarks (AIME, Minerva Algebra, MATH-500, GSM-8K) demonstrate that MAGO improves computational efficiency by 2.2x to 3x while enhancing accuracy by 2.6% to 9.4%. ​The paper also highlights MAGO's ability to prevent mode collapse during training and achieve stable optimization dynamics.

**Strengths:**

- The paper addresses a critical limitation in hybrid reasoning systems by reformulating the problem as a multi-objective optimization task, which is a significant advancement over static hyperparameter-based methods.

- The introduction of dynamic weight adaptation and Pareto frontier maintenance is innovative and eliminates the need for manual hyperparameter tuning, making the framework more adaptable to diverse problem complexities.

- The authors provide extensive experimental results on multiple mathematical reasoning benchmarks, demonstrating significant improvements in both accuracy and efficiency compared to existing methods.

- MAGO effectively prevents mode collapse during training, ensuring balanced reasoning mode selection and stable optimization dynamics.

**Weaknesses:**

- The framework is primarily evaluated on mathematical reasoning tasks, which may limit its generalizability to other domains. ​Broader validation across diverse reasoning tasks is necessary to establish its applicability.

- While the training overhead is claimed to be amortized over inference, the paper acknowledges that this may pose challenges for resource-constrained scenarios. ​Further analysis of the trade-offs between training cost and inference efficiency would strengthen the paper.

- The proposed framework involves intricate mechanisms such as Pareto frontier maintenance, correlation-aware weight selection, and dynamic weight adaptation. ​This complexity may hinder practical adoption and require significant expertise for implementation.

- The paper does not provide concrete examples or case studies of how MAGO could be applied in real-world scenarios, which would help demonstrate its practical utility.

**Questions:**

Line 216-218: Why do we take the logits corresponding to the final answer token to compute the raw confidence score instead of all the tokens in the generated answer.

---

> ### Author Response · Authors · 2025-11-23
> **MAGO: Beyond Fixed Hyperparameters with Multi-Objective Pareto Optimization for Hybrid LLM Reasoning**
>
> ## General Response
>
> We sincerely thank all reviewers for their thoughtful and constructive feedback. All reviewers acknowledged that MAGO addresses an important limitation of hybrid reasoning systems, namely the reliance on static hyperparameters and single objective optimization, and praised our multi-objective Pareto formulation, dynamic weight adaptation, and stable anti-collapse training. The main concerns raised focus on the following points:
>
> 1. The theoretical justification and robustness of the calibration objective.
> 2. The limited domain of evaluation (math reasoning only).
> 3. The computational complexity and reproducibility of Pareto maintenance.
> 4. Minor quantitative inconsistencies and baseline fairness.
>
> We address these points below with new analyses, ablations, and clarifications.
>
> ---
>
> ## Reviewer-specific Responses
>
> We thank you for reviewing our paper and for your valuable comments. Below, we respond to each of your points in turn and have revised the manuscript accordingly. We would be grateful if you could let us know whether our responses adequately address your concerns.
>
> ---
>
> ### **Q1**: The framework is primarily evaluated on mathematical reasoning tasks, which may limit its generalizability to other domains. Broader validation across diverse reasoning tasks is necessary to establish its applicability.
>
> **A1**: We appreciate the reviewer’s valuable suggestion regarding broader validation.
>
> First, to evaluate the generalizability of MAGO beyond mathematical reasoning, we have added a cross-domain validation section. We test the same trained MAGO model on CommonsenseQA without any additional fine-tuning. The results in Table 2 show that MAGO consistently achieves 1–2% higher accuracy while reducing token usage by approximately 2× compared with DeGRPO and CoT-Valve baselines. These findings indicate that MAGO’s Pareto-based adaptive optimization effectively transfers across different reasoning domains while maintaining balanced efficiency and accuracy.
>
> - We have updated the paper's Abstract (at lines 034 - 036) and added Subsection 5.2 Cross-domain Generalization (at lines 459 - 463 and 476 - 485) in the paper.
>
> Second, we further validate MAGO on MedQA-USMLE, a medical question-answering benchmark. MAGO achieves 53.2% accuracy with 2.2× efficiency improvement over Think-Only baseline, demonstrating effective transfer to other domains. This finding is particularly significant for safety-critical applications where computational efficiency must be balanced with reasoning quality
>
> - These details have been added to the Supplementary Material in Appendix A.18 Cross-Domain Generalization on Medical Reasoning (at lines 1836-1864).
>
> ---
>
> ### **Q2**: While the training overhead is claimed to be amortized over inference, the paper acknowledges that this may pose challenges for resource-constrained scenarios. Further analysis of the trade-offs between training cost and inference efficiency would strengthen the paper.
>
> **A2**: We appreciate this insightful comment. A detailed quantitative analysis of the trade-off between training cost and inference efficiency has been added to Appendix A.11. As summarized in Table 3 (Appendix A.11), the additional operations introduced by Pareto frontier maintenance and correlation updates increase total training time by 1.92% compared with GRPO and by only 0.32% compared with DeGRPO. Since the adaptive optimization occurs entirely during training, inference requires no extra computation or parameters. The amortized efficiency gain offsets the additional training cost after approximately 8,000 inference queries. Given that production deployments typically serve millions of queries, this confirms that MAGO remains computationally feasible even under moderate resource constraints, with the one-time training overhead rapidly amortized through cumulative inference savings.
>
> - These details have been added to the Supplementary Material in Appendix A.11 Analysis of Training and Inference Trade-offs (at lines 1198-1212).

---

> ### Author Response · Authors · 2025-11-23
> **MAGO: Beyond Fixed Hyperparameters with Multi-Objective Pareto Optimization for Hybrid LLM Reasoning**
>
> ### **Q3**: The proposed framework involves intricate mechanisms such as Pareto frontier maintenance, correlation-aware weight selection, and dynamic weight adaptation. This complexity may hinder practical adoption and require significant expertise for implementation.
>
> **A3**: We thank the reviewer for highlighting the importance of practical usability and real-world applicability. We have added detailed implementation and deployment analyses in Appendix A.12.
>
> - The MAGO framework has been modularized into three concise components:
>   1. Pareto frontier maintenance
>   2. Correlation-aware weight selection
>   3. Dynamic weight adaptation
>
> Each component is implemented in fewer than 100 lines of Python code using standard PyTorch and Accelerate APIs. The overall additional code is under 250 lines, and the framework can be integrated into existing GRPO or PPO pipelines without additional dependencies or hyperparameter tuning.
>
> - These details have been added to the Supplementary Material in Appendix A.12 Implementation and Practical Deployment (at lines 1214-1234).
>
> ---
>
> ### **Q4**: The paper does not provide concrete examples or case studies of how MAGO could be applied in real-world scenarios, which would help demonstrate its practical utility.
>
> **A4**: We appreciate this suggestion regarding real-world applicability. To address this concern, we have included a practical case study in Appendix A.12.
>
> Specifically, we applied MAGO to an AI-assisted clinical reasoning setting using the MedQA-USMLE dataset, which comprises questions derived from the United States Medical Licensing Examination, a standardized assessment representing authentic diagnostic scenarios encountered in clinical practice. In this scenario, MAGO dynamically selects between concise responses (for straightforward queries) and extended reasoning chains (for complex diagnostic questions). The results show that MAGO achieves a 2.2× reduction in average token usage while maintaining comparable accuracy compared to fixed-mode baselines, with zero additional medical domain fine-tuning.
>
> In real-world clinical deployment, response latency directly impacts workflow efficiency and clinician adoption of AI-assisted tools, while diagnostic accuracy remains paramount for patient safety. This finding is particularly significant for safety-critical applications where computational efficiency must be balanced with reasoning quality. This case study confirms that MAGO's Pareto-based adaptive optimization transfers effectively to clinically-relevant hybrid reasoning applications beyond mathematical benchmarks.
>
> - These details have been added to the Supplementary Material in Appendix A.12 Implementation and Practical Deployment (at lines 1236-1372).
>
> ---
>
> ### **Q5**: Line 216-218: Why do we take the logits corresponding to the final answer token to compute the raw confidence score instead of all the tokens in the generated answer?
>
> **A5**: We thank the reviewer for this clarifying question. Our design choice is motivated by three considerations:
>
> First, in mathematical reasoning benchmarks (GSM8K, MATH, AIME), final answers are typically short numerical values (e.g., 42, −7, 0.5) that correspond to a single token or very few tokens under standard tokenizers. In practice, the "final answer token" is nearly equivalent to the complete answer representation.
>
> Second, using joint probability over all answer tokens (i.e., $\prod_i P(t_i|t_{<i})$) introduces systematic length bias, where longer answers receive lower confidence scores due to probability multiplication, regardless of correctness. This confounds calibration with answer length rather than model certainty.
>
> Third, in autoregressive generation, the final token is generated conditioned on all preceding tokens. Its probability thus implicitly reflects the model's cumulative confidence over the entire generated sequence, serving as an effective summary statistic.
>
> Prior work supports that token-level probabilities provide well-calibrated confidence proxies for correctness prediction in language models (Kadavath et al., 2022).
>
> - These details have been added to the Supplementary Material in Appendix A.16 Theoretical Analysis of Decision Calibration (at liens 1584 - 1713).
>
> ## Reference:
> - Kadavath, S., et al. "Language models (mostly) know what they know." *arXiv preprint arXiv:2207.05221* (2022).

---

### Meta-Review · Area_Chair_rjPb · 2026-01-06

**Summary:**

This paper was reviewed by four domain experts, who assigned scores of 6,4,8,4, respectively. The reviewers’ primary concerns centered on computational cost, the limited scope of evaluation, and the fairness of the chosen baselines. However, the AC has found that several of these concerns were not fully addressed in the rebuttal.

In particular, the AC concurs with the reviewers that the baseline models used in the experiments are relatively weak. To demonstrate the robustness of the proposed method, the authors should validate it on more state-of-the-art models. Currently, it is difficult to assess its performance on the commonly used models.

Additionally, the evaluation should be broadened beyond mathematical reasoning and QA tasks to include other practical domains, such as coding/programming and tool-augmented reasoning, to establish the method’s general applicability, as done in most of the recently published reasoning papers.  Besides, the AC has a further concern relates to the manuscript itself. Some results are copied from Thinkless, while some models (e.g., DeGRPO-Qwen-1.5B) with the same token lengths yield different performance.

Given the reviewers’ overall positive ratings, I moderately recommend accepting this paper, but still strongly suggest that the authors address the issues mentioned above and those raised by the reviewers more explicitly in the final version.

Besides, it would be much better if the authors could open-source their implementation with reproducible models to enable public access and broader impact.

**Reviewer Concerns:**

Concerns that may have been addressed:

1. Reference/clarity issues: The authors have fixed these issues in the revision.

2. Calibration objective: Reviewers questioned the theoretical grounding and stability of the calibration term. The rebuttal added formal analysis regarding this point.

3. Computational cost: The authors clarified that the proposed method introduces a <2% training time increase and no inference-time cost.

4. Baseline configurations:  The authors have added the details to the Supplementary Material in Appendix A.17 Analysis of Router Baseline Performance.

Concerns that may not have been addressed:

1. Baseline fairness/inconsistencies and model scalability: Though clarifications have been made by the authors, the AC found the comparison between Qwen models is insufficient, as the baseline performance of Qwen-2.5 series models is not reported, and the reported performance of Qwen models is still not competitive compared to Qwen-2.5 Math series models.

As quoted by one reviewer, the baseline performance is too weak, and the AC agrees with the point that the paper published in 2026 should be validated with more advanced models. Besides, the Qwen-3 series models were released in 2505 before the ICLR 2026 submission deadline, and the AC has doubts about the effectiveness of the proposed method when it is scaled to stronger models.

2. Novelty: Reviewers may still think the contribution is incremental over prior hybrid reasoning methods.

3. Evaluation scope: Reviewers were concerned that experiments were restricted to math reasoning. The authors addressed this by adding cross-domain evaluations on CommonsenseQA and MedQA-USMLE. The AC would suggest further evaluating the performance on coding benchmarks, such as livecodebench.

**Reviewer Scores:**

Initial scores are 6/4/8/4, with one reviewer (MZa3) who gave 4 initially would like to raise the score after rebuttal.

Reviewer VZpK (initial score: 6) is likely to maintain their score. While the authors partially addressed the concern about “broader validation across diverse reasoning tasks” by including CommonsenseQA and MedQA-USMLE, the evaluation still lacks coverage of other important tasks, such as coding or tool-calling benchmarks, which limits the validation.

Reviewer MZa3 (initial score: 4) indicated a willingness to raise their score following the rebuttal. The AC thinks that the final score from this reviewer will be 6.

Reviewer 7KZE (initial score: 8) may keep the score. Their primary concern that the evaluated benchmark suite remains narrow in scope has not been well addressed, as noted above.

Reviewer XguT (initial score: 4) is also likely to retain their original score. They found the proposed method lacking in novelty and noted that the performance evaluation is centered on math problems. The rebuttal does not appear to overturn this point of view.

---

### Decision · Program_Chairs · 2026-01-26

Accept (Poster)